# Exploration of Novel Urolithin C Derivatives as Non-Competitive Inhibitors of Liver Pyruvate Kinase

**DOI:** 10.3390/ph16050668

**Published:** 2023-04-28

**Authors:** Umberto Maria Battisti, Leticia Monjas, Fady Akladios, Josipa Matic, Eric Andresen, Carolin H. Nagel, Malin Hagkvist, Liliana Håversen, Woonghee Kim, Mathias Uhlen, Jan Borén, Adil Mardinoğlu, Morten Grøtli

**Affiliations:** 1Department of Chemistry and Molecular Biology, University of Gothenburg, SE-412 96 Gothenburg, Swedenleticia.monjas@gmail.com (L.M.); fadydoc2000@yahoo.com (F.A.); josipa.matic@nofima.no (J.M.); carolin.nagel@chemie.uni-regensburg.de (C.H.N.); malin.hagkvist@hotmail.com (M.H.); 2Science for Life Laboratory, KTH—Royal Institute of Technology, SE-171 65 Stockholm, Swedenadilm@scilifelab.se (A.M.); 3Department of Molecular and Clinical Medicine, University of Gothenburg, SE-413 45 Gothenburg, Sweden; liliana.haversen@wlab.gu.se (L.H.); jan.boren@wlab.gu.se (J.B.); 4Sahlgrenska University Hospital, SE-413 45 Gothenburg, Sweden; 5Centre for Host-Microbiome Interactions, Faculty of Dentistry, Oral & Craniofacial Sciences, King’s College London, London SE1 9RT, UK

**Keywords:** NAFLD, liver pyruvate kinase, allosteric inhibitors, urolithin C, SAR studies

## Abstract

The inhibition of liver pyruvate kinase could be beneficial to halt or reverse non-alcoholic fatty liver disease (NAFLD), a progressive accumulation of fat in the liver that can lead eventually to cirrhosis. Recently, urolithin C has been reported as a new scaffold for the development of allosteric inhibitors of liver pyruvate kinase (PKL). In this work, a comprehensive structure–activity analysis of urolithin C was carried out. More than 50 analogues were synthesized and tested regarding the chemical features responsible for the desired activity. These data could pave the way to the development of more potent and selective PKL allosteric inhibitors.

## 1. Introduction

Non-alcoholic fatty liver disease (NAFLD) refers to the accumulation of fat in liver independent of alcohol consumption [1,2]. NAFLD is the most common chronic liver disease in the Western world, and is associated with the development of cardiovascular diseases and type 2 diabetes [3]. The overall global prevalence of NAFLD is approximately 25%, and it is estimated to grow in coming years [3,4]. However, no therapy is currently available to treat this disease; the only available options have been weight loss (e.g., calorie restriction, exercise) and supplementation of vitamins, e.g., vitamin E [5].

Recently, several groups identified liver pyruvate kinase (PKL) as a possible target to halt the progression of NAFLD [2,6,7]. Pyruvate kinase is responsible for the final step in glycolysis converting phosphoenolpyruvate (PEP) and adenosine diphosphate (ADP) to pyruvate (PYR) and adenosine triphosphate (ATP). Additionally, pyruvate kinase is present in four different isoforms, PKM1, PKM2, PKR, and PKL, each with multiple designations over time [8]. PKR is predominantly expressed in red blood cells, PKM1 is present in the skeletal muscle, brain, and heart, and PKM2 is expressed in proliferating cells, embryonic tissues, and tumors [9,10]. PKL is expressed mainly in the liver but also in pancreatic β-cells, the small intestine, and the renal proximal tubule [11]. Hence, development of PKL-specific inhibitors may be extremely beneficial for treating NAFLD.

Our group recently started an extensive program aimed at identifying new possible compounds to inhibit PKL [12,13,14]. These efforts resulted in the identification of two different classes of PKL inhibitors. The first, based on anthraquinone derivatives, behaved as competitive inhibitors of ADP, while the second, based on ellagic acid, shows a non-competitive inhibition of the enzyme based on the interaction with an allosteric site [12,14]. Deconstruction and simplification of ellagic acid identified one of its metabolites, urolithin C (**1**), as a more soluble and bioavailable PKL inhibitor. Here, we report the synthesis and structure–activity relationship (SAR) of novel PKL inhibitors based on compound **1** (Figure 1). The exploration, design, and synthesis were based on classical medicinal chemistry approaches (e.g., bioisosteric substitution, ring opening).

## 2. Results and Discussion

### 2.1. Deconstruction of Urolithin C

Previously, we identified compound **1** as a non-competitive inhibitor with a sub-micromolar activity for PKL (Table 1) but different selectivity for the PKR, PKM1, and PKM2 isoforms [14]. Considering that our previous effort to co-crystallize compound **1** with the enzyme failed, we decided to pursue a different approach. In particular, a deconstruction strategy to identify the key features for the activity of **1** was performed. The hydroxyl group and the catechol were replaced one at a time by a hydrogen atom. The lactone moiety was removed to give a biaryl compound. Compound **1** was synthesized as previously described [14], while **2** (urolithin A) was commercially available. The other compounds were synthesized as described below. Briefly, compound **6** was obtained by a Hurtley coupling from **3** and **4** and subsequent deprotection with BBr_3_ (Figure 1) [15,16].

The 3-deshydroxy analogue (**13**) was synthesized from **11** by K_2_S_2_O_8_-mediated C–H oxidation and deprotection [17]. Compound **11** was in turn obtained from 3,4-dimethoxybenzoic acid (**7**) in four steps by sequential bromination, protection, Suzuki coupling, and deprotection (Figure 2).

Finally, compound **17** was obtained from **14** and **15** by Pd-mediated coupling followed by deprotection with BBr_3_ (Figure 3).

The effects of derivatives on PKL activity were determined by an in vitro biochemical assay using recombinant PKL and PKR enzymes. The results are reported in Table 1. A significant decrease in the ability to inhibit PKL was observed upon the removal of the hydroxyl groups and the lactone moiety from **1**, indicating that all these features are essential for the activity. Interestingly, **2** and **17** showed weak PKL activation activity.

### 2.2. SAR Strategy

Based on the deconstruction of **1**, all the chemical features explored were necessary for the inhibitory activity vs. PKL. For these reasons, we explored and synthesized three series of urolithin C derivatives by modifying: (a) the phenol, (b) the catechol, and (c) the lactone (Figure 2). This strategy allowed us to study the impact of every single structural change on inhibition.

### 2.3. Replacement of the Phenol Group

Several substituents with differing electronic and steric properties were explored to replace the OH at position 3. A total of 19 analogues were obtained and tested for their inhibitory activity vs. PKL and PKR. The nitrogen analogues **26**, **28**, **33**–**36** were synthesized as described in Figure 4. Compound **9** was coupled under Pd catalysis with **18** or **19**. The corresponding biaryl derivatives were deprotected under basic hydrolysis and subjected to K_2_S_2_O_8_ or *N-*iodosuccinimide-mediated oxidative lactonization [17,18]. Compound **24** was then deprotected to obtain **26**. In a different process, **25** was reduced to result in the intermediate **27**, which was deprotected to afford the target compound **28**. The aniline **27** was further acylated with various reagents and deprotected with BBr_3_ to give **33**–**36**.

Derivatives **61**–**66** were similarly synthesized, as reported in Figure 5. Bromide **9** was coupled with the selected boronic acid, deprotected, and cyclized to give the corresponding lactones **55**–**60**. Deprotection of the methoxy groups with BB_3_ afforded the desired target molecules **61**–**66**.

Compounds **81**–**83** were obtained with minor modifications of the previous synthetic strategies, as shown in Figure 6. Aldehyde **67** was brominated and reacted under Suzuki coupling conditions to give **69**–**71**. These intermediates were oxidized and cyclized using AgNO_3_/K_2_S_2_O_8_ or Pd-catalyzed C-H activation [19]. Finally, deprotection yielded the urolithin derivatives **81**–**83**.

The methoxy derivative **90** and the acetyl ester **94** required a different approach as shown in Figure 7. Compound **8** was deprotected and alkylated with benzyl bromide to give **85**. The latter was coupled with **86**, deprotected, and cyclized to give **89**. Finally, removal of the benzyl groups with H_2_/Pd afforded **90**. The acetyl ester **94** was instead obtained by hydrolysis of **85** in basic media followed by Hurtley coupling, esterification, and finally, H_2_/Pd deprotection.

The final two urolithin derivatives, **98** and **103**, bearing an additional ring were synthesized as shown in Figure 8. Compound **98** was obtained by esterification of **91** and Heck intramolecular cyclization coupling followed by deprotection. Compound **103** instead was obtained again from **9** and the corresponding boronic acid with a four-analogue synthetic pathway similar to that shown above.

The effects of derivatives modified at the phenolic group are reported in Table 2. Most of the modifications and substitutions resulted in a marked loss of activity vs. PKL. Some compounds (e.g., **28**) showed a weak potentiation of PKL activity as observed for the deconstructed analogues of **1**. The IC_50_ values of **81** and **82** were similar to the IC_50_ of urolithin C, indicating that a carboxylic acid and an amide are tolerated.

### 2.4. Replacement of the Catechol Moiety

To further improve our understanding of the SAR of urolithin C as a PKL inhibitor, we then investigated the modifications of the catechol moiety. A total of 14 compounds were designed and synthesized to explore this part of the molecule. The dimethoxy analogue **118** and the two monomethoxy positional isomers **119** and **120** were obtained from the corresponding 2-bromobenzoic acid and resorcinol through an Hurtley coupling (Figure 9). The 2-bromobenzoic acids employed, **113** and **115**, were synthesized from the corresponding aldehyde by oxidation and deprotection. Differently, the 9- and 8-amino analogues (**124** and **126**) were obtained by Pd-mediated reduction and BBr_3_ deprotection of the key nitro intermediates **121** and **122**. The latter were synthesized again through Hurtley coupling from the respective acids. Compounds **127** and **128** were then isolated by simply acetylation/deprotection from **124** and **126**.

The 8,9-diamino analogue **135** was obtained as shown in Figure 10. 2-Bromo-4,5-dinitrobenzoic acid was reacted with benzylamine, esterified, and coupled with **131** to afford the desired biaryl compound **132**. Hydrolysis of the methyl ester followed by cyclization and reduction/deprotection yielded **135**. Its acetylated derivative **140** and the 2-methyl-1H-imidazole analogue **141** were synthesized similarly. Reduction of methyl 2-bromo-4,5-dinitrobenzoate (**110**) followed by acetylation and Suzuki coupling with the selected boronic acid gave **139**. One-step deprotection/cyclization yielded the desired compound **140**. Condensation of the latter in HCl afforded **141**.

The last four analogues with modifications of the catechol moiety were synthesized as reported in Figure 11. Methylenedioxy derivative **144** was obtained by alkylation of **84** with diiodomethane, followed by hydrolysis and Hurtley coupling. In a different process, compounds **150**, **153**, and **154** were synthesized from the key intermediate **136** by converting it into its 2-hydroxybenzimidazole, benzimidazole, and benzotriazole analogues (**145**–**147)**. Analogue **145** was hydrolyzed to its corresponding acid and reacted with resorcinol to produce **150**. In another synthesis, **147** was further protected to give **148**. Both **148** and **146** were transformed under Suzuki coupling conditions and finally deprotected and cyclized to yield **153** and **154**.

The synthesized compounds were then assessed using the in vitro biochemical assay (Table 3). Very low or no inhibition of enzyme activity was observed for most molecules with few exceptions. The two monomethoxy derivatives **119** and **120** showed almost full inhibition at 10 μM but with a dramatic reduction in their IC_50_ (1.9 μM and 1.5 μM, respectively). Even compound **141** retained some weak activity with an IC_50_ of 14 μM. Again, some small modifications (e.g., OH vs. NH_2_) resulted in the compounds being weak activators.

### 2.5. Replacement of the Lactone Moiety

Finally, we explored the impact of the lactone moiety on PKL inhibitions. We initially designed and prepared six analogues by switching the lactone position (**161**) or converting it to its corresponding lactam or methyl lactam (**165**, **167**). Alternatively, we removed the moiety altogether but kept the planarity (**178**) and H-bond acceptor properties (**179**, **180**), and increased the bulk (**183**). The lactone and lactam derivatives were obtained as shown in Figure 12. Compound **161** was obtained by esterification of the corresponding acid **155** and subsequent Suzuki coupling. Hydrolysis followed by cyclization and deprotection afforded the desired compound. Analogues **165** and **167** were synthesized converting acid **8** into the corresponding amide. The latter was coupled with 4-methoxyboronic acid, then **164** was obtained via copper-catalyzed cyclization [20]. This was either directly deprotected or methylated and deprotected to give the two target compounds.

The synthesis of phenanthrene **178** and phenanthridines **179** and **180** is reported in Figure 13. The corresponding benzaldehyde **68** was obtained as reported above while **169** was isolated after bromination of 3-methoxybenzaldheyde. These compounds were reacted with the selected boronic acids to produce **170** and **171**. The first compound was converted to **172** by a Wittig reaction and subjected to dehydrative cycloaromatization [21]. Deprotection with BBr_3_ gave the desired compound **178**. The analogues **179** and **180** were instead obtained converting the aldehyde to the phenanthridine core through iron-catalyzed intramolecular N-arylation [22]. Finally, deprotection with HBr in acetic acid gave **179** and **180**.

Finally, compound **183** was obtained from **1** by protecting the hydroxyl groups, reacting with CH_3_MgBr and final deprotection with H_2_/Pd (Figure 14). The compounds were then evaluated using an in vitro biochemical assay (Table 4).

Interestingly, switching the position of the lactone (**161**) resulted in a tremendous loss of activity. Similarly, the lactam derivatives (**165**, **167**) showed at least a 10-fold drop in IC_50_ vs. PKL. Removing the moiety but keeping the planarity and/or an H-bond acceptor group (**178**–**180**) afforded compounds with a very weak inhibitory activity. Surprisingly, the introduction of a geminal-dimethyl group instead of the carbonyl gave a quite potent activator (EC_50_ = 3 μM).

Subsequently, we decided to explore a molecular simplification of urolithin C to produce a fluorenone nucleus. A few analogues were synthesized with a different substitution pattern (**192**, **194**, **196**–**198, 202**). Moreover, further simplification of the fluorenone resulted in few benzophenone analogues (**209**–**211**). The synthesis of all these analogues is shown in Figure 15; Figure 16. The corresponding methyl ester was reacted with the selected boronic acids to afford the key intermediates **186**–**190**. These intermediates were converted into the corresponding fluorenones by intramolecular Friedel–Crafts acylation catalyzed by triflic acid. Deprotection of the methoxy groups gave **192**, **194**, and **196**, while **197** and **198** did not require any additional steps. Compound **202** was instead obtained from **192** by benzyl protection, reduction, and alkylation of the resulting alcohol. Finally, deprotection/reduction by H_2_/Pd gave the desired fluorene compound.

The benzophenone compounds were easily isolated in two steps by Friedel–Crafts acylation of 1,2-dimethoxybenzene with the corresponding acid and subsequent deprotection. 

Interestingly, the two fluorenone analogues of **1**, compounds **192** and **194**, retained most of the activity of their parent molecule (Table 5). In particular, the position of the phenolic group plays a relevant role since the 6-OH derivative (**194**) is slightly more active than **1** and twice as active as **192**. Adding an additional -OH (**196**) resulted in an 8–10-fold drop in the IC_50_ value. Alkylation of the catechol moiety resulted in completely inactive compounds (**197**, **198**). Similarly, removing the carbonyl group as in **202** resulted in a drop in activity. Finally, all the benzophenone analogues **209**–**211** showed no significant inhibitory effect.

### 2.6. SAR Summary

Three major SAR series were explored by modifying the phenol, catechol, and lactone moiety (Figure 3). At the phenolic position, a carboxylic acid or an amide seems to be tolerated while any other substitution led to a loss of activity.

Modifications of the catechol group with standard and non-standard bioisosteres and other chemical modifications resulted in a loss of activity, suggesting that this group is needed for the activity. The most promising group for further substitution seems to be the lactone. Molecular simplification to obtain a carbonyl resulted in **194**, which was slightly more active than the parent compound. In general, we discovered that these molecules need fine-tuning, and even slight modifications might lead to inactive compounds or—more surprisingly—to an activation of the enzyme.

## 3. Material and Methods

### 3.1. Chemistry General Methods

#### 3.1.1. General Information

All reagents were purchased from Sigma-Aldrich and Fluorochem and were used without further purification unless noted otherwise. Solvents were dried on a solvent purification system (PS-MD-5/7 Inert technology). Microwave reactions were performed in capped vials using a Biotage Initiator Sixty instrument with fixed hold time. The reactions were monitored by LC-MS (Perkin Elmer Series 200; Waters Symmetry C8 column 3.5 µm, 4.6 × 50 mm; water:CH_3_CN (0.1% formic acid)) or by thin-layer chromatography (TLC) on silica-plated aluminum sheets (Silica gel 60 F254, E. Merck, (Rahway, NJ, USA)) and detecting spots by UV light (λ = 254 nm). Flash column chromatography was performed on silica gel 60 (0.040−0.063 mm), manually or using a Biotage SP4 Flash instrument, using silica gel SNAP KP-Sil FSK0-1107 cartridges. NMR spectra were recorded on a Varian NMR 400, Brucker 700 or Oxford 800 (Brucker, (Billerica, MA, USA)) spectrometer at 25 °C unless noted otherwise, using CDCl_3_, CD_3_OD, or DMSO-d_6_ as solvent as indicated. Chemical shifts are reported in ppm with the solvent residual peak as internal standard: ^1^H—residual CHCl_3_ (*δ*_H_ 7.26), CD_3_OD (*δ*_H_ 3.31) or DMSO-d_6_ (*δ*_H_ 2.50) and ^13^C—CDCl_3_ (*δ*_C_ 77.16), CD_3_OD (*δ*_C_ 49.80) or DMSO-d_6_ (*δ*_C_ 39.52). NMR data are reported as follows: chemical shift, number of protons/carbons, multiplicity (s, singlet; d, doublet; t, triplet; q, quartet; m, multiplot; br, broadened), coupling constants (Hz). Melting points were recorded on a Büchi melting point apparatus B-545 or a Mettler FP82 hot stage equipped with a FP80 temperature controller and are uncorrected. High resolution mass spectra (HRMS) were recorded on an Agilent 6520 quadrupole time of flight instrument coupled to an Agilent 1290 infinity ultra-performance liquid chromatography instrument (Santa Clara, CA, USA). Samples were dissolved in acetonitrile and eluted using isocratic elution (100% acetonitrile) with a flow rate of 0.4 mL/min. A mass spectrometer was operated in positive electrospray ionization scanning mode between 50 and 1200 m/z. Ion source parameters were as follows: drying gas flow 10 L/min and temperature 325 °C and nebulizer pressure 35 psig. The mass spectrometer was calibrated before analyses.

#### 3.1.2. General Procedure A

The compound was obtained following the literature procedure [23]. To a suspension of the corresponding 2-bromo benzoic acid (1 mmol) and resorcinol (**4**) (3 mmol) in water (3.0 mL), an aqueous solution of NaOH (4 M aq, 1 mmol) was added and the mixture was stirred at room temperature until it became a clear solution. Then, Na_2_CO_3_ (2.2 mmol) was added, and the mixture was stirred at 50 °C for 10 min. Subsequently, CuI (0.3 mmol) was added, and the reaction mixture was stirred at 50 °C for 24 h (until full conversion, reaction monitored by LC-MS). The resulting suspension was filtered and the solid obtained dried. Crystallization or purification by flash column chromatography afforded the desired compound.

#### 3.1.3. General Procedure B

To a solution of the corresponding methyl ether (1 mmol) in CH_2_Cl_2_ (10.0 mL), BBr_3_ (1.0 M in CH_2_Cl_2_, 3 mmol per methyl ether in the starting material) was added dropwise at 0 °C. The solution was stirred at room temperature for 12–72 h (until full conversion, reaction monitored by LC-MS). Water (20.0 mL) was added to quench the reaction and the aqueous layer was extracted with EtOAc (3 × 20 mL). The combined organic layers were dried over Na_2_SO_4_, filtered, and concentrated under reduced pressure. Crystallization, trituration, or purification by flash column chromatography afforded the desired compound.

#### 3.1.4. General Procedure C

To a solution of the selected benzoic acid (1 mmol) in MeOH (3 mL) a few drops of concentrated H_2_SO_4_ (cat.) were added. The resulting mixture was refluxed for 12 h. The solvent was then removed under reduced pressure and the desired compound obtained by crystallization or by flash column chromatography.

#### 3.1.5. General Procedure D

The compound was obtained following the literature procedure [24]. The selected halide (1 mmol), boronic acid (1.1 mmol), Pd(PPh_3_)_4_ (0.1 mmol) and K_2_CO_3_ (2 mmol) or K_3_PO_4_ (4 mmol) were placed in a microwave vial. A degassed solution (nitrogen sparging) of 1,4-dioxane:water (1:1, 10.0 mL) was added. The mixture was stirred at 110 °C in the microwave for 2 h. The reaction mixture was then rinsed with water (10.0 mL) and extracted with EtOAc (3 × 10.0 mL). The combined organic layers were dried over Na_2_SO_4_, filtered, and concentrated under reduced pressure. Purification by flash column chromatography afforded the desired compound.

#### 3.1.6. General Procedure E

To a solution of methyl ester (1 mmol) in 1,4-dioxane (10.0 mL), LiOH (2 M aq, 2 mmol) was added. The reaction mixture was stirred at reflux for 12 h (or until full conversion, reaction monitored by TLC). Then, the reaction mixture was cooled down to room temperature and dioxane was evaporated under reduced pressure. The residue was diluted with water (20.0 mL) and acidified to pH 1–2 by addition of an aqueous solution of HCl (1 M). The aqueous solution was extracted with EtOAc (3 × 40.0 mL); the combined organic layers were dried over Na_2_SO_4_, filtered, and concentrated under reduced pressure to afford the corresponding carboxylic acid.

#### 3.1.7. General Procedure F

The compound was obtained following the literature procedure [17]. To a solution of the corresponding 2-aryl benzoic acid (1 mmol) in water:CH_3_CN (1:1, 10.0 mL), K_2_S_2_O_8_ (2 mmol) and AgNO_3_ (0.01 mmol) were added. The reaction mixture was stirred at 50 °C for 12 h (until full conversion, reaction monitored by LC-MS). Then, CH_3_CN was removed under reduced pressure. The remaining aqueous solution was basified to pH 9 with a saturated aqueous solution of NaHCO_3_ and extracted with EtOAc (3 × 10.0 mL), and the combined organic layers were dried over Na_2_SO_4_, filtered, and concentrated under reduced pressure. Purification by flash column chromatography afforded the desired compound.

#### 3.1.8. General Procedure G

This procedure was previously reported for the synthesis of dibenzopyranones [18]. To a suspension of the selected acid (1.00 mmol) in 1,2-dichloroethane (15 mL), *N*-iodosuccinimide (0.9 g, 4.0 mmol) was added. The reaction mixture was stirred at 80 °C for 20 h. The reaction mixture was cooled down to room temperature, EtOAc (70 mL) was added, the suspension was washed with a saturated aqueous solution of Na_2_S_2_O_3_ (3 × 30 mL) and water (2 × 30 mL), and then the organic layer was evaporated under reduced pressure. Crystallization, trituration, or purification by flash column chromatography afforded the desired compound.

#### 3.1.9. General Procedure H

To a solution of the selected benzylated derivative (1 mmol) in DMF/MeOH (2:1, 10.0 mL), 10% Pd/C was added (10% *w*/*w* for each group to be reduced/deprotected) and the reaction stirred at room temperature for 24 h (until full conversion, reaction monitored by LC-MS). The catalyst was then filtered on a Celite pad, and the solvent removed under reduced pressure to afford the desired compound.

#### 3.1.10. General Procedure I

To a suspension of the selected aniline (1 mmol) in CH_3_CN (15 mL), DIPEA (9 mL) and appropriate acyl chloride (4.5 mmol) were added. The reaction mixture was stirred at room temperature. After complete conversion of the starting material, the reaction was quenched with MeOH, and the solvent removed under reduced pressure. The obtained crude was suspended in CH_2_Cl_2_ and filtered. The resulting solid was washed with a mixture of CH_2_Cl_2_/MeOH to yield the acylated compound.

#### 3.1.11. General Procedure J

To a suspension of the selected aldehyde (0.75 mmol) in *t-*BuOH (12 mL) and CH_3_CN (2.5 mL), 2-methyl-2-butene (0.6 mL, 5.66 mmol), and a solution of NaClO_2_ (80%, 170 mg, 1.5 mmol) and KH_2_PO_4_ (204 mg, 1.5 mmol) in water (6 mL) were added dropwise (exothermic). The reaction mixture was stirred at room temperature for 24 h. A second portion of NaClO_2_ (85 mg) and KH_2_PO_4_ (102 mg) in water (3 mL) was added and the reaction mixture was stirred for additional 24 h. Then, an aqueous solution of HCl (1 M, 2 mL), water (30 mL), and EtOAc (40 mL) were added. The layers were separated, and the aqueous layer was extracted with EtOAc (3 × 20 mL). The combined organic layers were dried over Na_2_SO_4_, filtered, and concentrated under reduced pressure. The crude was triturated with pentane and Et_2_O to afford the desired product.

#### 3.1.12. General Procedure K

To a solution of the selected phenol (1 mmol) and K_2_CO_3_ (2.5 mmol per phenolic group) in acetone (30 mL), benzyl bromide was added (2.5 mmol per phenolic group). The mixture was stirred at room temperature for 12–24 h (until full conversion, reaction monitored by TLC). The reaction was then filtered, and the solvent concentrated under reduced pressure. Purification by flash chromatography afforded the desired compound.


**Synthesis of 3,9-dihydroxy-6H-benzo[c]chromen-6-one (6)**



*3-Hydroxy-9-methoxy-6H-benzo[c]chromen-6-one (*
**5**
*)*


Following general procedure A, 2-bromo-4-methoxybenzoic acid (**3**) (0.92 g, 4.00 mmol) was reacted and recrystallized from MeOH to afford 0.71 g (73%) of **5** as a pink solid. Mp = 272–274 °C; ^1^H NMR (400 MHz, DMSO-d_6_) δ 10.32 (s, 1H), 8.43–7.90 (m, 2H), 7.64 (s, 1H), 7.26–6.21 (m, 3H), 3.95 (s, 3H); ^13^C NMR (101 MHz, DMSO-d_6_) δ 164.81, 160.81, 132.04, 116.45, 112.67, 110.20, 105.17, 56.37.


*3,9-dihydroxy-6H-benzo[c]chromen-6-one (*
**6**
*)*


Following general procedure B, **5** (0.200 g, 0.82 mmol) was reacted and triturated with Et_2_O to afford 0.18 g (95%) of **6** as a grey solid. Mp > 350 °C. ^1^H NMR (400 MHz, DMSO-d_6_) δ 10.80 (s, 1H), 10.27 (s, 1H), 8.02 (d, *J* = 8.7 Hz, 1H), 7.95 (d, *J* = 8.8 Hz, 1H), 7.42 (d, *J* = 2.3 Hz, 1H), 6.95 (dd, *J* = 8.7, 2.2 Hz, 1H), 6.79 (dd, *J* = 8.7, 2.4 Hz, 1H), 6.69 (d, *J* = 2.3 Hz, 1H); ^13^C NMR (101 MHz, DMSO-d_6_) δ 164.17, 160.78, 160.25, 152.86, 137.83, 132.89, 125.13, 116.93, 113.43, 111.07, 109.81, 106.60, 103.28; HRMS (ESI) m/z: 227.0356 [M − H]^−^, calculated for C_12_H_8_O_4_ 227.0350.


**Synthesis of 8,9-dihydroxy-6H-benzo[*c*]chromen-6-one (13)**



*2-Bromo-4,5-dimethoxybenzoic acid (*
**8**
*)*


To a suspension of 3,4-dimethoxybenzoic acid (**7**) (5.00 g, 27.44 mmol) in HCl conc. (100.0 mL), bromine (4.82 g, 30.19 mmol) was added dropwise at room temperature. The reaction mixture was stirred for 7 h at room temperature. Water (100.0 mL) was then added, and the resulting precipitate was collected by filtration and recrystallized from MeOH to afford 6.1 g (85%) of **8** as a white solid. Mp = 185–187 °C; ^1^H NMR (400 MHz, DMSO-d_6_) δ 7.35 (s, 1H), 7.19 (s, 1H), 3.81 (s, 3H), 3.76 (s, 3H); ^13^C NMR (101 MHz, DMSO-d_6_) δ 166.91, 151.93, 148.01, 124.37, 117.18, 114.22, 112.83, 56.52, 56.14.


*Methyl 2-bromo-4,5-dimethoxybenzoate (*
**9**
*)*


Following general procedure C, **8** (5.00 g, 19.15 mmol) was reacted and crystallized from MeOH to afford 4.43 g (84%) of **9** as a white solid. Mp = 88–90 °C; ^1^H NMR (400 MHz, CDCl_3_) δ 7.29 (s, 1H), 6.99 (s, 1H), 3.91–3.71 (m, 9H); ^13^C NMR (101 MHz, CDCl_3_) δ 165.85, 151.92, 147.64, 122.62, 116.82, 114.05, 113.84, 56.15, 56.00, 52.17, 50.13.


*Methyl 4,5-dimethoxy-[1,1′-biphenyl]-2-carboxylate (*
**10**
*)*


Following general procedure D, **9** (0.50 g, 1.81 mmol), phenylboronic acid (0.24 g, 1.99 mmol), were reacted. The crude was purified with flash chromatography (pentane/EtOAc) to afford 0.27 g (55%) of **10** as a colorless oil. ^1^H-NMR (400 MHz, CDCl_3_): *δ* 7.43 (s, 1H), 7.25–7.45 (m, 5H), 6.80 (s, 1H), 3.94 (s, 3H), 3.90 (s, 3H), 3.60 (s, 3H); ^13^C NMR (101 MHz, CDCl_3_) δ 167.88, 150.92, 147.46, 141.43, 137.12, 128.20, 127.59, 126.71, 121.67, 113.31, 112.60, 55.84, 55.78, 51.45.


*4,5-Dimethoxy-[1,1′-biphenyl]-2-carboxylic acid (*
**11**
*)*


Following general procedure E, **10** (0.27 g, 0.99 mmol) was reacted to afford 0.24 g (94%) of **11** as a white solid. Mp = 198–200 °C; ^1^H-NMR (CDCl_3_, 400 MHz): *δ* 7.54 (s, 1H), 7.20–7.44 (m, 5H), 6.78 (s, 1H), 3.95 (s, 3H), 3.91 (s, 3H); ^13^C NMR (101 MHz, CDCl_3_) δ 172.75, 151.86, 147.63, 141.43, 138.67, 128.63, 127.87, 127.08, 120.35, 113.99, 113.56, 56.11, 56.08.


*8,9-Dimethoxy-6H-benzo[c]chromen-6-one (*
**12**
*)*


Following general procedure F, **11** (0.24 g, 0.92 mmol) was reacted and purified by flash chromatography (pentane/EtOAc) to afford 0.19 g (90%) of **12** as a white powder. Mp = 203–205 °C; ^1^H-NMR (400 MHz, CDCl_3_,): *δ* 7.93 (dd, *J* = 1.5, 7.9 Hz, 1H), 7.73 (s, 1H), 7.41–7.47 (m, 2H), 7.28–7.39 (m, 2H), 4.08 (s, 3H), 3.99 (s, 3H); ^13^C NMR (101 MHz, CDCl_3_) δ 161.03, 155.05, 150.96, 150.08, 129.86, 129.53, 124.31, 122.10, 118.02, 117.68, 114.46, 110.44, 102.61, 56.29, 56.28.


*8,9-Dihydroxy-6H-benzo[c]chromen-6-one (*
**13**
*)*


Following general procedure B, **12** (0.17 g, 0.66 mmol) was reacted and triturated with Et_2_O to afford 0.110 g (73%) of **13** as a white solid. Mp = 288–290 °C; ^1^H-NMR (400 MHz, DMSO-d*_6_*): δ 10.45 (br s, 1H), 10.22 (br s, 1H), 8.04 (d, *J* = 8.5 Hz, 1H), 7.60 (s, 1H), 7.55 (s, 1H), 7.44 (ddd, *J* = 1.5, 7.1, 8.5 Hz, 1H), 7.29–7.38 (m, 2H); ^13^C NMR (101 MHz, DMSO-d_6_) δ 160.37, 153.77, 150.61, 147.80, 129.60, 128.58, 124.95, 123.10, 118.46, 117.44, 114.77, 112.98, 108.28, 40.58, 40.22; HRMS (ESI) m/z: 227.036 [M − H]^−^, calculated for C_12_H_8_O_4_ 227.0350.


**[1,1′-Biphenyl]-3,4,4′-triol (17)**



*3,4,4′-Trimethoxy-1,1′-biphenyl (*
**16**
*)*


Following general procedure D, 3,4-dimethoxybenzeneboronic acid (**14**) (0.53 g, 2.94 mmol) and 4-bromoanisole (**15**) (0.50 g, 2.67 mmol) were reacted and purified by flash chromatography (pentane/EtOAc) to afford 0.57 g (87%) of **16** as a white solid. Mp = 104–106 °C; ^1^H NMR (400 MHz, CDCl_3_) δ 7.53–7.43 (m, 2H), 7.13–7.04 (m, 2H), 7.01–6.89 (m, 3H), 3.95 (s, 3H), 3.92 (s, 3H), 3.85 (s, 3H); ^13^C NMR (101 MHz, CDCl_3_) δ 158.80, 149.09, 148.14, 133.96, 133.66, 127.84, 118.91, 114.15, 111.49, 110.17, 55.98, 55.91, 55.35.


*[1,1′-Biphenyl]-3,4,4′-triol (*
**17**
*)*


Following general procedure B, **16** (0.300 g, 1.23 mmol) was reacted and triturated with Et_2_O to afford 0.19 g (78%) of **17** as a white solid. Mp = 246–248 °C; ^1^H NMR (400 MHz, CD_3_OD) δ 7.40–7.24 (m, 2H), 6.97 (d, *J* = 2.2 Hz, 1H), 6.86 (dd, *J* = 8.2, 2.2 Hz, 1H), 6.82–6.72 (m, 3H); ^13^C NMR (101 MHz, CD_3_OD) δ 155.90, 145.00, 143.85, 133.26, 132.70, 127.05, 117.51, 115.22, 114.98, 113.19; HRMS (ESI) m/z: 201.0564 [M − H]^−^, calculated for C_12_H_9_O_3_ 201.0557.


**8,9-dihydroxy-3-(methylamino)-6H-benzo[c]chromen-6-one (26)**



*Methyl 4′-((tert-butoxycarbonyl)(methyl)amino)-4,5-dimethoxy-[1,1′-biphenyl]-2-carboxylate (*
**20**
*)*


Following general procedure D, **9** (250 mg, 0.9 mmol) and *tert*-butyl-N-methyl-N-[4-(4,4,5,5-tetramethyl-1,3,2-dioxaborolan-2-yl)phenyl]carbamate (**18**) (0.25 mg, 1.0 mmol) were reacted and purified by flash chromatography (pentane/EtOAc) to afford **20** as a yellow oil (244 mg, 0.6 mmol, 67%). ^1^H NMR (400 MHz, DMSO-d_6_) δ 7.34–7.18 (m, 5H), 6.91 (s, 1H), 3.85 (s, 3H), 3.83 (s, 3H), 3.55 (s, 3H), 3.21 (s, 3H), 1.41 (s, 9H); ^13^C NMR (101 MHz, DMSO-d_6_) δ 167.87, 153.70, 150.91, 147.48, 142.30, 137.50, 135.08, 128.43, 124.67, 121.89, 113.63, 112.67, 79.61, 55.75, 55.72, 51.63, 51.61, 36.98, 27.95.


*4′-((tert-butoxycarbonyl)(methyl)amino)-4,5-dimethoxy-[1,1′-biphenyl]-2-carboxylic acid (*
**22**
*)*


Following general procedure E (reaction conducted at 65 °C), **20** (180 mg, 0.5 mmol) was reacted to afford 0.17 g (99%) of **22** as a white solid. The compound was used in the next step without any purification. ^1^H NMR (400 MHz, DMSO-d_6_) δ 7.42 (d, *J* = 8.6 Hz, 2H), 7.16 (d, *J* = 8.6 Hz, 2H), 6.99 (s, 1H), 6.72 (s, 1H), 3.75 (d, *J* = 4.2 Hz, 6H), 3.19 (s, 3H), 1.41 (s, 9H).


*Tert-butyl (8,9-dimethoxy-6-oxo-6H-benzo[c]chromen-3-yl)(methyl)carbamate (*
**24**
*)*


Following general procedure F, **22** (0.17 g, 0.49 mmol) was reacted and purified by flash chromatography (pentane/EtOAc) to afford **24** as an orange solid 0.09 mg (55%). ^1^H NMR (400 MHz, DMSO-d_6_) δ 8.35 (d, *J* = 9.4 Hz, 1H), 7.81 (s, 1H), 7.60 (s, 1H), 7.38–7.33 (m, 2H), 4.04 (s, 3H), 3.91 (s, 3H), 3.27 (s, 3H), 1.43 (s, 9H); ^13^C NMR (101 MHz, DMSO-d_6_) δ 160.00, 155.25, 153.33, 150.26, 149.72, 144.50, 129.41, 123.33, 121.20, 114.54, 112.90, 112.66, 109.82, 104.16, 80.29, 56.45, 55.77, 36.72, 27.91.


*8,9-dihydroxy-3-(methylamino)-6H-benzo[c]chromen-6-one (*
**26**
*)*


Following general procedure B, **24** (0.04 g, 0.1 mmol) was reacted and triturated with Et_2_O to afford 0.02 g (82%) of **25** as a brown solid. Mp = 279–281 °C; ^1^H NMR (400 MHz, CD_3_OD) δ 8.14 (d, *J* = 8.8 Hz, 1H), 7.64 (s, 1H), 7.56 (s, 1H), 7.35–7.20 (m, 2H), 3.09 (s, 3H); ^13^C NMR (101 MHz, CD_3_OD) δ 162.04, 155.04, 152.47, 149.18, 139.42, 129.24, 125.76, 125.74, 120.13, 118.43, 115.58, 114.26, 111.23, 108.70, 37.11; HRMS (ESI): m/z: 258,0773 [M + H]^+^; calculated C_14_H_12_NO_4_: 258,0761. 


**3-Amino-8,9-dihydroxy-6H-benzo[c]chromen-6-one (28)**



*Methyl 4,5-dimethoxy-4′-nitro-[1,1′-biphenyl]-2-carboxylate (*
**21**
*)*


Following general procedure D, **9** (3.080 g, 11.2 mmol) and 4-nitrobenzeneboronic acid (**19**) (2.056 g, 12.3 mmol), were reacted and purified by flash chromatography (pentane/EtOAc) to afford 1.92 g (54%) of **21** as a pale-orange solid. Mp = 134–136 °C; ^1^H-NMR (400 MHz, CDCl_3_) δ 8.26–8.23 (m, 2H), 7.52 (s, 1H), 7.45–7.41 (m, 2H), 6.73 (s, 1H), 3.98 (s, 3H), 3.93 (s, 3H), 3.65 (s, 3H); ^13^C NMR (101 MHz, CDCl_3_) δ 167.22, 151.73, 148.94, 148.68, 147.02, 135.49, 129.64, 123.25, 121.59, 113.34, 113.24, 56.36, 56.33, 52.10.


*4,5-Dimethoxy-4′-nitro-[1,1′-biphenyl]-2-carboxylic acid (*
**23**
*)*


Following general procedure E, **21** (1.910 g, 6.02 mmol) was reacted to afford 1.79 g (98%) of **23** as a pale-brown solid. Mp = 266–268 °C; ^1^H-NMR (400 MHz, DMSO-d_6_) δ 12.66 (br s, 1H), 8.24–8.21 (m, 2H), 7.59–7.56 (m, 2H), 7.44 (s, 1H), 6.92 (s, 1H), 3.86 (s, 3H), 3.85 (s, 3H); ^13^C NMR (101 MHz, DMSO-d_6_) δ 167.87, 150.89, 148.62, 148.11, 146.27, 133.98, 130.05, 122.85, 122.61, 113.71, 113.15, 55.84, 55.72.


*8,9-Dimethoxy-3-nitro-6H-benzo[c]chromen-6-one (*
**25**
*)*


Following general procedure G, **23** (1.090 g, 3.594 mmol) was reacted and triturated from CH_2_Cl_2_ to afford 0.9 g (84%) of **25** as a yellow solid. Mp = 322–324 °C; ^1^H-NMR (400 MHz, DMSO-d_6_) δ 8.73 (d, *J* = 8.7 Hz, 1H), 8.24–8.20 (m, 2H), 7.99 (s, 1H), 7.68 (s, 1H), 4.07 (s, 3H), 3.96 (s, 3H); ^13^C-NMR (201 MHz, DMSO-d_6,_ 60 °C because of low solubility in DMSO) δ 158.88, 155.24, 151.31, 149.83, 147.06, 127.40, 124.71, 123.77, 118.56, 114.36, 112.21, 110.19, 105.50, 56.46, 55.88.


*3-Amino-8,9-dimethoxy-6H-benzo[c]chromen-6-one (*
**27**
*)*


Following general procedure H (conducted at 50 °C), **25** (1.410 g, 4.68 mmol) was reacted to afford to afford 1.16 g (92%) of **27** as a beige solid. Mp = 318–320 °C; ^1^H-NMR (400 MHz, DMSO-d_6_) δ 7.99 (d, *J* = 8.6 Hz, 1H), 7.57 (s, 1H), 7.49 (s, 1H), 6.61 (dd, *J* = 8.6, 2.2 Hz, 1H), 6.46 (d, *J* = 2.2 Hz, 1H), 5.79–5.77 (m, 2H), 3.99 (s, 3H), 3.85 (s, 3H); ^13^C-NMR (101 MHz, DMSO-d_6_) δ 160.62, 155.28, 152.20, 150.98, 148.07, 131.51, 124.28, 111.38, 110.66, 109.65, 106.18, 102.60, 99.59, 56.20, 55.60.


*3-Amino-8,9-dihydroxy-6H-benzo[c]chromen-6-one (*
**28**
*)*


Following general procedure B, **27** (0.040 g, 0.15 mmol) was reacted to afford 0.030 g (84%) of **28** as a white solid. Mp > 350 °C; ^1^H-NMR (400 MHz, CD_3_OD): *δ* 8.20 (d, *J* = 9.2 Hz, 1H, Ar-H), 7.64 (s, 1H, Ar-H), 7.59 (s, 1H, Ar-H), 7.33–7.39 (m, 2H, Ar-H); ^13^C NMR (101 MHz, CD_3_OD) δ 160.55, 153.65, 150.90, 147.86, 130.89, 127.72, 124.25, 119.16, 118.67, 114.16, 112.93, 111.74, 107.37; HRMS (ESI) m/z: 242.0462 [M − H]^−^, calculated for C_13_H_8_NO_4_ 242.0459.


**N-(8,9-dihydroxy-6-oxo-6H-benzo[c]chromen-3-yl)acetamide (33)**



*N-(8,9-dimethoxy-6-oxo-6H-benzo[c]chromen-3-yl)acetamide (*
**29**
*)*


Following general procedure I, **27** (70.0 mg, 0.258 mmol) and acetyl chloride (83 µL, 1.162 mmol) were reacted to afford 0.06 g (71%) of **29** as a beige solid. Mp = 333–335 °C; ^1^H NMR (400 MHz, DMSO-d_6_) δ 10.29 (s, 1H), 8.30 (d, *J* = 8.8 Hz, 1H), 7.76 (d, *J* = 2.1 Hz, 1H), 7.73 (s, 1H), 7.56 (s, 1H), 7.47 (dd, *J* = 8.7, 2.1 Hz, 1H), 4.02 (s, 3H), 3.89 (s, 3H), 2.09 (s, 3H); ^13^C NMR (101 MHz, dm DMSO-d_6_) δ 168.81, 160.12, 155.26, 150.73, 149.39, 140.64, 129.84, 123.95, 115.13, 112.82, 112.43, 109.77, 106.23, 103.73, 56.38, 55.75, 39.99, 24.15.


*N-(8,9-dihydroxy-6-oxo-6H-benzo[c]chromen-3-yl)acetamide (*
**33**
*)*


Following general procedure B, **29** (0.033 g, 0.11 mmol) was reacted to afford 0.023 g (77%) of **33** as a beige solid. Mp > 350 °C; ^1^H NMR (400 MHz, DMSO-d_6_) δ 10.43 (s, 1H), 10.25 (s, 1H), 10.14 (s, 1H), 7.96 (d, *J* = 8.8 Hz, 1H), 7.71 (d, *J* = 2.1 Hz, 1H), 7.52 (s, 1H), 7.50 (s, 1H), 7.45 (dd, *J* = 8.7, 2.1 Hz, 1H), 2.08 (s, 3H); ^13^C NMR (101 MHz, DMSO-d_6_) δ 168.76, 160.14, 153.43, 150.45, 146.77, 140.12, 128.50, 123.08, 115.31, 114.28, 113.03, 111.68, 107.30, 106.32, 24.15; HRMS (ESI): m/z: 286,0712 [M + H]^+^; calculated for C_15_H_12_NO_5_: 286,071. 


**N-(8,9-dihydroxy-6-oxo-6H-benzo[c]chromen-3-yl)propionamide (34)**



*N-(8,9-dimethoxy-6-oxo-6H-benzo[c]chromen-3-yl)propionamide (*
**30**
*)*


Following general procedure I, **27** (0.05 g, 0.184 mmol) and propionyl chloride (48 µL, 0.553 mmol) were reacted to afford 0.04 g (61%) of **30** as a beige solid. Mp = 283–285 °C; ^1^H NMR (400 MHz, DMSO-d_6_) δ 10.21 (s, 1H), 8.37–8.21 (m, 1H), 7.77 (d, *J* = 2.3 Hz, 1H), 7.75–7.70 (m, 1H), 7.59–7.54 (m, 1H), 7.54–7.48 (m, 1H), 4.05–3.99 (m, 3H), 3.89 (t, *J* = 1.4 Hz, 3H), 2.42–2.33 (m, 2H), 1.16–1.05 (m, 3H); ^13^C NMR (101 MHz, DMSO-d_6_) δ 172.47, 160.14, 155.26, 150.74, 149.37, 140.72, 129.86, 123.94, 115.17, 112.73, 112.40, 109.77, 106.22, 103.72, 56.37, 55.74, 29.61, 9.46.


*N-(8,9-dihydroxy-6-oxo-6H-benzo[c]chromen-3-yl)propionamide (*
**34**
*)*


Following general procedure B, **30** (0.022 g, 0.07 mmol) was reacted to afford 0.016 g (78%) of **34** as a beige solid. Mp > 350 °C; ^1^H NMR (400 MHz, DMSO-d_6_) δ 10.71–9.79 (m, 3H), 7.96 (d, *J* = 8.8 Hz, 1H), 7.73 (d, *J* = 2.1 Hz, 1H), 7.52 (s, 1H), 7.50 (s, 1H), 7.47 (dd, *J* = 8.7, 2.1 Hz, 1H), 2.37 (q, *J* = 7.5 Hz, 2H), 1.10 (t, *J* = 7.5 Hz, 3H); ^13^C NMR (101 MHz, DMSO-d_6_) δ 172.41, 160.14, 153.43, 150.46, 146.74, 140.18, 128.51, 123.05, 115.32, 114.26, 112.93, 111.64, 107.28, 106.31, 29.61, 9.52; HRMS (ESI): m/z: 300,0872 [M + H]^+^; calculated for C_16_H_14_NO_5_: 300,0874.


**N-(8,9-dihydroxy-6-oxo-6H-benzo[c]chromen-3-yl)*butyramide* (35)**



*N-(8,9-dimethoxy-6-oxo-6H-benzo[c]chromen-3-yl)butyramide (*
**31**
*)*


Following general procedure I, **27** (0.07 g, 0.258 mmol) and butyryl chloride (40 μL, 0.387 mmol) were reacted to afford 0.054 g (61%) of **31** as a beige solid. Mp = 287–288 °C; ^1^H-NMR (400 MHz, DMSO-d_6_) δ 10.20 (s, 1H), 8.28 (d, *J* = 8.8 Hz, 1H), 7.76 (d, *J* = 2.1 Hz, 1H), 7.72 (s, 1H), 7.55 (s, 1H), 7.49 (dd, *J* = 8.8, 2.1 Hz, 1H), 4.00 (s, 3H), 3.88 (s, 3H), 2.32 (t, *J* = 7.4 Hz, 2H), 1.67–1.57 (m, 2H), 0.91 (t, *J* = 7.4 Hz, 3H); ^13^C-NMR (101 MHz, DMSO-d_6_) δ 171.64, 160.14, 155.26, 150.74, 149.38, 140.66, 129.86, 123.94, 115.20, 112.78, 112.42, 109.77, 106.26, 103.73, 56.38, 55.75, 38.41, 18.41, 13.65.


*N-(8,9-dihydroxy-6-oxo-6H-benzo[c]chromen-3-yl)butyramide (*
**35**
*)*


Following general procedure B, **31** (0.04 g, 0.117 mmol) was reacted to afford 0.031 g (84%) of **35** a pale-pink-brown solid. Mp > 350 °C; ^1^H-NMR (400 MHz, DMSO-d_6_) δ 10.42 (br s, 1H), 10.18 (s, 1H), 10.13 (br s, 1H), 7.96 (d, *J* = 8.8 Hz, 1H), 7.73 (d, *J* = 2.1 Hz, 1H), 7.52 (s, 1H), 7.50 (s, 1H), 7.47 (dd, *J* = 8.8, 2.1 Hz, 1H), 2.33 (t, *J* = 7.3 Hz, 2H), 1.68–1.58 (m, 2H), 0.93 (t, *J* = 7.3 Hz, 3H); ^13^C-NMR (101 MHz, DMSO-d_6_) δ 171.57, 160.13, 153.41, 150.44, 146.74, 140.12, 128.50, 123.04, 115.35, 114.26, 112.97, 111.65, 107.29, 106.34, 38.41, 18.45, 13.64; HRMS (ESI) calculated for C_17_H_16_NO_5_ [M + H]^+^ 314.1023, found 314.1031. 


***N*-(8,9-Dihydroxy-6-oxo-6*H*-benzo[*c*]chromen-3-yl)methanesulfonamide (36)**



*N-(8,9-Dimethoxy-6-oxo-6H-benzo[c]chromen-3-yl)methanesulfonamide (*
**32**
*)*


To a suspension of **27** (0.0045 g, 0.166 mmol) in pyridine (1.6 mL) at 0 °C, mesyl chloride (15 μL, 0.18 mmol) was added. The reaction mixture was stirred from 0 °C to room temperature for 15 h. Then, a second portion of mesyl chloride (15 μL, 0.18 mmol) was added and the reaction mixture was stirred for 2 h. Then, the reaction mixture was diluted with EtOAc (50 mL), washed with a saturated aqueous solution of Cu_2_SO_4_ (1 × 10 mL), water (1 × 10 mL), an aqueous solution of HCl (1 M, 2 × 10 mL), and water (3 × 20 mL). The organic layer was concentrated under reduced pressure and the residue was dried by co-evaporation with toluene (×3) to afford 0.042 g (73%) of **32** as a pale-brown solid. Mp = 304–306 °C; ^1^H-NMR (400 MHz, DMSO-d_6_) δ 10.21 (s, 1H), 8.33 (d, *J* = 8.4 Hz, 1H), 7.73 (s, 1H), 7.56 (s, 1H), 7.21–7.18 (m, 2H), 4.02 (s, 3H), 3.89 (s, 3H), 3.12 (s, 3H); ^13^C-NMR (101 MHz, DMSO-d_6_) δ 159.92, 155.27, 150.99, 149.51, 139.86, 129.61, 124.65, 115.11, 113.31, 112.47, 109.78, 106.24, 103.82, 56.42, 55.75.


*N-(8,9-Dihydroxy-6-oxo-6H-benzo[c]chromen-3-yl)methanesulfonamide *
*(*
**36**
*)*


Following general procedure B, **32** (0.033 g, 0.095 mmol) was reacted and triturated with Et_2_O and CH_2_Cl_2_ to afford 0.014 g (46%) of **35** a pale-brown solid. Mp = 323–325 °C; ^1^H-NMR (400 MHz, DMSO-d_6_) δ 10.44 (br s, 1H), 10.17 (br s, 1H), 10.15 (br s, 1H), 8.01 (d, *J* = 8.6 Hz, 1H), 7.53 (app s, 2H), 7.18–7.14 (m, 2H), 3.09 (s, 3H); ^13^C-NMR (101 MHz, DMSO-d_6_) δ 159.9, 153.5, 150.7, 146.9, 139.3, 128.3, 123.8, 115.3, 114.3, 113.6, 111.7, 107.4, 106.4, 39.6; HRMS (ESI) calculated for C_14_H_12_NO_6_S [M + H]^+^ 322.0380, found 322.0374.


**3-Fluoro-8,9-dihydroxy-6H-benzo[c]chromen-6-one (61)**



*Methyl 4′-fluoro-4,5-dimethoxy-[1,1′-biphenyl]-2-carboxylate (*
**43**
*)*


Following general procedure D, **9** (0.50 g, 1.81 mmol) and 4-fluorobenzeneboronic acid (**37**) (0.51 g, 2.00 mmol) were reacted. The compound was extracted and used directly for the next step.


*4′-Fluoro-4,5-dimethoxy-[1,1′-biphenyl]-2-carboxylic acid (*
**49**
*)*


Following general procedure E, crude **43** was reacted to afford 0.25 (48% over 2 steps) of **49** as a white solid. Mp = 212–214 °C; ^1^H NMR (400 MHz, CD_3_OD) δ 7.48 (s, 1H), 7.32–7.26 (m, 2H), 7.11–7.04 (m, 2H), 6.84 (s, 1H), 3.90 (s, 3H), 3.88 (s, 3H); ^13^C NMR (101 MHz, CD_3_OD) δ 163.55 (d, *J* = 243.9 Hz), 139.27 (d, *J* = 3.4 Hz), 131.53 (d, *J* = 8.1 Hz), 115.47 (d, *J* = 21.7 Hz); ^19^F NMR (659 MHz, DMSO-d_6_) δ -116.48.


*3-Fluoro-8,9-dimethoxy-6H-benzo[c]chromen-6-one (*
**55**
*)*


Following general procedure F, **49** (0.2 g, 0.72 mmol) was reacted and purified by flash chromatography (pentane/EtOAc) to afford 0.1 g (50%) of **55** as a brown solid. Mp = 256–258 °C; ^1^H NMR (400 MHz, CDCl_3_) δ 7.93 (dd, *J* = 8.6, 5.8 Hz, 1H), 7.74 (s, 1H), 7.37 (s, 1H), 7.13–7.02 (m, 2H), 4.10 (s, 3H), 4.01 (s, 3H); ^13^C NMR (101 MHz, cdcl_3_) δ 163.08 (d, *J* = 250.4 Hz), 151.98 (d, *J* = 12.3 Hz), 123.76 (d, *J* = 9.7 Hz), 114.83 (d, *J* = 3.1 Hz), 113.80 (d, *J* = 1.3 Hz), 112.35 (d, *J* = 22.5 Hz), 105.21 (d, *J* = 25.2 Hz); ^19^F NMR (659 MHz, DMSO) δ -116.48.


*3-Fluoro-8,9-dihydroxy-6H-benzo[c]chromen-6-one (*
**61**
*)*


Following general procedure B, **55** (0.05 g, 0.36 mmol) was reacted and triturated with Et_2_O to afford 0.26 g (85%) of **61** as a white solid. Mp = 312–314 °C; ^1^H NMR (700 MHz, DMSO-d_6_) δ 10.51 (s, 1H), 10.26 (s, 1H), 8.13 (dd, *J* = 9.5, 6.3 Hz, 1H), 7.59 (s, 1H), 7.55 (s, 1H), 7.34 (dd, *J* = 9.7, 3.1 Hz, 1H), 7.23 (tt, *J* = 9.6, 4.8 Hz, 1H); ^13^C NMR (151 MHz, DMSO-d_6_) δ 162.37 (d, J = 246.7 Hz), 160.22, 154.02, 151.45 (d, J = 12.6 Hz), 147.70, 129.07–123.24 (m), 115.43, 114.76, 112.49 (d, J = 22.0 Hz), 112.19, 108.36, 104.81 (d, J = 25.5 Hz); ^19^F NMR (564 MHz, DMSO) δ -111.40; HRMS (ESI) calculated for C_13_H_8_FO_4_ [M + H]^+^ 247.0407, found 247.0409. 


**8,9-Dihydroxy-3-(trifluoromethyl)-6*H*-benzo[*c*]chromen-6-one (62)**



*Methyl 4,5-dimethoxy-4′-(trifluoromethyl)-[1,1′-biphenyl]-2-carboxylate (*
**44**
*)*


Following general procedure D, **9** (0.50 g, 1.81 mmol) and 4-(trifluoromethyl)benzeneboronic acid (**38**) (0.38 g, 2.00 mmol) were reacted and purified by flash chromatography (pentane/EtOAc) to afford 0.55 (89%) of **44** as a beige solid. Mp = 94–97 °C; ^1^H-NMR (700 MHz, CDCl_3_) δ 7.64 (d, *J* = 8.0 Hz, 2H), 7.50 (s, 1H), 7.40 (d, *J* = 8.0 Hz, 2H), 6.75 (s, 1H), 3.98 (s, 3H), 3.93 (s, 3H), 3.64 (s, 3H); ^13^C-NMR (101 MHz, CDCl_3_) δ 167.6, 151.6, 148.3, 145.7 (q, *J* = 1.2 Hz), 136.3, 129.2 (q, *J* = 32.5 Hz) 129.0, 124.9 (q, *J* = 3.8 Hz), 124.4 (q, *J* = 272.1 Hz), 121.7, 113.5, 113.2, 56.3, 56.2, 52.0; ^19^F-NMR (659 MHz, CDCl_3_) δ –62.36. 


*4,5-Dimethoxy-4′-(trifluoromethyl)-[1,1′-biphenyl]-2-carboxylic acid (*
**50**
*)*


Following general procedure E, **44** (0.5 g, 0.99 mmol) was reacted to afford 0.46 g (98%) of **50** as a pale-yellow solid. Mp = 190–193 °C; ^1^H-NMR (400 MHz, CDCl_3_) δ 7.63 (d, *J* = 8.0 Hz, 2H), 7.58 (s, 1H), 7.41 (d, *J* = 8.0 Hz, 2H), 6.73 (s, 1H), 3.97 (s, 3H), 3.93 (s, 3H); ^13^C-NMR (101 MHz, CDCl_3_) δ 170.32, 152.26, 148.32, 145.40, 137.44, 129.18, 124.93, 124.41 (q, *J* = 272.3 Hz), 120.06, 113.89, 113.86, 56.35, 56.33.


*8,9-Dimethoxy-3-(trifluoromethyl)-6H-benzo[c]chromen-6-one (*
**56**
*)*


Following general procedure F, **50** (0.44 g, 1.35 mmol) was reacted and purified by flash chromatography (pentane/EtOAc) to afford 0.36 g (84%) of **56** as a brown solid. Mp = 233–234 °C; ^1^H-NMR (400 MHz, DMSO-d_6_) δ 8.67 (d, *J* = 8.1 Hz, 1H), 7.95 (s, 1H), 7.82 (d, *J* = 1.7 Hz, 1H), 7.74 (dd, *J* = 8.1, 1.7 Hz, 1H), 7.66 (s, 1H), 4.06 (s, 3H), 3.94 (s, 3H); ^13^C-NMR (101 MHz, DMSO-d_6_) δ 159.4, 155.2, 150.8, 150.2, 129.4 (q, *J* = 32.7 Hz), 128.1, 125.0, 123.6 (q, *J* = 270.2 Hz), 121.6, 120.7 (q, *J* = 3.7 Hz), 114.4 (q, *J* = 3.7 Hz), 114.2, 109.9, 105.0, 56.6, 55.9; ^19^F-NMR (659 MHz, DMSO-d_6_) δ –60.88. 


*8,9-Dihydroxy-3-(trifluoromethyl)-6H-benzo[c]chromen-6-one (*
**62**
*)*


Following general procedure B, **56** (0.34 g, 1.05 mmol) was reacted and triturated with Et_2_O to afford 0.26 g (84%) of **62** as a grey solid. Mp = 293–295 °C; ^1^H-NMR (400 MHz, DMSO-d_6_) δ 10.59 (br s, 1H), 10.44 (br s, 1H), 8.24 (d, *J* = 8.6 Hz, 1H), 7.69–7.68 (m, 2H), 7.62 (dd, *J* = 8.6, 1.7 Hz, 1H), 7.59 (s, 1H); ^13^C-NMR (101 MHz, DMSO-d_6_) δ 159.36, 153.44, 149.89, 148.4, 128.82 (q, *J* = 32.6 Hz), 126.64, 123.98, 123.71 (q, *J* = 272.2 Hz), 121.76, 120.82 (q, *J* = 3.8 Hz), 114.50, 114.22 (q, *J* = 3.8 Hz), 113.18, 108.61; ^19^F-NMR (659 MHz, DMSO-d_6_) δ –60.96; HRMS (ESI) calculated for C_14_H_8_F_3_O_4_ [M + H]^+^ 297.0369, found 297.0360. 


**8,9-Dihydroxy-3-methyl-6*H*-benzo[*c*]chromen-6-one (63)**



*Methyl 4,5-dimethoxy-4′-methyl-[1,1′-biphenyl]-2-carboxylate (*
**45**
*)*


Following general procedure D, **9** (0.50 g, 1.81 mmol) and 4-tolylboronic acid (**39**) (0.3 g, 2.181 mmol). were reacted and purified by flash chromatography (pentane/EtOAc) to afford 0.44 (84%) of **45** as a beige solid. Mp = 87–91 °C; ^1^H-NMR (400 MHz, DMSO-d_6_) δ 7.30 (s, 1H), 7.20–7.15 (m, 4H), 6.88 (s, 1H), 3.84 (s, 3H), 3.82 (s, 3H), 3.56 (s, 3H), 2.34 (s, 3H); ^13^C NMR (101 MHz, DMSO-d_6_) δ 167.92, 150.85, 147.33, 137.88, 136.02, 135.65, 128.53, 128.23, 121.86, 113.64, 112.62, 55.71, 55.68, 51.62, 20.70.


*4,5-Dimethoxy-4′-methyl-[1,1′-biphenyl]-2-carboxylic acid (*
**51**
*)*


Following general procedure E, **45** (0.39 g, 1.35 mmol) was reacted to afford 0.28 g (76%) of **51** as a white solid. Mp = 226–229 °C; ^1^H-NMR (400 MHz, DMSO-d_6_) δ 12.41 (br s, 1H), 7.31 (s, 1H), 7.20–7.15 (m, 4H), 6.82 (s, 1H), 3.82 (s, 3H), 3.82 (s, 3H), 2.33 (s, 3H); ^13^C-NMR (101 MHz, DMSO-d_6_) δ 168.83, 150.48, 147.18, 138.32, 135.82, 135.53, 128.42, 128.38, 123.07, 113.75, 112.81, 55.65, 55.63, 20.70.


*8,9-Dimethoxy-3-methyl-6H-benzo[c]chromen-6-one (*
**57**
*)*


Following general procedure F, **51** (0.25 g, 0.91 mmol) was reacted and purified by flash chromatography (pentane/EtOAc) to afford 0.09 g (38%) of **60** as a yellow solid. Mp = 217–221 °C; ^1^H-NMR (400 MHz, DMSO-d_6_) δ 8.28 (d, *J* = 8.5 Hz, 1H), 7.80 (s, 1H), 7.59 (s, 1H), 7.23–7.22 (m, 2H), 4.03 (s, 3H), 3.90 (s, 3H), 2.41 (s, 3H); ^13^C-NMR (101 MHz, DMSO-d_6_) δ 160.08, 155.20, 150.37, 149.60, 140.07, 129.79, 125.44, 123.38, 117.02, 115.26, 112.92, 109.76, 103.98, 56.40, 55.74, 20.83.


*8,9-Dihydroxy-3-methyl-6H-benzo[c]chromen-6-one (*
**63**
*)*


Following general procedure B, **57** (0.083 g, 0.307 mmol) was reacted and triturated with Et_2_O to afford 0.07 g (97%) of **67** as a brown solid. Mp = 270–272 °C; ^1^H-NMR (400 MHz, DMSO-d_6_) δ 9.84 (br s, 2H), 7.91 (d, *J* = 8.5 Hz, 1H), 7.56 (s, 1H), 7.54 (s, 1H), 7.16–7.15 (m, 2H), 2.37 (s, 3H); ^13^C-NMR (101 MHz, DMSO-d_6_) δ 160.10, 153.35, 150.16, 147.00, 139.33, 128.42, 125.49, 122.43, 117.03, 115.42, 114.30, 112.13, 107.54, 20.81; HRMS (ESI) calculated for C_14_H_11_O_4_ [M + H]^+^ 243.0657, found 243.0659. 


**8,9-Dihydroxy-6-oxo-6H-benzo[c]chromene-3-carbaldehyde (64)**



*Methyl 4′-formyl-4,5-dimethoxy-[1,1′-biphenyl]-2-carboxylate (*
**46**
*)*


Following general procedure D, **9** (0.600 g, 2.18 mmol), 4-formylbenzenboronic acid (**40**) (0.36 g, 2.40 mmol), were reacted and purified by flash chromatography (pentane/EtOAc) to afford 0.56 g (85%) of **46** as a white solid. Mp = 142–144 °C. ^1^H-NMR (400 MHz, CDCl_3_): *δ* 10.01 (s, 1H), 7.86 (d, *J* = 8.2 Hz, 2H), 7.46 (s, 1H,), 7.41 (d, *J* = 8.2 Hz, 2H), 6.75 (s, 1H), 3.93 (s, 3H), 3.89 (s, 3H), 3.59 (s, 3H), ^13^C NMR (101 MHz, CDCl_3_) δ 191.91, 167.39, 151.41, 148.24, 148.22, 136.15, 134.93, 129.26, 121.58, 113.17, 113.05, 56.13, 56.10, 51.83.


*4′-Formyl-4,5-dimethoxy-[1,1′-biphenyl]-2-carboxylic acid (*
**52**
*)*


Following general procedure E, **46** (0.65 g, 2.16 mmol) was reacted to afford 0.55 g (89%) of **52** as a white solid. Mp = 249–251 °C; ^1^H NMR (400 MHz, CDCl_3_) δ 10.04 (s, 1H), 7.91–7.83 (m, 2H), 7.56 (s, 1H), 7.48–7.41 (m, 2H), 6.74 (s, 1H), 3.95 (s, 3H), 3.93 (s, 3H); ^13^C NMR (101 MHz, CDCl_3_) δ 192.17, 171.83, 152.15, 148.21, 148.05, 137.42, 135.02, 129.40, 129.31, 120.03, 113.72, 113.51, 56.17.


*8,9-Dimethoxy-6-oxo-6H-benzo[c]chromene-3-carbaldehyde (*
**58**
*)*


Following general procedure F, **52** (0.55 g, 1.92 mmol) was reacted and purified by flash chromatography (pentane/EtOAc) to afford 0.22 g (40%) of **58** as a white powder. Mp = 301–303 °C; ^1^H NMR (400 MHz, DMSO-d_6_) δ 10.06 (s, 1H), 8.60 (d, *J* = 8.1 Hz, 1H), 8.02–7.75 (m, 3H), 7.62 (s, 1H), 4.04 (s, 3H), 3.92 (s, 3H); ^13^C NMR (101 MHz, DMSO-d_6_) δ 192.54, 160.00, 155.62, 151.40, 150.91, 136.98, 128.75, 125.00, 124.73, 123.54, 118.66, 110.44, 105.64, 57.04, 56.38.


*8,9-Dihydroxy-6-oxo-6H-benzo[c]chromene-3-carbaldehyde (*
**64**
*)*


Following general procedure B, **58** (0.07 g, 0.24 mmol) was reacted and triturated with Et_2_O to afford 0.030 g (47%) of **64** as yellow solid. Mp = 352–354 °C; ^1^H NMR (400 MHz, DMSO-d_6_) δ 10.73–10.35 (m, 2H), 10.04 (s, 1H), 8.26 (d, *J* = 8.5 Hz, 1H), 7.83 (dq, *J* = 3.8, 1.6 Hz, 2H), 7.70 (s, 1H), 7.59 (s, 1H); ^13^C NMR (101 MHz, DMSO-d_6_) δ 192.51, 159.99, 153.83, 150.66, 149.06, 136.52, 127.33, 124.95, 124.07, 123.80, 118.63, 114.99, 113.87, 109.35; HRMS (ESI) m/z: 255.0301 [M − H]^−^, calculated for C_14_H_7_O_5_ 255.0299.


**8,9-Dihydroxy-3-(methylsulfonyl)-6H-benzo[c]chromen-6-one (65)**



*Methyl 4,5-dimethoxy-4′-(methylsulfonyl)-[1,1′-biphenyl]-2-carboxylate (*
**47**
*)*


Following general procedure D, **9** (0.500 g, 1.82 mmol), 4-(methanesulfonyl)phenylboronic acid (**41**) (400 mg, 2.00 mmol), were reacted and purified by flash chromatography (pentane/EtOAc) to afford 0.52 g (82%) of **47** as a white solid. Mp = 301–303 °C; ^1^H NMR (400 MHz, DMSO-d_6_) δ 7.99–7.87 (m, 2H), 7.59–7.49 (m, 2H), 7.41 (s, 1H), 6.95 (s, 1H), 3.86 (s, 6H), 3.59 (s, 3H), 3.26 (s, 3H); ^13^C NMR (101 MHz, DMSO-d_6_) δ 166.95, 151.21, 148.05, 146.24, 139.15, 134.52, 129.46, 126.52, 121.43, 113.83, 112.88, 55.85, 55.78, 51.81, 43.59.


*4,5-Dimethoxy-4′-(methylsulfonyl)-[1,1′-biphenyl]-2-carboxylic acid (*
**53**
*)*


Following general procedure E, **47** (0.47 g, 1.34 mmol) was reacted to afford 0.31 g (69%) of **53** as a beige crystal. Mp = 201–203 °C; ^1^H NMR (400 MHz, DMSO-d_6_) δ 12.62 (s, 1H), 7.95–7.88 (m, 2H), 7.60–7.53 (m, 2H), 7.42 (s, 1H), 6.89 (s, 1H), 3.84 (s, 3H), 3.85 (s, 3H), 3.26 (s, 3H); ^13^C NMR (101 MHz, DMSO-d_6_) δ 167.98, 150.84, 147.92, 146.77, 139.02, 134.38, 129.60, 126.40, 122.69, 113.87, 113.09, 55.81, 55.72, 43.56.


*8,9-Dimethoxy-3-(methylsulfonyl)-6H-benzo[c]chromen-6-one (*
**59**
*)*


Following general procedure I, **53** (0.2 g, 0.59 mmol) was reacted and purified by flash chromatography (pentane/EtOAc) to afford 0.18 g (92%) of **59** as a white powder. Mp = 353–355 °C; ^1^H NMR (400 MHz, DMSO-d_6_) δ 8.72 (d, *J* = 8.4 Hz, 1H), 7.97 (s, 1H), 7.92 (d, *J* = 1.8 Hz, 1H), 7.89 (dd, *J* = 8.3, 1.9 Hz, 1H), 7.67 (s, 1H), 4.07 (s, 3H), 3.95 (s, 3H), 3.34 (s, 3H); ^13^C NMR (201 MHz, DMSO-d_6_) δ 158.97, 155.22, 150.98, 149.83, 141.07, 127.76, 124.64, 122.20, 121.87, 115.60, 114.23, 110.14, 105.18, 56.43, 55.83, 43.20.


*8,9-Dihydroxy-3-(methylsulfonyl)-6H-benzo[c]chromen-6-one (*
**65**
*)*


Following general procedure B, **59** (0.07 g, 0.21 mmol) was reacted and triturated with Et_2_O to afford 0.044 g (69%) of **65** as beige solid. Mp = 319–321 °C; ^1^H NMR (400 MHz, DMSO-d_6_) δ 10.57 (s, 2H), 8.33 (d, *J* = 8.4 Hz, 1H), 7.86 (d, *J* = 1.9 Hz, 1H), 7.83 (dd, *J* = 8.3, 1.9 Hz, 1H), 7.71 (s, 1H), 7.60 (s, 1H), 3.31 (s, 3H); ^13^C NMR (101 MHz, DMSO-d_6_) δ 159.32, 153.45, 149.74, 148.65, 140.49, 126.54, 124.00, 122.64, 122.35, 115.89, 114.51, 113.35, 108.92, 43.34; HRMS (ESI) m/z: 305.0125 [M − H]^−^, calculated for C_14_H_9_O_6_S 300.0127.


**8,9-Dihydroxy-6-oxo-6H-benzo[c]chromene-3-sulfonamide (66)**



*Methyl 4,5-dimethoxy-4′-sulfamoyl-[1,1′-biphenyl]-2-carboxylate (*
**48**
*)*


Following general procedure D, **9** (0.2 g, 0.73 mmol), (4-aminosulfonylphenyl)boronic acid (**42**) (0.16 g, 0.80 mmol), were reacted and purified by flash chromatography (pentane/EtOAc) to afford 0.15 g (58%) of **48** as a white solid. Mp = 198–200 °C; ^1^H NMR (400 MHz, DMSO-d_6_) δ 7.83–7.77 (m, 2H), 7.46–7.41 (m, 2H), 7.37 (d, *J* = 2.0 Hz, 3H), 6.92 (s, 1H), 3.83 (s, 3H), 3.83 (s, 3H), 3.57 (s, 3H); ^13^C NMR (101 MHz, DMSO-d_6_) δ 166.95, 151.21, 148.05, 146.24, 139.15, 134.52, 129.46, 126.52, 121.43, 113.83, 112.88, 55.85, 55.78, 51.81, 43.59.


*4,5-Dimethoxy-4′-sulfamoyl-[1,1′-biphenyl]-2-carboxylic acid (*
**54**
*)*


Following general procedure E, **48** (0.5 g, 1.42 mmol) was reacted to afford 0.48 g (quant.) of **54** as a white solid. Mp = 223–225 °C; ^1^H NMR (400 MHz, DMSO-d_6_) δ 12.56 (s, 1H), 7.85–7.77 (m, 2H), 7.52–7.44 (m, 2H), 7.40 (d, *J* = 3.5 Hz, 3H), 6.88 (s, 1H), 3.84 (s, 3H), 3.84 (s, 3H); ^13^C NMR (101 MHz, DMSO-d_6_) δ 168.17, 150.80, 147.80, 144.92, 142.37, 134.66, 129.15, 125.14, 122.73, 113.91, 113.07, 55.82, 55.80, 55.74.


*8,9-Dimethoxy-6-oxo-6H-benzo[c]chromene-3-sulfonamide (*
**60**
*)*


Following general procedure I, **54** (0.2 g, 0.59 mmol) was reacted and purified by flash chromatography (pentane/EtOAc) to afford 0.12 g (60%) of **60** as a beige powder. Mp = 320–322 °C; ^1^H NMR (400 MHz, DMSO-d_6_) δ 8.66–8.61 (m, 1H), 7.93 (s, 1H), 7.79 (dd, *J* = 8.3, 1.9 Hz, 1H), 7.76 (d, *J* = 1.7 Hz, 1H), 7.66 (s, 1H), 7.57 (s, 2H), 4.06 (s, 3H), 3.94 (s, 3H); ^13^C NMR (101 MHz, DMSO-d_6_) δ 159.52, 155.24, 150.73, 149.88, 144.63, 128.27, 124.61, 121.17, 120.90, 114.33, 114.10, 109.97, 104.98, 56.61, 55.94.


*8,9-Dihydroxy-6-oxo-6H-benzo[c]chromene-3-sulfonamide (*
**66**
*)*


Following general procedure B, **60** (0.12 g, 0.35 mmol) was reacted and triturated with Et_2_O to afford 0.073 g (67%) of **66** as beige solid. Mp = 319–321 °C; ^1^H NMR (400 MHz, DMSO-d_6_) δ 10.58 (s, 1H), 10.47 (s, 1H), 8.27 (d, *J* = 8.4 Hz, 1H), 7.74 (dd, *J* = 8.3, 1.9 Hz, 1H), 7.71 (d, *J* = 1.8 Hz, 1H), 7.68 (s, 1H), 7.59 (s, 1H), 7.54 (s, 2H); ^13^C NMR (101 MHz, DMSO-d_6_) δ 159.50, 153.44, 149.61, 148.32, 144.05, 126.88, 123.70, 121.34, 121.07, 114.47, 114.30, 113.11, 108.67; HRMS (ESI) m/z: 308.0229 [M + H]^+^, calculated for C_13_H_10_NO_6_S 308.0231.


**8,9-Dihydroxy-6-oxo-6H-benzo[c]chromene-3-carboxylic acid (81)**



*2-Bromo-4,5-dimethoxybenzaldehyde (*
**68**
*)*


The compound was obtained following the literature procedure [25]. To a solution of 3,4-dimethoxybenzaldehyde (5.00 g, 30.09 mmol) in acetic acid (40.0 mL), a solution of bromine (4.65 mL, 90.27 mmol) in acetic acid (15.0 mL) was added slowly through an addition funnel at ice-cold conditions. The resulting reaction mixture was allowed to stir at room temperature overnight. Addition of ice-cold water (40.0 mL) to the reaction mixture led to a solid precipitate, which was filtered and thoroughly washed with cold water. The crude product was recrystallized from a methanol–water mixture (6:1) to yield 6.0 g (81%) of **68** as a colorless crystalline solid. Mp = 156–157 °C; ^1^H NMR (400 MHz, DMSO-d_6_) δ 10.04 (s, 1H), 7.31 (s, 1H), 7.30 (s, 1H), 3.88 (s, 3H), 3.80 (s, 3H); ^13^C NMR (101 MHz, DMSO-d_6_) δ 190.65, 154.94, 149.04, 126.16, 119.77, 116.41, 110.95, 56.94, 56.15.


*Ethyl 2′-formyl-4′,5′-dimethoxy-[1,1′-biphenyl]-4-carboxylate (*
**72**
*)*


Following general procedure D, **68** (1.00 g, 4.08 mmol), 4-ethoxycarbonylphenylboronic acid (**69**) (0.38 g, 4.48 mmol), were reacted and purified by flash chromatography to afford 0.95 g (74%) of **72** as a white solid. Mp = 96–98 °C; ^1^H NMR (400 MHz, DMSO-d_6_) δ 9.69 (s, 1H), 8.08–7.95 (m, 2H), 7.65–7.55 (m, 2H), 7.43 (s, 1H), 7.03 (s, 1H), 4.34 (q, *J* = 7.1 Hz, 2H), 3.90 (s, 3H), 3.86 (s, 3H), 1.33 (t, *J* = 7.1 Hz, 3H); ^13^C NMR (101 MHz, DMSO-d_6_) δ 190.18, 165.92, 153.74, 149.26, 142.42, 139.56, 131.05, 129.64, 129.45, 126.53, 113.56, 109.26, 61.32, 56.52, 56.12, 14.63.


*4′-(Ethoxycarbonyl)-4,5-dimethoxy-[1,1′-biphenyl]-2-carboxylic acid (*
**75**
*)*


Following general procedure J, **72** (0.500 g, 1.59 mmol) was reacted to afford 0.40 g (76%) of **75** as a white solid. Mp = 151–153 °C; ^1^H NMR (400 MHz, CDCl_3_) δ 8.12 (s, 1H), 8.09–8.01 (m, 2H), 7.57 (s, 1H), 7.42–7.32 (m, 2H), 6.74 (s, 1H), 4.40 (q, *J* = 7.1 Hz, 2H), 3.96 (s, 3H), 3.93 (s, 3H), 1.41 (t, *J* = 7.1 Hz, 3H); ^13^C NMR (101 MHz, CDCl_3_) δ 171.94, 166.58, 152.06, 148.04, 146.23, 137.77, 130.05, 129.55, 129.13, 128.69, 120.05, 113.68, 60.98, 56.16, 56.15, 14.37.


*Ethyl 8,9-dimethoxy-6-oxo-6H-benzo[c]chromene-3-carboxylate (*
**78**
*)*


The compound was obtained following the literature procedure [19]. In a 20 mL vial **75** (0.38 g, 1.15 mmol), Pd(OAc)_2_ (0.013 g, 0.05 mmol), N-acetylglycine (0.020 g, 0.17 mmol), KOAc (0.22 g, 2.30 mmol), and diacetoxiodo benzene (0.74 g, 2.30 mmol) were solubilized in t-BuOH (15.0 mL). The resulting mixture was stirred at 80 °C for 12 h. The solution was then rinsed with EtOAc (20.0 mL) and filtered through a Celite cake. Water (40.0 mL) was then added, and the aqueous layer was extracted with EtOAc (3 × 30 mL). The organic phase was dried (Na_2_SO_4_), filtered, and concentrated in vacuum to give 0.3 g of crude product. Crystallization from MeOH afforded 0.04 g (11%) of **78** as a white solid. Mp = 275–277 °C; ^1^H NMR (400 MHz, CDCl_3_) δ 8.04–7.89 (m, 3H), 7.75 (s, 1H), 7.45 (s, 1H), 4.42 (q, *J* = 7.1 Hz, 2H), 4.11 (s, 3H), 4.01 (s, 3H), 1.43 (t, *J* = 7.1 Hz, 3H); ^13^C NMR (101 MHz, CDCl_3_) δ 165.35, 160.53, 155.14, 150.94, 150.54, 131.31, 128.73, 125.06, 122.11, 121.82, 118.91, 115.19, 110.62, 103.19, 61.48, 56.40, 56.38, 14.28.


*8,9-Dihydroxy-6-oxo-6H-benzo[c]chromene-3-carboxylic acid (*
**81**
*)*


Following general procedure B, **78** (0.030 g, 0.09 mmol) was reacted and triturated with Et_2_O to afford 0.015 g (60%) of **81** as dark-brown powder. Mp = 352–354 °C; ^1^H NMR (400 MHz, DMSO-d_6_) δ 10.48 (br s, 3H), 8.16 (d, *J* = 8.4 Hz, 1H), 7.85 (dd, *J* = 8.3, 1.6 Hz, 1H), 7.77 (d, *J* = 1.6 Hz, 1H), 7.65 (s, 1H), 7.57 (s, 1H); ^13^C NMR (101 MHz, DMSO-d_6_) δ 171.56, 156.76, 156.59, 152.65, 144.03, 143.40, 142.46, 130.93, 118.23, 109.29, 108.81, 100.74, 100.59; HRMS (ESI) m/z: 271.0252 [M − H]^−^, calculated for C_14_H_7_O_6_ 271.0248.


**8,9-Dihydroxy-6-oxo-6*H*-benzo[*c*]chromene-3-carboxamide (82):**



*2′-Formyl-4′,5′-dimethoxy-[1,1′-biphenyl]-4-carboxamide (*
**73**
*)*


Following general procedure D, **68** (500 mg, 2.04 mmol), (4-carbamoylphenyl)boronic acid (**70**) (0.4 g, 2.45 mmol), were reacted and purified by flash chromatography to afford 0.49 g (84%) of **73** as a white solid. Mp = 173–175 °C. ^1^H-NMR (400 MHz, DMSO-d_6_) δ 9.72 (s, 1H), 8.08 (br s, 1H), 8.00–7.97 (m, 2H), 7.55–7.53 (m, 2H), 7.46 (br s, 1H), 7.43 (s, 1H), 7.04 (s, 1H), 3.92 (s, 3H), 3.87 (s, 3H). ^13^C-NMR (101 MHz, DMSO-d_6_) δ 189.83, 167.41, 153.30, 148.66, 139.98, 139.64, 133.58, 130.13, 127.44, 126.06, 113.19, 108.65, 56.06, 55.67.


*4′-Carbamoyl-4,5-dimethoxy-[1,1′-biphenyl]-2-carboxylic acid (*
**76**
*)*


Following general procedure J, **73** (0.46 g, 1.62 mmol) was reacted to afford 0.44 g (91%) of **76** as a white solid. Mp = 241–242 °C. ^1^H-NMR (400 MHz, DMSO-d_6_) δ 12.49 (br s, 1H), 7.99 (br s, 1H), 7.88–7.85 (m, 2H), 7.38–7.35 (m, 4H), 6.88 (s, 1H), 3.84 (s, 3H), 3.84 (s, 3H). ^13^C-NMR (101 MHz, DMSO-d_6_) δ 168.45, 167.69, 150.65, 147.60, 144.23, 135.03, 132.52, 128.45, 126.99, 122.95, 113.74, 112.94, 55.74, 55.69.


*8,9-Dimethoxy-6-oxo-6H-benzo[c]chromene-3-carboxamide (*
**79**
*)*


Following general procedure G, **76** (0.2 g, 0.66 mmol) was reacted and triturated from MeOH and CH_2_Cl_2_ to afford 0.07 g (34%) of **79** as a yellow solid. Mp = 333–335 °C; ^1^H-NMR (400 MHz, DMSO-d_6_) δ 8.49 (d, *J* = 8.2 Hz, 1H), 8.17 (br s, 1H), 7.89–7.86 (m, 3H), 7.62 (s, 1H), 7.59 (br s, 1H), 4.05 (s, 3H), 3.92 (s, 3H); ^13^C-NMR (101 MHz, DMSO-d_6_) δ 166.43, 159.81, 155.18, 150.43, 150.03, 135.04, 128.78, 123.65, 123.31, 120.30, 115.99, 113.89, 109.88, 104.77, 56.54, 55.85.


*8,9-Dihydroxy-6-oxo-6H-benzo[c]chromene-3-carboxamide (*
**82**
*)*


Following general procedure B, **79** (0.048 g, 0.16 mmol) was reacted and triturated with Et_2_O to afford 0.035 g (81%) of **82** as a white powder. Mp > 350 °C; ^1^H-NMR (400 MHz, DMSO-d_6_) δ 10.51 (br s, 1H), 10.38 (br s, 1H), 8.15–8.13 (m, 1H), 8.12 (br s, 1H), 7.85–7.82 (m, 2H), 7.67 (s, 1H), 7.59 (s, 1H), 7.54 (br s, 1H); ^13^C-NMR (101 MHz, DMSO-d_6_) δ 166.53, 159.83, 153.37, 149.80, 147.98, 134.51, 127.42, 123.47, 122.72, 120.52, 115.99, 114.42, 112.99, 108.45; HRMS (ESI) calculated for C_14_H_10_NO_5_ [M + H]^+^ 272.0553, found 272.0546. 


**8,9-Dihydroxy-6-oxo-6*H*-benzo[*c*]chromene-3-carbonitrile (83)**



*2′-Formyl-4′,5′-dimethoxy-[1,1′-biphenyl]-4-carbonitrile (*
**74**
*)*


Following general procedure D, **68** (490 mg, 2.00 mmol), 4-cyanophenylboronic acid (**71**) (0.32 g, 2.20 mmol), were reacted and purified by flash chromatography to afford 0.43 g (81%) of **74** as a white solid. Mp = 218–220 °C; ^1^H-NMR (400 MHz, DMSO-d_6_) δ 9.70 (s, 1H), 7.97–7.95 (m, 2H), 7.69–7.67 (m, 2H), 7.46 (s, 1H), 7.05 (s, 1H), 3.92 (s, 3H), 3.88 (s, 3H); ^13^C-NMR (101 MHz, DMSO-d_6_) δ 189.64, 153.27, 148.96, 142.22, 138.32, 132.14, 131.18, 126.06, 118.74, 113.26, 110.60, 109.12, 56.12, 55.72.


*4′-Cyano-4,5-dimethoxy-[1,1′-biphenyl]-2-carboxylic acid (*
**77**
*)*


Following general procedure J, **74** (200 mg, 0.75 mmol) was reacted to afford 0.17 g (82%) of **77** as a white solid. Mp = 232–233 °C; ^1^H-NMR (400 MHz, DMSO-d_6_) δ 12.59 (br s, 1H), 7.85–7.83 (m, 2H), 7.51–7.48 (m, 2H), 7.41 (s, 1H), 6.88 (s, 1H), 3.84 (s, 3H), 3.84 (s, 3H); ^13^C-NMR (101 MHz, DMSO-d_6_) δ 167.97, 150.83, 147.95, 146.47, 134.35, 131.63, 129.75, 122.62, 119.00, 113.70, 113.07, 109.40, 55.81, 55.70.


*8,9-Dimethoxy-6-oxo-6H-benzo[c]chromene-3-carbonitrile (*
**80**
*)*


Following general procedure G, **77** (0.16 g, 0.56 mmol) was reacted and triturated from MeOH and CH_2_Cl_2_ to afford 0.03 g (19%) of **80** as a pale-yellow solid. Mp 337–339 °C; ^1^H-NMR (400 MHz, DMSO-d_6_) δ 8.64 (d, *J* = 8.4 Hz, 1H), 8.01 (d, *J* = 1.6 Hz, 1H), 7.95 (s, 1H), 7.86 (dd, *J* = 8.4, 1.6 Hz, 1H), 7.65 (s, 1H), 4.05 (s, 3H), 3.94 (s, 3H); ^13^C-NMR (101 MHz, DMSO-d_6_) δ 159.19, 155.24, 151.07, 149.98, 127.94, 127.76, 124.97, 122.44, 121.11, 118.08, 114.41, 111.30, 110.01, 105.27, 56.67, 55.97.


*8,9-Dihydroxy-6-oxo-6H-benzo[c]chromene-3-carbonitrile (*
**83**
*)*


Following general procedure B, **80** (0.025 g, 0.09 mmol) was reacted and triturated with Et_2_O to afford 0.008 g (37%) of **83** as a pale-brown solid. Mp = 348–350 °C; ^1^H-NMR (400 MHz, DMSO-d_6_) δ 8.25 (d, *J* = 8.3 Hz, 1H), 7.94 (d, *J* = 1.6 Hz, 1H), 7.75 (dd, *J* = 8.3, 1.6 Hz, 1H), 7.69 (s, 1H), 7.58 (s, 1H); ^13^C-NMR (101 MHz, DMSO-d_6_) δ 159.17, 153.70, 149.76, 148.82, 127.77, 126.51, 124.00, 122.68, 121.06, 118.17, 114.36, 113.18, 110.61, 108.85; HRMS (ESI) calculated for C_14_H_6_NO_4_ [M − H]^–^ 252.0302, found 252.0310. 


**8,9-Dihydroxy-3-methoxy-6H-benzo[c]chromen-6-one (90)**



*Methyl 4-(benzyloxy)-2-bromo-5-hydroxybenzoate (*
**84**
*)*


To a solution of **8** (1.20 g, 4.59 mmol) in CH_2_Cl_2_ (50 mL) BBr_3_ in CH_2_Cl_2_ (1.0 M, 27.57 mmol) was added dropwise at 0 °C. The solution was stirred at room temperature for 2 h. MeOH (60.0 mL) was added to quench the reaction and the resulting solvent was concentrated under reduced pressure. Thionyl chloride (10.0 mL) was added, and the solution was refluxed for 2 h. The solvent was then removed under reduced pressure and MeOH (30.0 mL) was added. The resulting solution was stirred at room temperature for 1 h (until full conversion, reaction monitored by TLC). The solvent was then removed under reduces pressure to afford 1.1 g (97%) of **84** as a pink solid. Mp = 149–151 °C; ^1^H NMR (400 MHz, CDCl_3_) δ 7.48 (s, 1H), 7.17 (s, 1H), 3.90 (s, 3H); ^13^C NMR (101 MHz, CDCl_3_) δ 166.88, 148.05, 142.49, 122.66, 121.07, 118.56, 113.56, 52.62.


*Methyl 4,5-bis(benzyloxy)-2-bromobenzoate (*
**85**
*)*


Following general procedure K, **84** (0.700 g, 2.83 mmol) was reacted and purified by flash chromatography (pentane/EtOAc) to afford 0.96 g (79%) of **85** as a colorless oil. ^1^H NMR (400 MHz, CDCl_3_) δ 7.52 (s, 1H), 7.46–7.41 (m, 4H), 7.41–7.29 (m, 7H), 7.18 (s, 1H), 5.17 (s, 2H), 5.15 (s, 2H), 3.89 (s, 3H); ^13^C NMR (101 MHz, CDCl_3_) δ 165.73, 152.03, 147.41, 136.37, 135.86, 128.65, 128.55, 128.21, 128.08, 127.39, 127.25, 123.30, 119.53, 117.56, 114.59, 71.42, 71.09, 52.25.


*Methyl 4,5-bis(benzyloxy)-4′-methoxy-[1,1′-biphenyl]-2-carboxylate (*
**87**
*)*


Following general procedure D, **85** (0.96 g, 2.24 mmol), 4-methoxyphenylboronic acid (**86**) (0.34 g, 2.24 mmol), were reacted and purified by flash chromatography (pentane/EtOAc) to afford 0.71 g (69%) of **87** as a white solid. Mp = 151–153 °C; ^1^H NMR (400 MHz, CDCl_3_) δ 7.51 (d, *J* = 1.0 Hz, 1H), 7.50–7.45 (m, 2H), 7.46–7.41 (m, 2H), 7.41–7.29 (m, 6H), 7.19–7.12 (m, 2H), 6.94–6.88 (m, 2H), 6.88 (d, *J* = 0.9 Hz, 1H), 5.21 (s, 2H), 5.19 (s, 2H), 3.84 (s, 3H), 3.63 (s, 3H); ^13^C NMR (101 MHz, CDCl_3_) δ 168.25, 158.72, 151.10, 147.27, 137.38, 136.85, 136.51, 133.79, 129.51, 128.54, 128.52, 127.98, 127.93, 127.39, 127.26, 122.42, 116.57, 116.44, 113.31, 71.40, 70.90, 55.24, 51.79.


*4,5-Bis(benzyloxy)-4′-methoxy-[1,1′-biphenyl]-2-carboxylic acid (**88**)*


Following general procedure E, **87** (0.71 g, 1.56 mmol) was reacted to afford 0.65 g (95%) of **88** as a white solid. Mp = 218–220 °C; ^1^H NMR (400 MHz, CDCl_3_) δ 7.62 (s, 1H), 7.52–7.46 (m, 2H), 7.46–7.41 (m, 2H), 7.42–7.28 (m, 6H), 7.22–7.14 (m, 2H), 6.93–6.86 (m, 2H), 6.86 (s, 1H), 5.21 (s, 2H), 5.20 (s, 2H), 3.84 (s, 3H); ^13^C NMR (101 MHz, CDCl_3_) δ 172.18, 158.86, 151.77, 147.16, 138.49, 136.76, 136.37, 133.47, 129.73, 128.56, 128.53, 128.03, 127.95, 127.40, 127.25, 120.77, 117.05, 116.77, 113.40, 71.29, 70.84, 55.24.


*8,9-Bis(benzyloxy)-3-methoxy-6H-benzo[c]chromen-6-one (*
**89**
*)*


Following general procedure F, **88** (0.65 g, 1.47 mmol) was reacted to afford 0.21 g (33%) of **89** as a white powder. Mp = 208–210 °C; ^1^H NMR (400 MHz, CDCl_3_) δ 7.81 (d, *J* = 1.9 Hz, 1H), 7.70 (d, *J* = 8.9 Hz, 1H), 7.57–7.45 (m, 4H), 7.45–7.28 (m, 7H), 6.87 (dd, *J* = 8.8, 2.6 Hz, 1H), 6.82 (d, *J* = 2.5 Hz, 1H), 5.35 (s, 2H), 5.24 (s, 2H), 3.85 (d, *J* = 1.2 Hz, 3H); ^13^C NMR (101 MHz, CDCl_3_) δ 161.24, 160.84, 154.88, 152.25, 148.89, 136.27, 136.02, 130.50, 128.74, 128.58, 128.25, 128.05, 127.32, 127.10, 123.07, 113.23, 113.00, 112.28, 111.18, 104.72, 101.48, 71.02, 70.91, 55.63.


*8,9-Dihydroxy-3-methoxy-6H-benzo[c]chromen-6-one (*
**90**
*)*


Following general procedure H, **89** (0.21 g, 0.48 mmol) was reacted to afford 0.12 g (97%) of **90** as a white powder. Mp = 338–340 °C; ^1^H NMR (400 MHz, DMSO-d_6_) δ 10.56–9.86 (m, 2H), 7.93 (d, *J* = 9.6 Hz, 1H), 7.49 (d, *J* = 7.9 Hz, 2H), 6.97–6.83 (m, 2H), 3.81 (s, 3H); ^13^C NMR (101 MHz, DMSO-d_6_) δ 160.63, 160.52, 153.91, 151.86, 146.89, 129.20, 124.11, 114.65, 112.44, 111.58, 111.49, 107.61, 101.77, 56.10; HRMS (ESI) m/z: 257.0464 [M − H]^−^, calculated for C_14_H_9_O_5_ 257.0455.


**8,9-Dihydroxy-6-oxo-6*H*-benzo[*c*]chromen-3-yl acetate (94)**



*4,5-Bis(benzyloxy)-2-bromobenzoic acid (91)*


Following general procedure E, **85** (3.30 g, 7.37 mmol) was reacted to afford 3.09 g (97%) of **91** as a white solid. Mp = 152–154 °C; ^1^H NMR (400 MHz, CDCl_3_) δ 7.68 (d, *J* = 1.2 Hz, 1H), 7.50–7.26 (m, 10H), 7.21 (s, 1H), 5.19 (s, 2H), 5.16 (s, 2H); ^13^C NMR (101 MHz, CDCl_3_) δ 169.92, 152.71, 147.33, 136.24, 135.72, 128.68, 128.59, 128.27, 128.11, 127.39, 127.26, 121.62, 119.68, 118.11, 115.71, 71.28, 71.08.


*8,9-Bis(benzyloxy)-3-hydroxy-6H-benzo[c]chromen-6-one (*
**92**
*)*


Following general procedure A, **91** (0.64 g, 1.55 mmol) was reacted to afford 0.32 g of **92** that was ca. 85% pure according to NMR and was used in the next step without further purification. ^1^H NMR (400 MHz, DMSO-d_6_) δ 8.14 (OH, 1H), 7.82–7.25 (m, 15H), 5.43 (s, 2H), 5.26 (s, 2H)


*8,9-Bis(benzyloxy)-6-oxo-6H-benzo[c]chromen-3-yl acetate (*
**93**
*)*


To a solution of **92** (85% purity 150 mg = 128 mg, 0.301 mmol) in acetic anhydride (3 mL), sodium acetate (87 mg, 1.06 mmol) was added. The reaction mixture was stirred at 50 °C for 18 h. Then, the reaction mixture was cooled down to room temperature and water (5 mL) was added. The formed precipitate was collected by filtration, washed with water, dried, and triturated with MeOH to afford 0.094 g (67%) of **93** as a white solid. Mp = 200–205 °C; ^1^H-NMR (400 MHz, DMSO-d_6_) δ 8.43 (d, *J* = 8.7 Hz, 1H), 8.00 (s, 1H), 7.73 (s, 1H), 7.56–7.21 (m, 12H), 5.46 (s, 2H), 5.30 (s, 2H), 2.31 (s, 3H); ^13^C NMR (101 MHz, DMSO-d_6_) δ 168.94, 159.70, 154.55, 151.18, 150.74, 149.07, 136.56, 136.30, 129.28, 128.56, 128.48, 128.14, 127.95, 127.82, 127.45, 118.47, 115.66, 113.11, 110.66, 106.07, 70.39, 70.03, 20.89.


*8,9-Dihydroxy-6-oxo-6H-benzo[c]chromen-3-yl acetate (*
**94**
*)*


Following general procedure H, **89** (0.08 g, 0.17 mmol) was reacted and purified by flash chromatography (CH_2_Cl_2_/MeOH) to afford 0.01 g (20%) of **94** as a white powder. Mp = 210–215 °C; ^1^H-NMR (400 MHz, DMSO-d_6_) δ 8.10 (d, *J* = 9.1 Hz, 1H), 7.58 (s, 1H), 7.55 (s, 1H), 7.21 (s, 1H), 7.14 (d, *J* = 9.1 Hz, 1H), 2.30 (s, 3H); ^13^C-NMR (101 MHz, DMSO-d_6_) δ 168.97, 159.79, 153.49, 150.61, 150.44, 147.33, 127.82, 123.55, 118.43, 115.88, 114.30, 112.02, 110.58, 107.91, 20.89; HRMS (ESI) calculated for C_15_H_9_O_6_ [M − H]^–^ 285.0405, found 285.0416. 


**7,8-Dihydroxyisochromeno[3,4-e]indol-5(1H)-one (98)**



*1H-Indol-4-yl 4,5-bis(benzyloxy)-2-bromobenzoate (*
**96**
*)*


A solution of **91** (1.24 g, 3.00 mmol) in SOCl_2_ (20.0 mL) was refluxed for 2 h. The solvent was removed under reduced pressure and diluted with dichloromethane (20.0 mL). The benzyl chloride solution was then added dropwise at 0 °C to a solution of 4-hydroxyindole (**95**) (0.40 g, 3.00 mmol) and triethylamine (0.91 g, 9.00 mmol) in dichloromethane (20.0 mL). The reaction was stirred for 2 h at room temperature and then quenched with water (40.0 mL). The organic layer was washed with water (30.0 mL), dried over anhydrous sodium sulfate, and concentrated under reduced pressure to give a yellow oil. Purification by flash chromatography (65/35 pentane/EtOAc) afforded 1.03 g (65%) of **96** as a colorless oil. ^1^H NMR (400 MHz, CDCl_3_) δ 8.52 (t, *J* = 2.3 Hz, 1H), 7.91 (s, 1H), 7.53–7.36 (m, 10H), 7.34 (s, 1H), 7.22–7.17 (m, 2H), 7.07–7.03 (m, 1H), 7.00 (dd, *J* = 3.3, 2.4 Hz, 1H), 6.47 (ddd, *J* = 3.3, 2.1, 0.5 Hz, 1H), 5.22 (s, 4H); ^13^C NMR (101 MHz, cdcl_3_) δ 163.99, 152.55, 147.52, 143.58, 137.85, 136.33, 135.83, 128.77, 128.69, 128.37, 128.22, 127.60, 127.44, 124.95, 122.74, 121.96, 121.21, 119.75, 118.06, 115.56, 111.80, 109.65, 99.20, 71.50, 71.19.


*7,8-Bis(benzyloxy)isochromeno[3,4-e]indol-5(1H)-one (*
**97**
*)*


The compound was obtained following the literature procedure [26]. To a solution of **96** (1.00 g, 1.89 mmol) in dry DMA (10.0 mL), NaOAc (0.31 g, 3.78 mmol), Pd(OAc)_2_ (0.042 g, 0.19 mmol), and tricyclohexylphosphine tetrafluoroborate (0.210 g, 0.57 mmol) were added. The mixture was degassed and then heated at 175 °C for 1 h in the microwave. The reaction mixture was cooled to room temperature and then diluted with diethyl ether and filtered through Celite to remove the catalyst. The solvent was removed under reduced pressure to give 1.5 g of a dark oil. Purification by flash chromatography (60/40 pentane/EtOAc) and recrystallization from MeOH afforded 0.24 g (28%) of **97** as a white solid. Mp = 197–199 °C; ^1^H NMR (400 MHz, DMSO-d_6_) δ 11.55 (s, 1H), 7.99 (d, *J* = 8.8 Hz, 1H), 7.91 (s, 1H), 7.72 (s, 1H), 7.58–7.51 (m, 2H), 7.51–7.45 (m, 2H), 7.45–7.26 (m, 8H), 6.69 (ddd, *J* = 3.0, 2.0, 0.9 Hz, 1H), 5.45 (s, 2H), 5.27 (s, 2H); ^13^C NMR (101 MHz, DMSO-d_6_) δ 160.85, 155.06, 148.27, 144.66, 137.89, 137.20, 136.94, 132.45, 128.99, 128.90, 128.50, 128.33, 128.17, 127.89, 126.61, 117.14, 116.60, 112.72, 112.42, 109.42, 108.46, 105.96, 98.96, 70.67, 70.45.


*7,8-Dihydroxyisochromeno[3,4-e]indol-5(1H)-one (*
**98**
*)*


Following general procedure H, **97** (0.10 g, 0.22 mmol) was reacted to afford 0.05 g (84%) of **98** as a yellow solid. Mp > 350 °C; ^1^H NMR (400 MHz, DMSO-d_6_) δ 11.49 (s, 1H), 10.09 (br s, 2H), 7.68 (d, *J* = 8.7 Hz, 1H), 7.54 (s, 1H), 7.52 (s, 1H), 7.40 (d, *J* = 3.1 Hz, 1H), 7.37 (dd, *J* = 8.6, 0.9 Hz, 1H), 6.65 (dd, *J* = 3.1, 0.8 Hz, 1H); ^13^C NMR (101 MHz, DMSO-d_6_) δ 160.95, 153.91, 146.35, 144.24, 137.50, 130.95, 126.39, 117.22, 115.82, 114.75, 111.54, 109.42, 108.62, 107.68, 98.82; HRMS (ESI) m/z: 266.0472 [M − H]^−^, calculated for C_15_H_8_NO_4_ 266.0459.


**8,9-Dihydroxy-6H-dibenzo[*c,h*]chromen-6-one (103)**



*Methyl 4,5-dimethoxy-2-(naphthalen-2-yl)benzoate (*
**100**
*)*


Following general procedure D, **9** (0.55 g, 2.00 mmol), 2-naphtalenlboronic acid (**99**) (0.38 g, 2.20 mmol), were reacted and purified by flash chromatography (pentane/EtOAc) to afford 0.37 g (57%) of **100** as a colorless oil. ^1^H NMR (400 MHz, CDCl_3_) δ 7.90–7.75 (m, 4H), 7.53 (s, 1H), 7.51–7.39 (m, 3H), 6.91 (s, 1H), 3.97 (s, 3H), 3.90 (s, 3H), 3.58 (s, 3H); ^13^C NMR (101 MHz, CDCl_3_) δ 168.15, 151.34, 147.87, 139.42, 137.36, 133.26, 132.39, 128.00, 127.72, 127.54, 127.09, 126.66, 126.18, 125.88, 122.00, 113.94, 113.03, 56.12, 56.05, 51.78.


*4,5-Dimethoxy-2-(naphthalen-2-yl)benzoic acid (*
**101**
*)*


Following general procedure E, **100** (0.37 g, 1.14 mmol) was reacted to afford 0.29 g (82%) of **101** as a white solid. Mp = 233–235 °C; ^1^H NMR (400 MHz, CDCl_3_) δ 7.92–7.81 (m, 2H), 7.81–7.73 (m, 2H), 7.57 (s, 1H), 7.54–7.45 (m, 2H), 7.42 (dd, *J* = 8.4, 1.8 Hz, 1H), 6.85 (s, 1H), 3.95 (s, 3H), 3.93 (s, 3H); ^13^C NMR (101 MHz, CDCl_3_) δ 172.31, 151.97, 147.74, 139.21, 138.70, 133.15, 132.43, 128.02, 127.67, 127.63, 127.10, 126.73, 126.10, 125.89, 120.31, 114.29, 113.65, 56.14, 56.12.


*8,9-dimethoxy-6H-dibenzo[c,h]chromen-6-one (*
**102**
*)*


Following general procedure F, **101** (0.29 g, 0.99 mmol) was reacted to afford 0.200 g (69%) of **102** as a white powder. Mp = 202–204 °C; ^1^H NMR (400 MHz, CDCl_3_) δ 8.50–8.35 (m, 1H), 7.78 (d, *J* = 8.9 Hz, 1H), 7.76–7.72 (m, 1H), 7.63 (s, 1H), 7.62–7.58 (m, 1H), 7.58–7.48 (m, 2H), 7.30 (s, 1H), 4.04 (s, 3H), 3.96 (s, 3H); ^13^C NMR (101 MHz, CDCl_3_) δ 161.11, 155.06, 149.77, 146.43, 133.57, 130.41, 127.48, 127.40, 126.90, 124.22, 123.65, 121.93, 118.69, 114.06, 112.81, 110.20, 102.71, 56.24, 56.22.


*8,9-Dihydroxy-6H-dibenzo[c,h]chromen-6-one (*
**103**
*)*


Following general procedure B, **102** (0.120 g, 0.66 mmol) was reacted to afford 0.075 g (69%) of 103 as a dark brown solid. Mp > 350 °C; ^1^H NMR (400 MHz, DMSO-d_6_) δ 10.50 (br s, 1H), 10.28 (br s, 1H), 8.36–8.30 (m, 1H), 8.11 (d, *J* = 8.9 Hz, 1H), 8.02–7.94 (m, 1H), 7.88–7.78 (m, 1H), 7.71–7.58 (m, 4H); ^13^C NMR (101 MHz, DMSO-d_6_) δ 160.32, 153.99, 147.79, 145.80, 133.55, 129.24, 128.36, 127.72, 127.63, 124.62, 123.52, 121.45, 120.31, 114.82, 113.73, 113.00, 108.71; HRMS (ESI) m/z: 277.0515 [M − H]^−^, calculated for C_17_H_9_O_4_ 277.0506.


**3-Hydroxy-8,9-dimethoxy-6H-benzo[c]chromen-6-one (118)**


Following general procedure A, **8** (3.13 g, 12.00 mmol) was reacted to afford 2.91 g (89%) of **118** as a white solid. Mp = 329–331 °C; ^1^H NMR (400 MHz, DMSO-d_6_) δ 10.19 (s, 1H), 8.15 (s, 1H), 7.81–7.31 (m, 2H), 7.07–6.43 (m, 2H), 3.98 (s, 3H), 3.85 (s, 3H); ^13^C NMR (101 MHz, DMSO-d_6_) δ 160.68, 159.43, 155.46, 152.12, 149.23, 130.74, 124.99, 113.16, 112.00, 109.87, 103.67, 103.13, 56.63, 56.00; HRMS (ESI) m/z: 271.0617 [M − H]^−^, calculated for C_15_H_11_O_5_ 271.0612.


**3,9-Dihydroxy-8-methoxy-6H-benzo[c]chromen-6-one (119)**



*5-Bromo-4-formyl-2-methoxyphenyl acetate (*
**106**
*)*


To a solution of KBr (6.7 g, 52.10 mmol) and 4-formyl-2-methoxyphenyl acetate (104) (3.2 g, 16.50 mmol) in H2O (80 mL) was added dropwise Br_2_ (3.2 g, 20.02 mmol). The resulting mixture was stirred at room temperature for 12 h. The precipitate formed was collected by filtration to give 4.5 g (95%) of **106** as an orange powder. Mp = 137–139 °C; ^1^H NMR (400 MHz, CDCl_3_) δ 10.26 (d, *J* = 0.4 Hz, 1H), 7.51 (s, 1H), 7.35 (s, 1H), 3.88 (s, 3H), 2.33 (s, 3H); ^13^C NMR (101 MHz, CDCl_3_) δ 190.81, 167.90, 151.24, 144.95, 131.61, 127.98, 117.87, 112.11, 56.27, 20.53.


*4-Acetoxy-2-bromo-5-methoxybenzoic acid (*
**112**
*)*


The compound was synthetized following the literature procedure [27]. Compound **106** (1.00 g, 3.66 mmol) was dissolved in acetone (20.0 mL) and heated to reflux. A mixture of water (20.0 mL) and KMnO_4_ (1.15 g, 7.32 mmol) was added at refluxing temperature and stirred for 90 min at 50 °C. The solvent was removed under reduced pressure. The undesired by-product MnO_2_ was removed by filtration on a Celite bed and washed with 10% KOH solution (10 mL). The pH of the filtrate was adjusted to 2–2.5 using 1 N aq. HCl and extracted with EtOAc (3 × 30.0 mL). The organic phase was dried (Na_2_SO_4_), filtered, and concentrated under reduced pressure afforded 0.96 g (99%) of **112** as a white solid. The compound was used directly for the next step.


*2-Bromo-4-hydroxy-5-methoxybenzoic acid (*
**113**
*)*


The compound was synthetized following the literature procedure [28]. A solution of **112** (0.96 g, 3.64 mmol) in NaOH aq. solution (5%, 20.0 mL) was heated to reflux for 1 h. The reaction was then cooled down to 0 °C, acidified with 1M HCl and extracted with EtOAc (3 × 30.0 mL). The organic phase was dried (Na_2_SO_4_), filtered, and concentrated under reduced pressure to afford 0.43 g (45%) of **113** as a white solid. Mp = 193–195 °C; ^1^H-NMR (400 MHz, DMSO-d_6_) δ 12.91 (br s, 1H), 10.23 (s, 1H), 7.38 (s, 1H), 7.05 (s, 1H), 3.79 (s, 3H); ^13^C-NMR (101 MHz, DMSO-d_6_) δ 166.34, 150.38, 146.66, 122.09, 120.42, 114.81, 112.57, 55.76.


*3,9-Dihydroxy-8-methoxy-6H-benzo[c]chromen-6-one (*
**119**
*)*


Following general procedure A, **113** (0.2 g, 0.81 mmol) was reacted to afford 0.1 g (40%) of **119** as a dark brown solid. Mp = 320–322 °C; ^1^H NMR (400 MHz, DMSO-d_6_) δ 10.52 (s, 1H), 10.15 (s, 1H), 7.86 (d, *J* = 8.8 Hz, 1H), 7.53 (s, 1H), 7.47 (s, 1H), 6.79 (dd, *J* = 8.6, 2.4 Hz, 1H), 6.69 (d, *J* = 2.4 Hz, 1H), 3.88 (s, 3H); ^13^C NMR (101 MHz, DMSO-d_6_) δ 160.74, 159.41, 154.44, 152.14, 148.57, 130.99, 124.50, 113.40, 111.07, 110.96, 109.89, 107.17, 103.22, 56.20; HRMS (ESI) m/z: 257.0456 [M − H]^−^, calculated for C_14_H_9_O_5_ 257.0454.


**3,8-Dihydroxy-9-methoxy-6H-benzo[c]chromen-6-one (120)**



*4-Bromo-5-formyl-2-methoxyphenyl acetate (*
**107**
*)*


The compound was synthetized following the literature procedure [29]. To a stirred solution of 2-bromo-4-hydroxy-5-methoxybenzaldehyde (**105**) (1.00 g, 4.33 mmol) in CH_2_Cl_2_ (30.0 mL), acetic anhydride (0.82 mL, 8.66 mmol), DMAP (0.53 g, 4.33 mmol), and Et_3_N (0.60 mL, 4.33 mmol) were added at room temperature. The reaction mixture was stirred until all the starting material was consumed as monitored by TLC. Water (25.0 mL) and CH_2_Cl_2_ (25 mL) were added, and the two phases were separated. The aqueous layer was extracted with CH_2_Cl_2_ (20 mL). The combined organic layers were washed with brine (10.0 mL), dried over Na_2_SO_4_, filtered, and concentrated under reduced pressure to give the crude product. Purification by flash chromatography (40/60 EtOAc/hexanes) afforded 1.05 g (89%) of **107** as a white solid. Mp = 97–99 °C; ^1^H NMR (400 MHz, CDCl_3_) δ 10.18 (s, 1H), 7.62 (s, 1H), 7.17 (s, 1H), 3.92 (s, 3H), 2.32 (s, 3H); ^13^C NMR (101 MHz, CDCl_3_) δ 189.94, 168.32, 156.34, 139.60, 126.84, 125.61, 123.72, 116.81, 56.57, 20.47.


*5-Acetoxy-2-bromo-4-methoxybenzoic acid (*
**114**
*)*


The compound was synthetized following the literature procedure [27]. Compound **107** (1.00 g, 3.66 mmol) was dissolved in acetone (20.0 mL) and heated to reflux. A mixture of water (20.0 mL) and KMnO_4_ (1.15 g, 7.32 mmol) was added at refluxing temperature and stirred for 90 min at 50 °C. The solvent was removed under reduced pressure. The undesired by-product MnO_2_ was removed by filtration on a Celite bed and washed with 10% KOH solution (10 mL). The pH of the filtrate was adjusted to 2–2.5 using 1 N aq. HCl and extracted with EtOAc (3 × 30.0 mL). The organic phase was dried (Na_2_SO_4_), filtered, and concentrated under reduced pressure afforded 0.92 g (87%) of **114** as a white solid. Mp = 182–184 °C; ^1^H NMR (400 MHz, CDCl_3_) δ 7.81 (s, 1H), 7.25 (s, 1H), 3.90 (s, 3H), 2.33 (s, 3H); ^13^C NMR (101 MHz, CDCl_3_) δ 169.40, 168.40, 154.84, 138.43, 127.25, 121.70, 121.56, 118.60, 56.42, 20.49.


*2-Bromo-5-hydroxy-4-methoxybenzoic acid (*
**115**
*)*


The compound was synthetized following the literature procedure [28]. A solution of **114** (0.92 g, 3.18 mmol) in NaOH aq. solution (5%, 20.0 mL) was heated to reflux for 1 h. The reaction was then cooled down to 0 °C, acidified with 1M HCl, and extracted with EtOAc (3 × 30.0 mL). The organic phase was dried (Na_2_SO_4_), filtered, and concentrated under reduced pressure to afford 0.71 g (90%) of **115** as a white solid. Mp = 203–205 °C; ^1^H NMR (400 MHz, DMSO-d_6_) δ 12.72 (s, 1H), 9.64 (s, 1H), 7.28 (s, 1H), 7.15 (s, 1H), 3.82 (s, 3H); ^13^C NMR (101 MHz, DMSO-d_6_) δ 166.82, 151.21, 145.98, 124.27, 118.20, 117.59, 110.77, 56.47.


*3,8-Dihydroxy-9-methoxy-6H-benzo[c]chromen-6-one (*
**120**
*)*


Following general procedure A, **115** (0.64 g, 2.59 mmol) was reacted to afford 0.31 g (46%) of **120** as a dark-brown solid. Mp = 341–343 °C; ^1^H NMR (400 MHz, D_2_O, Na salt) δ 6.63 (s, 1H), 6.49 (d, *J* = 8.1 Hz, 1H), 6.45 (s, 1H), 5.81 (d, *J* = 2.4 Hz, 1H), 5.68 (dd, *J* = 8.1, 2.5 Hz, 1H), 3.57 (s, 3H); ^13^C NMR (101 MHz, d_2_o) δ 180.59, 165.83, 164.41, 152.94, 149.83, 133.05, 131.04, 125.17, 119.97, 116.67, 115.52, 108.98, 105.53, 55.83; HRMS (ESI) m/z: 257.0456 [M − H]^−^, calculated for C_14_H_9_O_5_ 257.0455.


**9-Amino-3,8-dihydroxy-6H-benzo[c]chromen-6-one (124)**



*2-Bromo-5-methoxy-4-nitrobenzoic acid (*
**116**
*)*


To a suspension of 3-methoxy-4-nitrobenzoic acid (**108**) (2.00 g, 10.15 mmol) in concentrated H_2_SO_4_ (80 mL), NBS (1.896 g, 10.65 mmol) was added in 3 portions (every 15 min) at room temperature. The reaction mixture was stirred at room temperature for 2 days. Then, the reaction mixture was poured into ice/water (ca. 200 mL) and the yellow precipitate filtered off and dried by co-evaporation with toluene to afford 1.98 g (71%) of **116** as a yellow solid. Mp 195–197 °C; ^1^H NMR (400 MHz, DMSO-d_6_) δ 8.24 (s, 1H), 7.63 (s, 1H), 3.96 (s, 3H); ^13^C NMR (101 MHz, DMSO-d_6_) δ 166.35, 150.87, 140.34, 139.26, 129.08, 115.63, 108.73, 57.27.


*3-Hydroxy-8-methoxy-9-nitro-6H-benzo[c]chromen-6-one (*
**121**
*)*


Following general procedure A, **116** (1.00 g, 3.62 mmol) was reacted and triturated with MeOH to afford 0.67 g (64%) of **121** as a yellow solid. Mp = 285–287 °C; ^1^H NMR (400 MHz, acetone-d_6_) δ 9.30 (s, 1H), 8.61 (s, 1H), 8.15 (d, *J* = 8.7 Hz, 1H), 7.99 (s, 1H), 6.94 (dd, *J* = 8.7, 2.4 Hz, 1H), 6.83 (d, *J* = 2.4 Hz, 1H), 4.13 (s, 3H); ^13^C-NMR (101 MHz, acetone-d_6_) δ 160.81, 160.14, 153.24, 151.09, 129.98, 125.51, 123.85, 118.90, 114.54, 114.28, 110.92, 110.08, 104.18, 57.62.


*9-Amino-3-hydroxy-8-methoxy-6H-benzo[c]chromen-6-one (*
**123**
*)*


To a suspension of **121** (250 mg, 0.870 mmol) in EtOH (5 mL), SnCl_2_ (825 mg, 4.35 mmol) was added. The reaction mixture was stirred at 70 °C for 30 h. Then, the reaction mixture was poured into ice/water, and neutralized by addition of a saturated aqueous solution of NaHCO_3_. The mixture was extracted with EtOAc (6 × 100 mL), dried over Na_2_SO_4_, filtered, and concentrated under reduced pressure. Purification by flash column chromatography (CH_2_Cl_2_:MeOH 100:0 to 98:2) afforded 0.08 g (36%) of **123** as a white solid. Mp = 330–332 °C; ^1^H-NMR (400 MHz, DMSO-d_6_) δ 10.10 (br s, 1H), 7.74 (d, *J* = 8.8 Hz, 1H), 7.38 (s, 1H), 7.19 (s, 1H), 6.80 (dd, *J* = 8.8, 2.4 Hz, 1H), 6.68 (d, *J* = 2.4 Hz, 1H), 6.15 (br s, 2H), 3.89 (s, 3H); ^13^C-NMR (101 MHz, DMSO-d_6_) δ 160.50, 158.69, 151.79, 146.13, 145.73, 130.72, 123.51, 112.74, 109.72, 108.55, 106.47, 102.76, 102.20, 55.53.


*9-Amino-3,8-dihydroxy-6H-benzo[c]chromen-6-one (*
**124**
*)*


Following general procedure B, **123** (0.035 g, 0.136 mmol) was reacted and triturated with Et_2_O, to afford 0.021 g (64%) of **124** as a brown solid. Mp > 350 °C; ^1^H-NMR (400 MHz, DMSO-d_6_) δ 10.02 (br s, 1H), 10.00 (br s, 1H), 7.71 (d, *J* = 8.8 Hz, 1H), 7.34 (s, 1H), 7.15 (s, 1H), 6.77 (dd, *J* = 8.8, 2.4 Hz, 1H), 6.65 (d, *J* = 2.4 Hz, 1H), 5.92 (br s, 2H); ^13^C-NMR (101 MHz, DMSO-d_6_) δ 160.43, 158.29, 151.54, 145.32, 144.08, 129.51, 123.21, 112.60, 112.02, 110.05, 106.81, 102.73, 102.55; HRMS (ESI) calculated for C_13_H_11_NO_4_ [M + H]^+^ 244.0604, found 244.0614.


**
*N*
**
**-(3,8-Dihydroxy-6-oxo-6*H*-benzo[*c*]chromen-9-yl)acetamide (127)**


To a suspension of **124** (58 mg, 0.226 mmol) in 1,4-dioxane (4 mL), a saturated aqueous solution of NaHCO_3_ (2 mL) and acetic anhydride (5 mL) were added. The reaction mixture was stirred at 100 °C for 1 h. The reaction mixture was cooled down to 0 °C, water was added, and a white solid precipitated, which was collected by filtration and washed with water. The crude (triacetylated compound) was suspended in CH_2_Cl_2_ (0.85 mL) and BBr_3_ (1 M in CH_2_Cl_2_, 0.85 mL, 0.85 mmol) was added. The reaction mixture was stirred at room temperature for 48 h. Then, the reaction was quenched with MeOH and the solvent removed under reduced pressure. Purification by flash column chromatography (CH_2_Cl_2_:MeOH 10:0 to 9:1) afforded 0.022 g (41%) of **127** as a white solid. Mp > 350 °C; ^1^H-NMR (400 MHz, DMSO-d_6_) δ 10.84 (s, 1H), 10.15 (s, 1H), 9.55 (s, 1H), 8.89 (s, 1H), 7.77 (d, *J* = 8.8 Hz, 1H), 7.59 (s, 1H), 6.84 (dd, *J* = 8.8, 2.4 Hz, 1H), 6.72 (d, *J* = 2.4 Hz, 1H), 2.21 (s, 3H); ^13^C-NMR (101 MHz, DMSO-d_6_) δ 169.76, 160.16, 158.68, 151.35, 146.72, 134.48, 127.76, 123.36, 114.02, 113.13, 113.07, 111.70, 109.79, 102.95, 24.30; HRMS (ESI) calculated for C_15_H_12_NO_5_ [M + H]^+^ 286.0710, found 286.0700.


**8-Amino-3,9-dihydroxy-6H-benzo[c]chromen-6-one (126)**



*Methyl 2-bromo-4,5-dinitrobenzoate (*
**110**
*)*


To a solution of 2-bromo-4-nitrobenzoic acid methyl ester (**109**) (5.00 g, 19.23 mmol) in sulfuric acid (15.0 mL), nitric acid (8.0 mL) was added. The reaction was stirred at 50 °C for 1 h and then poured into ice-cold water. The solid obtained was recrystallized from MeOH to afford 4.95 g (84%) of **110** as a yellow solid. Mp = 98–100 °C; ^1^H NMR (400 MHz, DMSO-d_6_) δ 8.71 (s, 1H), 8.60 (s, 1H), 3.15 (s, 3H); ^13^C NMR (101 MHz, DMSO-d_6_) δ 163.90, 143.59, 140.18, 137.29, 130.99, 128.07, 127.27, 54.02.


*Methyl 2-bromo-4-methoxy-5-nitrobenzoate (*
**111**
*)*


To a solution of **110** (2.00 g, 6.55 mmol) in MeOH (30.0 mL) at 0 °C, a solution of KOH (0.72 g, 13.11 mmol) in MeOH (10.0 mL) was added. The reaction was stirred at room temperature for 1 h. The solvent was removed under reduced pressure, and the solid obtained was taken up with water and filtered. Recrystallization from MeOH afforded 1.62 g (85%) of **111** as a white solid. Mp = 137–139 °C; ^1^H NMR (400 MHz, DMSO-d_6_) δ 8.37 (s, 1H), 7.74 (s, 1H), 4.02 (s, 3H), 3.84 (s, 3H); ^13^C NMR (101 MHz, DMSO-d_6_) δ 164.28, 154.68, 137.96, 128.50, 127.94, 123.48, 120.79, 58.26, 53.22.


*2-Bromo-4-methoxy-5-nitrobenzoic acid (*
**117**
*)*


Following general procedure E, **111** (1.5 g, 5.17 mmol) was reacted to afford 1.05 g (74%) of **117** as a white solid. Mp = 175–177 °C; ^1^H NMR (400 MHz, DMSO-d_6_) δ 13.60 (s, 1H), 8.34 (s, 1H), 7.70 (s, 1H), 4.01 (s, 3H); ^13^C NMR (101 MHz, DMSO-d_6_) δ 165.38, 154.32, 137.96, 128.33, 127.86, 124.74, 120.70, 58.13.


*3-Hydroxy-9-methoxy-8-nitro-6H-benzo[c]chromen-6-one (*
**122**
*)*


Following general procedure A, **117** (1.05 g, 3.80 mmol) was reacted to afford 0.100 g (9%) of **122** as a yellow solid. Mp = 355–357 °C; ^1^H NMR (400 MHz, DMSO-d_6_) δ 10.64 (s, 1H), 8.57 (s, 1H), 8.34 (d, *J* = 8.9 Hz, 1H), 7.92 (s, 1H), 6.86 (dd, *J* = 8.8, 2.4 Hz, 1H), 6.74 (d, *J* = 2.4 Hz, 1H), 4.14 (s, 3H); ^13^C NMR (101 MHz, DMSO-d_6_) δ 162.04, 159.57, 157.37, 153.75, 141.19, 138.71, 128.24, 127.14, 113.85, 111.87, 108.76, 106.44, 103.48, 58.26, 40.59, 40.43, 40.22.


*8-Amino-3-hydroxy-9-methoxy-6H-benzo[c]chromen-6-one (*
**125**
*)*


Following general procedure H, **25 122** (0.08 g, 0.28 mmol) was reacted to afford to afford 0.070 g (98%) of **125** as a yellow solid. Mp = 340–342 °C; ^1^H NMR (400 MHz, DMF-*d_7_*) δ 10.46 (s, 1H), 8.30 (d, *J* = 8.7 Hz, 1H), 7.80 (s, 1H), 7.69 (s, 1H), 7.03 (dd, *J* = 8.7, 2.4 Hz, 1H), 6.95 (d, *J* = 2.3 Hz, 1H), 5.59 (s, 2H), 4.27 (s, 3H); ^13^C NMR (101 MHz, DMF-*d_7_*) δ 161.13, 158.79, 153.49, 151.79, 139.18, 126.81, 123.82, 113.21, 112.86, 111.44, 111.25, 103.02, 102.10, 56.10.


*8-Amino-3,9-dihydroxy-6H-benzo[c]chromen-6-one (*
**126**
*)*


Following general procedure B, **125** (0.070 g, 0.27 mmol) was reacted to afford 0.06 g (91%) of **126** as a white solid. Mp > 350 °C; ^1^H NMR (400 MHz, DMF-*d_7_*) δ 8.40 (s, 1H), 8.09 (d, *J* = 8.7 Hz, 1H), 8.00 (s, 1H), 7.10 (dd, *J* = 8.7, 2.4 Hz, 1H), 6.99 (d, *J* = 2.4 Hz, 1H); ^13^C-NMR (101 MHz, DMF-*d_7_*): *δ* 162.3, 162.2, 158.2, 154.6, 136.6, 127.1, 126.3, 124.7, 115.2, 113.4, 111.4, 108.6, 105.0; HRMS (ESI) m/z: 242.0461 [M − H]^−^, calculated for C_13_H_8_NO_4_ 242.0459.


***N*-(3,9-Dihydroxy-6-oxo-6*H*-benzo[*c*]chromen-8-yl)acetamide (128)**


To a solution of **126** (50 mg, 0.206 mmol) in 1,4-dioxane (0.5 mL), a saturated aqueous solution of NaHCO_3_ (0.5 mL) and acetic anhydride (0.2 mL, 2.06 mmol) were added. The reaction mixture was stirred at room temperature for 1 h. Then, the solvent was removed under reduced pressure and dried by co-evaporation with toluene. The crude (triacetylated compound) was suspended in CH_2_Cl_2_ (0.85 mL) and BBr_3_ (1 M in CH_2_Cl_2_, 0.85 mL, 0.85 mmol) was added. The reaction mixture was stirred at room temperature for 48 h. Then, the reaction was quenched with MeOH and the solvent removed under reduced pressure. Purification by flash column chromatography (CH_2_Cl_2_:MeOH 10:0 to 9:1) afforded 0.031 g (53%) of **128** as a pale-brown/beige solid. Mp > 350 °C; ^1^H-NMR (400 MHz, DMSO-d_6_) δ 10.27 (br s, 1H), 9.47 (s, 1H), 8.76 (s, 1H), 7.82 (d, *J* = 8.7 Hz, 1H), 7.49 (s, 1H), 6.83 (dd, *J* = 8.7, 2.4 Hz, 1H), 6.71 (d, *J* = 2.4 Hz, 1H), 2.15 (s, 3H); ^13^C-NMR (101 MHz, DMSO-d_6_) δ 169.2, 160.3, 159.4, 154.4, 152.0, 132.2, 127.5, 124.0, 121.8, 113.1, 110.4, 109.2, 105.9, 102.9, 23.9. HRMS (ESI) calculated for C_15_H_12_NO_5_ [M + H]^+^ 286.0710, found 286.0699.


**8,9-Diamino-3-hydroxy-6*H*-benzo[*c*]chromen-6-one (135)**



*2-Bromo-4,5-dinitrobenzoic acid (*
**129**
*)*


To a solution of 2-bromo-4-nitrobenzoic acid (5.00 g, 20.3 mmol) in H_2_SO_4_ (15.0 mL), HNO_3_ (8.0 mL) was added. The reaction mixture was stirred at 50 °C for 100 min and then poured into ice/water. The solid obtained was filtered, washed with water, and dried by co-evaporation with toluene (×3) to give 4.5 g (76%) of **129** as a pale-yellow solid. Mp = 170–172 °C; ^1^H-NMR (400 MHz, DMSO-d_6_) δ 8.69 (s, 1H), 8.55 (s, 1H); ^13^C-NMR (101 MHz, DMSO-d_6_) δ 164.65, 142.64, 140.00, 138.71, 130.52, 126.99, 126.16.


*Methyl 4-(benzylamino)-2-bromo-5-nitrobenzoate (*
**130**
*)*


To a solution of **129** (2.180 g, 7.49 mmol) in DMF (60 mL), benzyl amine (2.5 mL, 23.2 mmol) was added. The reaction mixture was stirred at room temperature for 16 h. Then, the solvent was removed under reduced pressure, and the resulting solid was triturated with Et_2_O and a mixture of Et_2_O:CH_2_Cl_2_ to afford a yellow solid. This compound was dissolved in MeOH (100 mL), and H_2_SO_4_ (1.5 mL) was added. The reaction mixture was stirred at reflux for 8 h. Then, the reaction mixture was cooled down to room temperature, a saturated aqueous solution of NaHCO_3_ (20 mL) was added, and the solvent was removed under reduced pressure. A saturated aqueous solution of NaHCO_3_ (100 mL) and EtOAc (100 mL) was added to the residue and the phases were separated, the aqueous layer was extracted with EtOAc (3 × 80 mL), and the combined organic layers were dried over Na_2_SO_4_, filtered, and concentrated under reduced pressure. Purification by flash column chromatography (pentane:EtOAc 100:0 to 0:100) afforded 2.15 g (46%) of **130** as a yellow solid. Mp = 187–189 °C; ^1^H-NMR (400 MHz, CDCl_3_) δ 8.85 (s, 1H), 8.52 (br s, 1H), 7.43–7.33 (m, 5H), 7.18 (s, 1H), 4.55 (d, *J* = 5.5 Hz, 2H), 3.90 (s, 3H); ^13^C-NMR (101 MHz, CDCl_3_) δ 164.33, 146.28, 135.95, 131.56, 131.19, 129.35, 128.43, 127.43, 119.86, 117.84, 52.54, 47.56.


*Methyl 5-(benzylamino)-4′-(benzyloxy)-4-nitro-[1,1′-biphenyl]-2-carboxylate (*
**132**
*)*


Following general procedure D, **130** (0.25 g, 0.68 mmol) and 4-(benzyloxy)phenylboronic acid (**131**) (0.23 g, 1.03 mmol) were reacted and purified by flash chromatography (pentane/EtOAc) to afford 0.32 g (90%) of **132** as a yellow solid. Mp = 193–195 °C; ^1^H-NMR (400 MHz, CDCl_3_) δ 8.86 (s, 1H), 8.63 (t, *J* = 5.5 Hz, 1H), 7.47–7.32 (m, 10H), 7.16–7.13 (m, 2H), 7.01–6.97 (m, 2H), 6.74 (s, 1H), 5.10 (s, 2H), 4.57 (d, *J* = 5.5 Hz, 2H), 3.70 (s, 3H); ^13^C-NMR (101 MHz, CDCl_3_) δ 166.59, 159.04, 150.61, 146.14, 136.88, 136.57, 132.69, 130.93, 129.42, 129.19, 128.76, 128.20, 128.16, 127.67, 127.42, 117.72, 116.39, 116.12, 114.47, 70.16, 52.09, 47.42.


*5-(Benzylamino)-4′-(benzyloxy)-4-nitro-[1,1′-biphenyl]-2-carboxylic acid (*
**133**
*)*


Following general procedure E, **132** (0.3 g, 0.64 mmol) was reacted to afford 0.15 g (52%) of **133** as a yellow solid. Mp = 216–218 °C; ^1^H-NMR (400 MHz, DMSO-d_6_) δ 12.59 (br s, 1H), 8.97 (t, *J* = 6.0 Hz, 1H), 8.57 (s, 1H), 7.48–7.45 (m, 2H), 7.42–7.38 (m, 2H), 7.37–7.32 (m, 5H), 7.29–7.25 (m, 1H), 7.11–7.09 (m, 2H), 7.01–6.98 (m, 2H), 6.72 (s, 1H), 5.11 (s, 2H), 4.72 (d, *J* = 6.0 Hz, 2H); ^13^C-NMR (101 MHz, DMSO-d_6_) δ 167.09, 158.42, 148.82, 145.62, 138.07, 136.96, 132.15, 129.62, 129.43, 129.30, 128.65, 128.47, 127.92, 127.78, 127.21, 126.99, 117.93, 116.68, 114.15, 69.28, 45.77.


*9-(Benzylamino)-3-(benzyloxy)-8-nitro-6H-benzo[c]chromen-6-one (*
**134**
*)*


Following general procedure G, **133** (0.14 g, 0.31 mmol) was reacted and purified by flash chromatography (CH_2_Cl_2_/MeOH) to afford 0.06 g (43%) of **134** as a yellow solid. Mp = 284–286 °C; ^1^H-NMR (400 MHz, DMSO-d_6_) δ 9.10 (t, *J* = 6.1 Hz, 1H), 8.84 (s, 1H), 8.13 (d, *J* = 8.6 Hz, 1H), 7.51–7.47 (m, 5H), 7.43–7.34 (m, 5H), 7.28–7.24 (m, 1H), 7.08–7.05 (m, 2H), 5.23 (s, 2H), 4.87 (d, *J* = 6.1 Hz, 2H); ^13^C-NMR (201 MHz, DMSO-d_6_) δ 161.65, 159.23, 153.29, 147.62, 139.53, 137.99, 136.25, 131.91, 130.50, 128.63, 128.51, 128.10, 127.92, 127.31, 127.27, 125.99, 113.19, 109.55, 107.36, 105.07, 102.58, 69.90, 45.79.


*8,9-Diamino-3-hydroxy-6H-benzo[c]chromen-6-one (*
**135**
*)*


Following general procedure H, **134** (0.03 g, 0.07 mmol) was reacted to afford 0.009 g (53%) of **135** as a pale brown solid. M.p. > 350 °C; ^1^H-NMR (400 MHz, DMSO-d_6_) δ 9.92 (s, 1H), 7.67 (d, *J* = 8.7 Hz, 1H), 7.23 (s, 1H), 7.09 (s, 1H), 6.75 (dd, *J* = 8.7, 2.4 Hz, 1H), 6.63 (d, *J* = 2.4 Hz, 1H), 5.82 (br s, 2H), 5.09 (br s, 2H); ^13^C-NMR (101 MHz, DMSO-d_6_) δ 160.68, 157.67, 151.15, 143.45, 135.40, 127.04, 122.68, 112.47, 111.75, 110.50, 107.97, 102.71, 102.69; HRMS (ESI) calculated for C_13_H_11_N_2_O_3_ [M + H]^+^ 243.0764, found 243.0762.


**
*N*
**
**,*N′*-(3-Hydroxy-6-oxo-6*H*-benzo[*c*]chromene-8,9-diyl)diacetamide (140)**



*Methyl 4,5-diamino-2-bromobenzoate (*
**136**
*)*


To a solution of **110** (1.81 g, 5.94 mmol) in acetic acid (120 mL), iron (3.98 g, 71.28 mmol) was added. The reaction mixture was stirred at 75 °C for 3 h. Then, the reaction was cooled down to room temperature and filtered to remove the iron residues (wash with water and EtOAc, ca. 100 mL of each). Then, the solvent was removed under reduced pressure, water (200 mL) and EtOAc (200 mL) were added, the layers were separated, and the aqueous layer was extracted with EtOAc (3 × 200 mL), the combined organic layers were dried over Na_2_SO_4_, filtered, and concentrated under reduced pressure. Purification by flash column chromatography (CH_2_Cl_2_:MeOH 99:1 to 98:2) afforded 0.85 g (58%) of **136** as a pale-brown solid. Mp = 111–112 °C; ^1^H-NMR (400 MHz, CDCl_3_) δ 7.33 (s, 1H), 6.93 (s, 1H), 3.86 (s, 3H), 3.79 (br s, 2H), 3.33 (br s, 2H); ^13^C-NMR (101 MHz, CDCl_3_) δ 166.22, 140.44, 132.49, 120.95, 120.77, 120.47, 113.82, 52.07.


*Methyl 4,5-diacetamido-2-bromobenzoate (*
**137**
*)*


To a suspension of **136** (1.77 g, 7.22 mmol) in 1,4-dioxane (50 mL), a saturated aqueous solution of NaHCO_3_ (50 mL) and acetyl chloride (2.6 mL, 36.1 mmol) were added. The reaction mixture was stirred at room temperature for 24 h. Then, more portions of NaHCO_3_ (2.0 g, 23.8 mmol) and acetyl chloride (2.6 mL, 36.1 mmol) were added 6 times over the following 6 days to achieve full conversion. The reaction mixture was poured into ice/water (200 mL) and the solid was filtered off, washed with water, and dried to give 1.13 g (48%) of **137** as a pale-orange solid. Mp = 251–252 °C; ^1^H-NMR (400 MHz, DMSO-d_6_) δ 9.60 (bs, 2H), 8.16 (s, 1H), 8.05 (s, 1H), 3.83 (s, 3H), 2.12 (s, 3H), 2.09 (s, 3H); ^13^C-NMR (101 MHz, DMSO-d_6_) δ 169.16, 169.03, 165.08, 134.74, 128.45, 128.04, 127.56, 126.11, 115.64, 52.45, 23.97, 23.72.


*Methyl 4,5-diacetamido-2′,4′-dimethoxy-[1,1′-biphenyl]-2-carboxylate (*
**139**
*)*


Following general procedure D, **137** (0.8 g, 2.43 mmol) 2,4-dimethoxybenzeneboronic acid (**138**) (0.88 g, 4.86 mmol) were reacted and purified by flash chromatography (CH_2_Cl_2_/MeOH) to afford 0.68 g (73%) of **139** as a pale-brown solid. Mp = 110–112 °C; ^1^H-NMR (400 MHz, CDCl_3_) δ 8.76 (s, 1H), 8.72 (s, 1H), 7.81 (s, 1H), 7.18 (s, 1H), 7.07 (d, *J* = 8.3 Hz, 1H), 6.53 (dd, *J* = 8.3, 2.3 Hz, 1H), 6.44 (d, *J* = 2.3 Hz, 1H), 3.84 (s, 3H), 3.71 (s, 3H), 3.65 (s, 3H), 2.03 (s, 3H), 2.01 (s, 3H); ^13^C-NMR (101 MHz, CDCl_3_) δ 170.86, 170.49, 167.55, 160.84, 156.99, 136.95, 133.22, 130.39, 129.02, 128.53, 127.80, 127.07, 122.15, 104.39, 98.30, 55.50, 55.20, 51.93, 23.65, 23.59.


*N,N′-(3-Hydroxy-6-oxo-6H-benzo[c]chromene-8,9-diyl)diacetamide (*
**140**
*)*


Following general procedure B, **139** (0.6 g, 1.55 mmol) was reacted and triturated with MeOH and CH_2_Cl_2_ to afford 0.43 g (84%) of **140** as a pale-gray solid. Mp 340–342 °C; ^1^H-NMR (400 MHz, DMSO-d_6_) δ 10.31 (s, 1H), 9.67 (s, 1H), 9.58 (s, 1H), 8.54 (s, 1H), 8.38 (s, 1H), 7.89 (d, *J* = 8.7 Hz, 1H), 6.85 (dd, *J* = 8.7, 2.4 Hz, 1H), 6.74 (d, *J* = 2.4 Hz, 1H), 2.18 (s, 3H), 2.13 (s, 3H); ^13^C-NMR (101 MHz, DMSO-d_6_) δ 169.32, 169.12, 160.03, 159.61, 152.01, 137.32, 131.73, 129.23, 125.42, 124.21, 115.24, 114.66, 113.25, 109.12, 103.02, 24.16, 23.82; HRMS (ESI) calculated for C_17_H_15_N_2_O_5_ [M + H]^+^ 327.0975, found 327.0967.


**3-Hydroxy-9-methylbenzo[3,4]isoch**
**romeno[6,7-*d*]imidazol-6(10*H*)-one (141)**


A solution of **140** (0.04 mg, 0.011 mmol) in concentrated HCl (3 mL) was stirred at 105 °C in a sealed microwave vial for 4 h. The reaction mixture was cooled down to room temperature, diluted with ice/water, and the precipitate was filtered off to afford 0.015 g (50%) of **141** as a gray solid. Mp > 350 °C; ^1^H-NMR (400 MHz, CD_3_OD) δ 8.51 (d, *J* = 0.7 Hz, 1H), 8.31 (s, 1H), 8.09 (d, *J* = 8.8 Hz, 1H), 6.87 (dd, *J* = 8.8, 2.4 Hz, 1H), 6.72 (d, *J* = 2.4 Hz, 1H), 2.88 (s, 3H); ^13^C-NMR (101 MHz, CD_3_OD) δ 162.66, 161.68, 157.25, 153.27, 138.42, 134.48, 132.77, 125.74, 118.78, 117.08, 114.59, 110.78, 106.67, 104.23, 13.09; HRMS (ESI) calculated for C_15_H_11_N_2_O_3_ [M + H]^+^ 267.0764, found 267.0769.


**3-Hydroxy-6H-[1,3]diox**
**olo[4′,5′:4,5]benzo[1,2-c]chromen-6-one (144)**



*3-Hydroxy-6H-[1,3]dioxolo[4′,5′:4,5]benzo[1,2-c]chromen-6-one (*
**142**
*)*


A mixture of **84** (0.50 g, 2.02 mmol), diiodomethane (0.81 g, 3.03 mmol) and anhydrous K_2_CO_3_ (1.40 g, 10.12 mmol) in dry DMF (15 mL) was heated at reflux for 2 h. After cooling, the solvent was evaporated under reduced pressure to give the crude compound. Purification by flash chromatography (pentane/EtOAc) afforded 0.32 g (61%) of **142** as a white solid. Mp = 85–87 °C; ^1^H NMR (400 MHz, CDCl_3_) δ 7.33 (s, 1H), 7.09 (s, 1H), 6.05 (s, 2H), 3.89 (s, 3H); ^13^C NMR (101 MHz, CDCl_3_) δ 165.71, 150.95, 147.11, 129.53, 124.50, 114.38, 110.99, 102.48, 52.34.


*6-bromobenzo[d][1,3]diox*
*ole-5-carboxylic acid (*
**143**
*)*


Following general procedure E, **142** (0.32 g, 1.23 mmol) was reacted to afford 0.21 g (69%) of **143** as a white solid. Mp = 202–204 °C; ^1^H NMR (400 MHz, CDCl_3_) δ 7.50 (s, 1H), 7.14 (s, 1H), 6.08 (s, 2H); ^13^C NMR (101 MHz, CDCl_3_) δ 166.71, 150.42, 147.01, 126.00, 113,66 113.07, 112.91, 110.20, 102.77.


*3-Hydroxy-6H-[1,3]dioxolo[4′,5′:4,5]benzo[1,2-c]chromen-6-one (*
**144**
*)*


Following general procedure A, **143** (0.21 g, 0.87 mmol) was reacted to afford 0.16 g (73%) of **144** as a white solid. Mp = 331–333 °C; ^1^H NMR (400 MHz, DMSO-d_6_) δ 10.26 (s, 1H), 8.07 (d, *J* = 8.8 Hz, 1H), 7.80 (s, 1H), 7.49 (s, 1H), 6.79 (dd, *J* = 8.7, 2.4 Hz, 1H), 6.71 (d, *J* = 2.4 Hz, 1H), 6.22 (s, 2H); ^13^C NMR (101 MHz, DMSO-d_6_) δ 160.48, 159.77, 154.52, 152.01, 147.96, 133.28, 125.19, 113.58, 113.45, 110.15, 107.35, 103.09, 103.06, 101.31; HRMS (ESI) m/z: 255.0308 [M − H]^−^, calculated for C_14_H_7_O_5_ 255.0299.


**3-Hydroxy-8,10-dihydrobenzo[3,4]isochromeno**
**[6,7-*d*]imidazole-6,9-dione (150)**



*Methyl 6-bromo-2-oxo-2,3-dihydro-1H-benzo[d]imidazole-5-carboxylate (*
**145**
*)*


To a solution of **136** (0.23 g, 0.939 mmol) in anhydrous THF (3 mL), CDI (0.23 g, 1.41 mmol) was added. The reaction mixture was stirred under an atmosphere of nitrogen at room temperature for 16 h. Then, the solvent was removed under reduced pressure. Flash column chromatography (CH_2_Cl_2_:MeOH 97:3 to 90:10) afforded the desired product together with imidazole, and after trituration with MeOH to give 0.2 g (78%) of **145** obtained as a pale-pink solid. Mp = 287–289 °C; ^1^H-NMR (400 MHz, DMSO-d_6_) δ 11.05 (br s, 2H), 7.36 (s, 1H), 7.20 (s, 1H), 3.81 (s, 3H); ^13^C-NMR (101 MHz, DMSO-d_6_) δ 165.78, 155.19, 133.71, 129.00, 122.76, 113.27, 112.27, 110.79, 52.25.


*6-Bromo-2-oxo-2,3-dihydro-1H-benzo[d]imidazole-5-carboxylic acid (*
**149**
*)*


Following general procedure E, **145** (0.17 g, 0.65 mmol) was reacted to afford 0.16 g (98%) of **149** as a pale-pink solid. Mp = 360–361 °C; ^1^H-NMR (400 MHz, DMSO-d_6_) δ 12.97 (br s, 1H), 11.03 (br s, 1H), 10.94 (br s, 1H), 7.36 (s, 1H), 7.17 (s, 1H); ^13^C-NMR (101 MHz, DMSO-d_6_) δ 166.85, 155.21, 133.38, 128.94, 123.89, 113.22, 112.27, 110.86.


*3-Hydroxy-8,10-dihydrobenzo[3,4]isochromeno*
*[6,7-d]imidazole-6,9-dione (*
**150**
*)*


Following general procedure A, **149** (0.13 g, 0.51 mmol) was reacted and recrystallized from MeOH to afford 0.04 g (33%) of **150** as a brown solid. Mp > 350 °C; ^1^H-NMR (400 MHz, DMSO-d_6_) δ 11.38 (br s, 1H), 11.09 (br s, 1H), 10.18 (br s, 1H), 8.06 (d, *J* = 8.8 Hz, 1H), 7.63 (s, 1H), 7.61 (s, 1H), 6.81 (dd, *J* = 8.8, 2.4 Hz, 1H), 6.72 (d, *J* = 2.4 Hz, 1H); ^13^C-NMR (101 MHz, DMSO-d_6_) δ 160.88, 158.86, 155.53, 151.16, 136.75, 130.16, 130.03, 124.22, 112.96, 112.15, 110.00, 107.68, 102.76, 99.85; HRMS (ESI) calculated for C_14_H_7_N_2_O_4_ [M − H]^–^ 267.0411, found 267.0413.


**3-Hydroxybenzo[3,4]isochromeno**
**[6,7-*d*]imidazol-6(10*H*)-one (153)**



*Methyl 6-bromo-1H-benzo[d]imidazole-5-carboxylate (*
**146**
*)*


This compound was synthesized following a reported method for the synthesis of benzimidazoles [30]. To a solution of **136** (0.8 mg, 3.26 mmol) in DMF (10 mL), PhSiH_3_ (1.61 mL, 13.06 mmol) was added. The reaction mixture was stirred at 120 °C for 15 h. Then, the reaction was cooled down to room temperature and water (100 mL) was added (a white solid precipitates). The mixture was extracted with EtOAc (3 × 80 mL), the combined organic layers were dried over Na_2_SO_4_, filtered, and concentrated in vacuo. Purification by flash column chromatography (CH_2_Cl_2_:MeOH 100:0 to 95:5) afforded 0.27 g (32%) of **146** as a yellow solid. Mp 95–97 °C. ^1^H-NMR (400 MHz, DMSO-d_6_) δ 8.41 (s, 1H), 8.05 (s, 1H), 7.94 (s, 1H), 3.86 (s, 3H); ^13^C-NMR (101 MHz, DMSO-d_6_) δ 166.65, 52.42; only peaks of ester were visible (CO and OMe), but not the heterocycle.


*Methyl 6-(2,4-dimethoxyphenyl)-1H-benzo[d]imidazole-5-carboxylate (*
**151**
*)*


Following general procedure D, **146** (0.1 g, 0.39 mmol) and **138** (0.14 g, 0.78 mmol) were reacted and purified by flash chromatography (CH_2_Cl_2_/MeOH) to afford 0.07 g (56%) of **151** as a yellow solid. Mp = 110–112 °C. ^1^H-NMR (400 MHz, DMSO-d_6_) δ 12.68 (br s, 1H), 8.35 (s, 1H), 7.97 (s, 1H), 7.39 (s, 1H), 7.16 (d, *J* = 8.2 Hz, 1H), 6.59 (dd, *J* = 8.2, 2.4 Hz, 1H), 6.55 (d, *J* = 2.4 Hz, 1H), 3.81 (s, 3H), 3.61 (s, 3H), 3.60 (s, 3H); ^13^C-NMR (201 MHz, DMSO-d_6_) δ 168.24, 159.80, 156.86, 144.09, 130.30, 125.77, 123.52, 104.62, 98.01, 55.18, 55.01, 51.41 (5 carbons of the benzimidazole ring do not appear, probably because of tautomerism).


*3-Hydroxybenzo[3,4]isochromeno*
*[6,7-d]imidazol-6(10H)-one (*
**153**
*)*


Following general procedure B, **151** (0.05 g, 0.16 mmol) was reacted and triturated with Et_2_O to afford 0.011 g (37%) of **153** as a pale-yellow solid. Mp > 350 °C; ^1^H-NMR (400 MHz, DMSO-d_6_) δ 10.30 (br s, 1H), 9.14 (br s, 1H), 8.53 (s, 1H), 8.49 (s, 1H), 8.28 (d, *J* = 8.9 Hz, 1H), 6.87 (dd, *J* = 8.9, 2.4 Hz, 1H), 6.77 (d, *J* = 2.4 Hz, 1H); ^13^C-NMR (201 MHz, DMSO-d_6_) δ 160.61, 159.77, 151.35, 145.55, 137.71, 132.35, 131.77, 125.11, 117.05, 116.66, 113.29, 109.36, 106.75, 102.98; HRMS (ESI) calculated for C_14_H_9_N_2_O_3_ [M + H]^+^ 253.0608, found 253.0608.


**3-Hydroxybenzo[3,4]isochromeno[6,7-d][1,2,3]triazol**
**-6(10*H*)-one (154)**



*Methyl 6-bromo-1H-benzo[d][1,2,3]*
*triazole-5-carboxylate (*
**147**
*)*


To a solution of **136** (0.72 g, 2.96 mmol) in MeOH (25 mL) at 0 °C, NaNO_2_ (0.22 mg, 3.25 mmol) and an aqueous solution of HCl (1 M, 22 mL) were added. The reaction mixture was stirred from 0 °C to room temperature for 15 h. Then, MeOH was removed under reduced pressure, a saturated aqueous solution of NaHCO_3_ (50 mL) was added and the mixture was extracted with EtOAc (3 × 60 mL), and the combined organic layers were dried over Na_2_SO_4_, filtered, and concentrated under reduced pressure, to afford 0.73 g (96%) of **147** as a brown-orange solid. Mp = 174–175 °C; ^1^H-NMR (400 MHz, DMSO-d_6_) δ 8.39 (d, *J* = 0.6 Hz, 1H), 8.33 (d, *J* = 0.6 Hz, 1H), 3.90 (s, 3H); ^13^C-NMR (101 MHz, DMSO-d_6_) δ 166.27, 139.22, 138.65, 129.29, 119.18, 118.74, 116.06, 52.80.


*Methyl 1-acetyl-6-bromo-1H-benzo[d][1,2,3]triazole-5-carboxylate + Methyl 1-acetyl-5-bromo-1H-benzo[d][1,2,3]triazole-6-carb*
*oxylate (*
**148**
*)*


To a suspension of **147** (0.6 mg, 2.34 mmol) in dry CH_2_Cl_2_ (20 mL) at 0 °C, Et_3_N (0.52 mL, 3.75 mmol) and AcCl (0.23 mL, 3.28 mmol) were added. The reaction mixture was stirred at room temperature for 3 h. Then, an aqueous solution of HCl (1 M, 10 mL) was added, the mixture was diluted with water (20 mL) and CH_2_Cl_2_ (40 mL), and the layers were separated. The aqueous layer was extracted with CH_2_Cl_2_ (2 × 40 mL), and the combined organic layers were dried over Na_2_SO_4_, filtered, and concentrated under reduced pressure to give 0.7 g (99%) of 1**48** as a pale-brown solid. The product was obtained as a mixture of regioisomers (major: minor 6:4). ^1^H-NMR (400 MHz, CDCl_3_) δ 8.66 (d, *J* = 0.6 Hz, 1H major), 8.64 (d, *J* = 0.6 Hz, 1H minor), 8.55 (d, *J* = 0.6 Hz, 1H major), 8.44 (d, *J* = 0.6 Hz, 1H minor), 4.01 (s, 6H, major and minor overlap), 3.02 (s, 3H, minor), 3.01 (s, 3H, major); ^13^C-NMR (101 MHz, CDCl_3_) δ 169.22, 169.19, 166.16, 165.80, 147.62, 144.80, 134.63, 132.68, 130.61, 129.70, 125.35, 123.74, 123.16, 119.94, 117.65, 116.74, 53.23, 53.12, 23.27, 23.22 (both major and minor isomers reported).


*Methyl 6-(2,4-dimethoxyphenyl)-1H-benzo[d][1,2,3]triazole*
*-5-carboxylate (*
**152**
*)*


Following general procedure D, **148** (0.5 g, 1.68 mmol) and **138** (0.61 g, 3.35 mmol) were reacted and purified by flash chromatography (CH_2_Cl_2_/MeOH) to afford 0.27 g (51%) of **152** as a white solid. Mp = 98–100 °C; ^1^H-NMR (400 MHz, DMSO-d_6_) δ 8.29 (br s, 1H), 7.70 (br s, 1H), 7.26 (d, *J* = 8.3 Hz, 1H), 6.63 (dd, *J* = 8.3, 2.4 Hz, 1H), 6.57 (d, *J* = 2.4 Hz, 1H), 3.82 (s, 3H), 3.65 (s, 3H), 3.63 (s, 3H).;^13^C-NMR (201 MHz, DMSO-d_6_) δ 167.62, 160.39, 156.76, 130.51, 122.17, 104.93, 98.02, 55.26, 55.05, 51.79 (the 6 carbons of the benzotriazole ring do not appear, probably because of tautomerism).


*3-Hydroxybenzo[3,4]isochromeno[6,7-d][1,2,3]triazol*
*-6(10H)-one (*
**154**
*)*


Following general procedure B, **152** (0.04 g, 0.121 mmol) was reacted and triturated with Et_2_O to afford 0.011 g (37%) of **154** as a pale-yellow solid. Mp > 350 °C; ^1^H-NMR (400 MHz, DMSO-d_6_) δ 10.35 (br s, 1H), 8.86 (br s, 1H), 8.64 (br s, 1H), 8.34 (d, *J* = 8.8 Hz, 1H), 6.86 (dd, *J* = 8.8, 2.4 Hz, 1H), 6.76 (d, *J* = 2.4 Hz, 1H); ^13^C-NMR (201 MHz, DMSO-d_6_) δ 160.80, 159.84, 151.39, 125.44, 113.24, 109.51, 102.99 (the 6 carbons of the benzotriazole ring do not appear, probably because of tautomerism); HRMS (ESI) calculated for C_13_H_8_N_3_O_3_ [M + H]^+^ 254.0560, found 254.0563.


**2,3,8-Trihydroxy-6H-benzo[c]chromen-6-one (161)**



*Methyl 2-bromo-5-methoxybenzoate (*
**156**
*)*


Following general procedure C, 2-bromo-5-methoxynenzoic acid (**155**) (1.50 g, 6.49 mmol) was reacted to afford 1.57 g (99%) of **156** as a yellow oil. ^1^H NMR (400 MHz, CDCl_3_) δ 7.52 (d, *J* = 8.8 Hz, 1H), 7.31 (d, *J* = 3.1 Hz, 1H), 6.89 (dd, *J* = 8.8, 3.1 Hz, 1H), 3.93 (d, *J* = 0.3 Hz, 3H), 3.81 (s, 3H); ^13^C NMR (101 MHz, CDCl_3_) δ 166.47, 158.51, 135.02, 132.68, 119.04, 116.20, 111.93, 55.64, 52.51.


*Methyl 3′,4,4′-trimethoxy-[1,1′-biphenyl]-2-carboxylate (*
**158**
*)*


Following general procedure B, **156** (1.00 g, 4.08 mmol) and 3,4-methoxyphenylboronic acid (**157**) (0.74 g, 4.08 mmol), were reacted to afford 1.04 g (84%) of **158** as a yellow oil. ^1^H NMR (400 MHz, CDCl_3_) δ 7.33–7.24 (m, 2H), 7.05 (dd, *J* = 8.5, 2.8 Hz, 1H), 6.92–6.78 (m, 3H), 3.91 (s, 3H), 3.87 (s, 3H), 3.86 (s, 3H), 3.66 (s, 3H); ^13^C NMR (101 MHz, CDCl_3_) δ 169.33, 158.40, 148.46, 148.10, 134.41, 133.68, 131.90, 131.76, 120.60, 117.38, 114.12, 111.79, 110.79, 55.85, 55.56, 52.11.


*3′,4,4′-Trimethoxy-[1,1′-biphenyl]-2-carboxylic acid (*
**159**
*)*


Following general procedure E, **158** (1.04 g, 3.44 mmol) was reacted to afford 0.91 g (92%) of **159** as a white solid. Mp = 166–168 °C; ^1^H NMR (400 MHz, CDCl_3_) δ 11.67 (s, 1H), 7.43 (d, *J* = 2.7 Hz, 1H), 7.28 (d, *J* = 8.6 Hz, 1H), 7.07 (dd, *J* = 8.5, 2.8 Hz, 1H), 6.85 (dd, *J* = 3.5, 1.5 Hz, 3H), 3.88 (s, 3H), 3.85 (s, 3H), 3.84 (s, 3H); ^13^C NMR (101 MHz, CDCl_3_) δ 173.77, 158.36, 148.36, 148.23, 135.42, 133.42, 132.34, 130.21, 120.89, 118.35, 114.92, 112.18, 110.86, 55.82, 55.81, 55.55.


*2,3,8-Trimethoxy-6H-benzo[c]chromen-6-one (*
**160**
*)*


Following general procedure F, **159** (0.91 g, 3.15 mmol) was reacted to afford 0.45 g (50%) of **160** as a white powder. Mp = 201–203 °C; ^1^H NMR (400 MHz, CDCl_3_) δ 7.87 (d, *J* = 8.8 Hz, 1H), 7.76 (d, *J* = 2.7 Hz, 1H), 7.36 (dd, *J* = 8.8, 2.8 Hz, 1H), 7.31 (s, 1H), 6.84 (d, *J* = 1.2 Hz, 1H), 3.99 (s, 3H), 3.93 (s, 3H), 3.92 (s, 3H); ^13^C NMR (101 MHz, CDCl_3_) δ 161.65, 159.14, 150.53, 146.47, 145.28, 128.56, 124.35, 122.72, 121.23, 111.02, 110.14, 103.51, 100.74, 56.42, 56.20, 55.74.


*2,3,8-Trihydroxy-6H-benzo[c]chromen-6-one (*
**161**
*)*


Following general procedure B, **160** (0.30 g, 1.05 mmol) was reacted to afford 0.24 g (94%) of **161** as yellow solid. Mp > 350 °C; ^1^H NMR (400 MHz, DMSO-d_6_) δ 10.15 (s, 1H), 9.78 (s, 1H), 9.15 (s, 1H), 7.92 (d, *J* = 8.9 Hz, 1H), 7.49 (d, *J* = 2.7 Hz, 1H), 7.41 (s, 1H), 7.29 (dd, *J* = 8.7, 2.7 Hz, 1H), 6.73 (s, 1H); ^13^C NMR (101 MHz, DMSO-d_6_) δ 161.23, 157.25, 147.64, 143.87, 143.57, 127.41, 124.49, 123.90, 120.81, 114.02, 109.70, 108.03, 103.89; HRMS (ESI) m/z: 243.0306 [M − H]^−^, calculated for C_13_H_7_O_5_ 243.0299.


**3,8,9-Trihydroxyphenanthridin-6(5*H*)-one (165)**



*2-Bromo-4,5-dimethoxybenzamide (*
**162**
*)*


A solution of **8** (4.00 g, 15.3 mmol) in SOCl_2_ (35 mL) was stirred at reflux for 4 h. The reaction mixture was cooled down to room temperature and evaporated under reduced pressure to afford the corresponding acyl chloride as a pale-yellow solid. This solid was dissolved in THF (80 mL) and added dropwise to a solution of NH_3_ in THF (0.4 M, 115 mL) at 0 °C, and the reaction mixture was stirred from 0 °C to room temperature for 20 h. Then, the solvent was removed under reduced pressure. Purification by flash column chromatography (CH_2_Cl_2_:MeOH 98.5:1.5) afforded 3.03 g (76%) of **162** as a white solid. Mp = 177–179 °C; ^1^H-NMR (400 MHz, CDCl_3_) δ 7.35 (s, 1H), 7.01 (s, 1H), 6.50 (br s, 1H), 6.18 (br s, 1H), 3.90 (s, 3H), 3.90 (s, 3H); ^13^C-NMR (101 MHz, CDCl_3_) δ 168.55, 151.32, 148.51, 127.72, 116.05, 113.53, 110.39, 110.13, 56.43, 56.28.


*4,4′,5-Trimethoxy-[1,1′-biphenyl]-2-carboxamide (*
**163**
*)*


Following general procedure B, **162** (1.00 g, 3.85 mmol) and **86** (0.64 g, 4.23 mmol), were reacted and purified by flash chromatography (CH_2_Cl_2_/MeOH) to afford 0.61 g (55%) of **163** as a white solid. Mp = 167–168 °C; ^1^H-NMR (400 MHz, CDCl_3_) δ 7.45 (s, 1H), 7.36–7.32 (m, 2H), 6.98–6.95 (m, 2H), 6.74 (s, 1H), 5.50 (br s, 1H), 5.20 (br s, 1H), 3.96 (s, 3H), 3.91 (s, 3H), 3.85 (s, 3H); ^13^C-NMR (101 MHz, CDCl_3_) δ 170.45, 159.61, 150.63, 148.22, 133.51, 132.68, 130.34, 125.73, 114.37, 113.23, 112.56, 56.26, 56.20, 55.50.


*3,8,9-Trimethoxyphenanthridin-6(5H)-one (*
**164**
*)*


To a solution of **163** (0.3 g, 1.04 mmol) in *o*-xylene (30 mL), CuI (0.07 g, 0.366 mmol), PPh_3_ (0.19 g, 0.731 mmol), and KO*t*Bu (0.82 mg, 7.31 mmol) were added in 3 portions over 3 days, the mixture was stirred at 120 °C for 4 days in total. Then, water (200 mL) was added, and the mixture was extracted with CH_2_Cl_2_ (3 × 100 mL). The combined organic layers were washed with brine (100 mL), dried over Na_2_SO_4_, filtered, and concentrated under reduced pressure. Recrystallization from MeOH afforded 0.05 g (18%) of **164** as a pale-yellow solid. Mp = 275–277 °C. ^1^H-NMR (400 MHz, DMSO-d_6_) δ 11.47 (s, 1H), 8.28 (d, *J* = 8.7 Hz, 1H), 7.76 (s, 1H), 7.65 (s, 1H), 6.87–6.83 (m, 2H), 4.00 (s, 3H), 3.88 (s, 3H), 3.81 (s, 3H); ^13^C-NMR (101 MHz, DMSO-d_6_) δ 160.73, 159.58, 153.33, 148.56, 137.47, 129.43, 124.61, 117.85, 111.28, 109.87, 107.77, 103.56, 99.24, 56.04, 55.48, 55.25.


*3,8,9-Trihydroxyphenanthridin-6(5H)-one (*
**165**
*)*


Following general procedure B, **164** (0.024 g, 0.0084 mmol) was reacted to afford and purified by reversed phase (C18) flash column chromatography (H_2_O:CH_3_CN 10:0 to 7:3) to afford 0.02 g (88%) of **165** as a red solid. Mp > 350 °C; ^1^H-NMR (400 MHz, CD_3_OD) δ 7.92 (d, *J* = 8.7 Hz, 1H), 7.67 (s, 1H), 7.56 (s, 1H), 6.77–6.73 (m, 2H); ^13^C-NMR (176 MHz, CD_3_OD) δ ^13^C NMR (176 MHz, MeOD) δ 164.17, 159.21, 152.96, 146.86, 137.96, 131.48, 124.77, 118.09, 113.04, 112.85, 112.84, 107.43, 102.47; HRMS (ESI) m/z: 244.0610 [M + H]^+^, calculated for C_13_H_9_NO_4_ 244.0612.


**3,8,9-Trihydroxy-5-methylphenanthridin-6(5*H*)-one (167)**



*3,8,9-Trimethoxy-5-methylphenanthridin-6(5H)-one (*
**166**
*)*


To a suspension of **164** (0.016 g, 0.056 mmol) and Cs_2_CO_3_ (0.091 g, 0.28 mmol) in THF (6 mL) under an atmosphere of nitrogen, methyl iodide (11 µL, 0.17 mmol) was added. The mixture was stirred at reflux (sealed microwave vial, pre-heated heating block at 86 °C) for 5 h. Then, the solvent was removed under reduced pressure. Water (30 mL) and CH_2_Cl_2_ (30 mL) were added, the layers were separated, and the aqueous layer was extracted with CH_2_Cl_2_ (2 × 20 mL). The combined organic layers were washed with brine (30 mL), dried over Na_2_SO_4_, filtered, and concentrated under reduced pressure. The crude was triturated with Et_2_O to afford 0.014 (83%) **166** as a yellow solid. Mp = 179–180 °C; ^1^H-NMR (400 MHz, CDCl_3_) δ 8.02 (d, *J* = 8.6 Hz, 1H), 7.87 (s, 1H), 7.45 (s, 1H), 6.88 (dd, *J* = 8.7, 2.4 Hz, 1H), 6.85 (d, *J* = 2.4 Hz, 1H), 4.07 (s, 3H), 4.02 (s, 3H), 3.93 (s, 3H), 3.77 (s, 3H); ^13^C-NMR (101 MHz, CDCl_3_) δ 161.66, 160.26, 153.44, 149.14, 139.10, 128.72, 124.05, 118.32, 112.94, 109.17, 108.93, 102.14, 100.28, 56.31, 56.20, 55.68, 30.12.


*3,8,9-Trihydroxy-5-methylphenanthridin-6(5H)-one (*
**167**
*)*


Following general procedure B, **166** (0.01 g, 0.033 mmol) was reacted and triturated with MeOH to afford 0.003 g (35%) of **167** as yellow solid. ^1^H-NMR (400 MHz, CD_3_OD) δ 8.00 (d, *J* = 8.8 Hz, 1H), 7.69 (s, 1H), 7.55 (s, 1H), 6.91 (d, *J* = 2.3 Hz, 1H), 6.81 (dd, *J* = 8.8, 2.3 Hz, 1H), 3.72 (s, 3H); ^13^C-NMR (176 MHz, CD_3_OD) δ 163.53, 159.48, 152.77, 146.98, 139.62, 130.30, 125.21, 117.87, 113.61, 113.39, 112.32, 107.09, 102.38, 30.38; HRMS (ESI) m/z: 258.0766 [M + H]^+^, calculated for C_14_H_12_NO_4_ 258.0765.


**Phenanthrene-2,3,7-triol (178)**



*4,4′,5-Trimethoxy-[1,1′-biphenyl]-2-carbaldehyde (*
**170**
*)*


Following general procedure D, **68** (2.0 g, 8.16 mmol) and **86** (1.275 g, 8.39 mmol) were reacted and purified by flash chromatography (pentane/EtOAc) to afford 1.81 g (81%) of **170** as a white solid. Mp = 106–107 °C; ^1^H NMR (400 MHz, CDCl_3_) δ 9.83 (s, 1H), 7.52 (s, 1H), 7.32–7.28 (m, 2H), 7.02–6.97 (m, 2H), 6.83 (s, 1H), 3.98 (s, 3H), 3.97 (s, 3H), 3.87 (s, 3H); ^13^C NMR (101 MHz, CDCl_3_) δ 191.36, 159.67, 153.48, 148.62, 141.33, 131.42, 129.93, 127.02, 113.90, 112.69, 108.67, 56.28, 56.19, 55.49.


*(E/Z)-4,4′,5-Trimethoxy-2-(2-methoxyvinyl)-1,1′-biphenyl (*
**172**
*)*


To a solution of methoxymethyltriphenylphosphonium chloride (2.57 g, 7.5 mmol) in THF (25 mL), a solution of KOtBu (842 mg, 7.5 mmol) in THF (15 mL) was added at 0 °C. After stirring for 30 min, a solution of **170** (0.91 mg, 5.0 mmol) in THF (10 mL) was added and stirred at room temperature for 2 h. The reaction mixture was quenched with water (40 mL) and extracted with EtOAc (3 × 20 mL). The combined organic layer was dried over MgSO_4_, and the organic solvent was removed under reduced pressure. The crude was purified by flash chromatography (pentane/EtOAc 90/10) to give 0.5 g (90%) of **172** as a colorless oil (mixture of E/Z isomer). ^1^H NMR (400 MHz, CDCl_3_) δ 7.32–7.25 (m, 2H), 6.97–6.92 (m, 2H), 6.88 (d, *J* = 5.5 Hz, 1H), 6.76 (d, *J* = 9.6 Hz, 1H), 5.76 (d, *J* = 12.9 Hz, 1H), 3.93 (d, *J* = 1.6 Hz, 3H), 3.87 (d, *J* = 1.9 Hz, 3H), 3.84 (d, *J* = 2.9 Hz, 3H), 3.75 (s, 1H), 3.55 (s, 2H); ^13^C NMR (101 MHz, CDCl_3_) δ 158.48, 158.45, 148.14, 147.79, 147.54, 147.37, 146.84, 146.40, 133.96, 133.63, 133.00, 132.15, 130.95, 130.80, 126.47, 125.97, 113.47, 113.40, 112.93, 112.21, 108.15, 104.51, 103.81, 60.54, 56.43, 55.98, 55.97, 55.86, 55.84, 55.27.


*2,3,7-Trimethoxyphenanthrene (*
**175**
*)*


To a solution of **172** (0.41 g, 1.3 mmol) in CH_2_Cl_2_ (10 mL) CF_3_SO_3_H (0.1 mL) was added under argon at 0 °C, and the solution was stirred overnight at room temperature. The organic layers were evaporated to dryness. The residue was purified by crystallization from MeOH to give 0.25 g (68%) of **175** as a grey solid. Mp = 157–158 °C; ^1^H NMR (400 MHz, CDCl_3_) δ 8.43 (d, *J* = 8.9 Hz, 1H), 7.91 (s, 1H), 7.67–7.54 (m, 2H), 7.29–7.23 (m, 2H), 4.10 (s, 3H), 4.03 (s, 3H), 3.96 (s, 3H); ^13^C NMR (101 MHz, CDCl_3_) δ 157.52, 149.44, 148.59, 132.61, 126.54, 125.95, 125.08, 124.68, 124.22, 123.73, 117.03, 108.32, 108.30, 102.77, 55.96, 55.89, 55.38.


*Phenanthrene-2,3,7-triol (*
**178**
*)*


Following general procedure B, **175** (0.08 g, 0.30 mmol) was reacted and triturated with MeOH to afford 0.05 g (77%) of **178** as a grey solid. Mp = 266–268 °C; ^1^H NMR (400 MHz, DMSO-d_6_) δ 9.84–9.06 (m, 3H), 8.23 (d, *J* = 8.9 Hz, 1H), 7.83 (s, 1H), 7.46 (d, *J* = 8.8 Hz, 1H), 7.35 (d, *J* = 8.8 Hz, 1H), 7.13 (s, 1H), 7.11–7.04 (m, 2H); ^13^C NMR (101 MHz, DMSO-d_6_) δ 155.33, 147.17, 145.76, 132.57, 126.58, 125.23, 124.85, 123.97, 123.42, 123.08, 117.25, 112.43, 111.40, 106.87; HRMS (ESI) calculated for C_14_H_9_O_3_ [M − H]^−^ 225.0557, found 225.0571.


**Phenanthridine-3,8,9-triol (179)**



*(E)-4,4′,5-trimethoxy-[1,1′-biphenyl]-2-carbaldehyde O-acetyl oxime (*
**173**
*)*


To a solution of **170** (1.69 g, 6.20 mmol) in EtOH (16 mL), pyridine (1.41 mL, 17.38 mmol) and hydroxylamine hydrochloride (0.65 g, 9.31 mmol) were added. The resulting mixture was stirred at 60 °C for 75 min. Then, the reaction mixture was cooled to room temperature, and water (40 mL) and EtOAc (40 mL) were added. The layers were separated, and the aqueous layer was extracted with EtOAc (3 × 40 mL). The combined organic layers were washed with aqueous HCl (1 M, 1 × 10 mL) and brine (1 × 10 mL), and dried over Na_2_SO_4_, filtered, and concentrated under reduced pressure. Then, the resulting crude was dissolved in pyridine (16 mL), and DMAP (ca 8 mg, 64 μmol) and acetic anhydride (1.17 mL, 12.41 mmol) were added. The reaction mixture was stirred at room temperature for 1.5 h. Then, water (40 mL) and EtOAc (40 mL) were added. The layers were separated, and the aqueous layer was extracted with EtOAc (3 × 40 mL). The combined organic layers were washed with aqueous HCl (1 M, 1 × 40 mL) and brine (1 × 40 mL), and dried over Na_2_SO_4_, filtered, and concentrated under reduced pressure to give 1.9 g (95%) of **173** as a pale-yellow solid. Mp = 132–133 °C; ^1^H NMR (400 MHz, CDCl_3_) δ 8.23 (s, 1H), 7.56 (s, 1H), 7.22–7.18 (m, 2H), 6.99–6.95 (m, 2H), 6.79 (s, 1H), 3.96 (s, 3H), 3.91 (s, 3H), 3.86 (s, 3H), 2.15 (s, 3H); ^13^C NMR (101 MHz, CDCl_3_) δ 168.79, 159.35, 155.41, 151.58, 148.42, 137.83, 131.16, 130.99, 119.98, 113.90, 112.56, 108.50, 56.25, 56.01, 55.40, 19.66.


*3,8,9-Trimethoxyphenanthridine (*
**176**
*)*


This compound was synthesized following a reported method for the synthesis of phenanthridines [22]. To a solution of **173** (1.68 g, 5.11 mmol) in acetic acid (35 mL), Fe(acac)_3_ (0.36 g, 1.02 mmol) was added. The reaction mixture was stirred under an atmosphere of nitrogen at 80 °C for 2 h. The reaction mixture was then allowed to cool down to room temperature, neutralized with a saturated aqueous solution of NaHCO_3_, and extracted with EtOAc (3 × 5 mL). The combined organic layers were dried over Na_2_SO_4_, filtered, and concentrated under reduced pressure. Purification by flash column chromatography (CH_2_Cl_2_/MeOH 95/5) afforded 0.42 g (31%) of the desired product **176** as a yellow solid. Mp = 167–168 °C; ^1^H NMR (400 MHz, CDCl_3_) δ 9.11 (s, 1H), 8.33 (d, *J* = 9.0 Hz, 1H), 7.78 (s, 1H), 7.55 (d, *J* = 2.7 Hz, 1H), 7.33 (s, 1H), 7.28 (dd, *J* = 9.0, 2.6 Hz, 1H), 4.13 (s, 3H), 4.06 (s, 3H), 3.98 (s, 3H); ^13^C NMR (101 MHz, CDCl_3_) δ 159.57, 153.22, 152.24, 149.39, 145.75, 128.72, 123.03, 121.04, 118.15, 118.06, 109.82, 107.85, 101.43, 56.30, 56.23, 55.67.


*Phenanthridine-3,8,9-triol (*
**179**
*)*


To a solution of **176** (0.1 g, 0.371 mmol) in acetic acid (1.1 mL), HBr (48% aqueous, 3.3 mL) was added. The reaction mixture was stirred at 120 °C for 6 days. Then, the solvent was removed under reduced pressure. Et_2_O was added and the precipitate formed was filtered off. Purification by flash chromatography (CH_2_Cl_2_/MeOH) afforded 0.01 g (11%) of **179** as an orange solid. Mp = 203–205 °C; ^1^H NMR (400 MHz, DMSO-d_6_) δ 8.91 (s, 1H), 8.23 (d, *J* = 8.9 Hz, 1H), 7.78 (s, 1H), 7.33 (s, 1H), 7.26 (s, 1H), 7.12 (d, *J* = 8.7 Hz, 1H); ^13^C NMR (176 MHz, DMSO-d_6_) δ 163.50, 161.57, 157.33, 154.45, 150.76, 123.78, 120.01, 117.70, 116.92, 112.06, 105.56, 41.80, 41.68; HRMS (ESI) calculated for C_13_H_10_NO_3_ [M + H]^+^ 228.0655, found 228.0658.


**Phenanthridine-2,3,8-triol (180)**



*Bromo-5-methoxybenzaldehyde (*
**169**
*)*


To a solution of KBr (1.19 g, 10.0 mmol) in water (100 mL), *m*-anisaldehyde (**168**) (1.22 mL, 10.0 mmol) was added. Bromine (0.52 mL, 10.0 mmol) was added dropwise, and the reaction mixture was stirred at room temperature for 4 h. Then, the precipitate was filtered off, washed with water, and dried by co-evaporation with toluene (3 × 10 mL) to afford 1.91 g (89%) of **169** as a pale-orange solid. Mp = 63–65 °C; ^1^H-NMR (400 MHz, CDCl_3_) δ 10.31 (s, 1H), 7.52 (d, *J* = 8.8 Hz, 1H), 7.41 (d, *J* = 3.2 Hz, 1H), 7.03 (dd, *J* = 8.8, 3.2 Hz, 1H), 3.84 (s, 3H); ^13^C-NMR (101 MHz, CDCl_3_) δ 191.93, 159.40, 134.70, 134.10, 123.28, 118.12, 112.80, 55.88.


*3′,4,4′-Trimethoxy-[1,1′-biphenyl]-2-carbaldehyde (*
**171**
*)*


Following general procedure D, **169** (0.43 g, 2.00 mmol) and **157** (0,40 g, 2.20 mmol) were reacted and purified by flash chromatography (pentane/EtOAc) to afford 0.45 (83%) of **171** as a white solid. Mp = 103–104 °C; ^1^H-NMR (400 MHz, CDCl_3_) δ 9.96 (s, 1H), 7.49 (d, *J* = 2.8 Hz, 1H), 7.37 (d, *J* = 8.5 Hz, 1H), 7.19 (dd, *J* = 8.5, 2.8 Hz, 1H), 6.96–6.93 (m, 1H), 6.88–6.85 (m, 2H), 3.94 (s, 3H), 3.90 (s, 3H), 3.90 (s, 3H); ^13^C-NMR (101 MHz, CDCl_3_) δ 192.61, 159.09, 149.05, 148.93, 139.10, 134.72, 132.14, 130.25, 123.07, 121.54, 113.34, 111.08, 109.86, 56.14, 56.13, 55.76.


*(E)-3′,4,4′-trimethoxy-[1,1′-biphenyl]-2-carbaldehyde O-acetyl oxime (174)*


To a solution of **171** (0.42 g, 1.54 mmol) in EtOH (4 mL), pyridine (0.35 mL, 4.32 mmol) and hydroxylamine hydrochloride (0.16 g, 2.31 mmol) were added. The resulting mixture was stirred at 60 °C for 75 min. Then, the reaction mixture was cooled to room temperature, and water (10 mL) and EtOAc (10 mL) were added. The layers were separated, and the aqueous layer was extracted with EtOAc (3 × 10 mL). The combined organic layers were washed with aqueous HCl (1 M, 1 × 10 mL) and brine (1 × 10 mL), and dried over Na_2_SO_4_, filtered, and concentrated under reduced pressure. Then, the resulting crude (444 mg, white solid) was dissolved in pyridine (4 mL), and DMAP (ca 2 mg, 16 μmol) and acetic anhydride (0.30 mL, 3.08 mmol) were added. The reaction mixture was stirred at room temperature for 1.5 h. Then, water (10 mL) and EtOAc (10 mL) were added. The layers were separated, and the aqueous layer was extracted with EtOAc (3 × 10 mL). The combined organic layers were washed with aqueous HCl (1 M, 1 × 10 mL) and brine (1 × 10 mL), and dried over Na_2_SO_4_, filtered, and concentrated under reduced pressure to give 0.5 g (99%) of. **174** as a pale-yellow solid. Mp = 107–109 °C; ^1^H-NMR (400 MHz, CDCl_3_) δ 8.30 (s, 1H), 7.57 (d, *J* = 2.7 Hz, 1H), 7.30 (d, *J* = 8.6 Hz, 1H), 7.07 (dd, *J* = 8.6, 2.7 Hz, 1H), 6.94–6.92 (m, 1H), 6.81–6.78 (m, 2H), 3.93 (s, 3H), 3.89 (s, 3H), 3.88 (s, 3H), 2.18 (s, 3H); ^13^C-NMR (101 MHz, CDCl_3_) δ 168.83, 158.94, 155.90, 148.85, 148.77, 136.48, 131.52, 131.49, 128.79, 122.51, 119.01, 113.16, 111.13, 110.33, 56.12, 56.11, 55.82, 19.74.


*2,3,8-Trimethoxyphenanthridine (*
**177**
*)*


This compound was synthesized following a reported method for the synthesis of phenanthridines [22]. To a solution of **174** (0.47 g, 1.43 mmol) in acetic acid (9 mL), Fe(acac)_3_ (0.1 g, 0.286 mmol) was added. The reaction mixture was stirred under an atmosphere of nitrogen at 80 °C for 2 h. The reaction mixture was then allowed to cool down to room temperature, neutralized with a saturated aqueous solution of NaHCO_3_, and extracted with EtOAc (3 × 5 mL). The combined organic layers were dried over Na_2_SO_4_, filtered, and concentrated under reduced pressure. Purification by flash column chromatography (CH_2_Cl_2_/MeOH 95/5) afforded 0.11 g (49%) of the desired product **177** as a pale-orange sticky oil (undesired regioisomer was also recovered). ^1^H-NMR (400 MHz, CDCl_3_) δ 9.07 (s, 1H), 8.32 (d, *J* = 9.1 Hz, 1H), 7.71 (s, 1H), 7.53 (s, 1H), 7.41 (dd, *J* = 9.1, 2.6 Hz, 1H), 7.30 (d, *J* = 2.6 Hz, 1H), 4.08 (s, 3H), 4.04 (s, 3H), 3.96 (s, 3H), ^13^C-NMR (101 MHz, CDCl_3_) δ 158.14, 150.52, 150.28, 149.74, 139.70, 127.03, 126.84, 123.16, 122.12, 118.69, 109.96, 107.60, 101.35, 56.19, 56.15, 55.63.


*Phenanthridine-2,3,8-triol (*
**180**
*)*


To a solution of **177** (0.1 g, 0.371 mmol) in acetic acid (1.1 mL), HBr (48% aqueous, 3.3 mL) was added. The reaction mixture was stirred at 120 °C for 6 days. Then, the solvent was removed under reduced pressure. Et_2_O was added and the precipitate formed was filtered off. Purification by preparative HPLC (H_2_O (0.1% formic acid): CH_3_CN (0.1% formic acid) 9:1 to 6:4 over 25 min) afforded 0.037 G (37%) of **180** formate salt as a dark-yellow-orange solid. Mp = 210–211 °C; ^1^H-NMR (400 MHz, DMSO-d_6_) δ 9.97–9.72 (m, 3H), 8.91 (s, 1H), 8.29 (d, *J* = 8.8 Hz, 1H), 8.14 (s, 1H, formic acid), 7.79 (s, 1H), 7.35–7.30 (m, 3H). ^13^C-NMR (101 MHz, DMSO-d_6_) δ 163.11(formic acid), 155.58, 149.19, 146.98, 146.90, 138.38, 126.62, 124.72, 123.18, 121.54, 117.82, 113.08, 110.51, 105.26; HRMS (ESI) calculated for C_13_H_10_NO_3_ [M + H]^+^ 228.0655, found 228.0660.


**6,6-dimethyl-6H-benzo[c]chromene-3,8,9-triol (183)**



*3,8,9-tris(benzyloxy)-6H-benzo[c]chromen-6-one (*
**181**
*)*


Following general procedure K, **1** (0.50 g, 2.04 mmol) was reacted to afford 0.79 g of **181** (75%) as a brown solid. Mp = 173–175 °C; ^1^H NMR (400 MHz, CDCl_3_) δ 7.82 (s, 1H), 7.72 (d, *J* = 8.8 Hz, 1H), 7.60–7.28 (m, 15H), 6.95 (dd, *J* = 8.8, 2.5 Hz, 1H), 6.90 (d, *J* = 2.5 Hz, 1H), 5.35 (s, 2H), 5.25 (s, 2H), 5.12 (s, 2H); ^13^C NMR (101 MHz, CDCl_3_) δ 161.25, 159.90, 154.89, 152.17, 148.94, 136.25, 136.09, 135.99, 130.47, 128.75, 128.70, 128.58, 128.26, 128.07, 127.51, 127.33, 127.10, 123.12, 113.28, 113.01, 112.97, 111.43, 104.77, 102.61, 71.02, 70.92, 70.39.


*3,8,9-Tris(benzyloxy)-6,6-dimethyl-6H-benzo[c]chromene (*
**182**
*)*


The compound was synthetized following the literature procedure [31]. To a solution of methyl magnesium bromide in diethyl ether (3.0 M, 2.43 mL, 7.29 mmol) under nitrogen at 0 °C, a suspension of **181** (0.75 g, 1.46 mmol) in anhydrous THF (30.0 mL) was added. The resulting mixture was refluxed for 12 h and then cooled to room temperature. The solution was poured into concentrated HCl conc. (3.0 mL) and stirred for 5 min. Water (30.0 mL) was then added and the solution was extracted with EtOAc (3 × 30.0 mL). The organic phase was dried (Na_2_SO_4_), filtered, and concentrated under reduced pressure to give 0.7 g of crude product. Purification by flash chromatography (90/10 pemtane/EtOAc) afforded 0.31 g (40%) of **182** as a colorless oil. ^1^H NMR (400 MHz, CDCl_3_) δ 7.61–7.46 (m, 6H), 7.46–7.29 (m, 8H), 7.26 (d, *J* = 2.0 Hz, 1H), 6.83 (s, 1H), 6.69 (dd, *J* = 8.5, 2.5 Hz, 1H), 6.64 (d, *J* = 2.5 Hz, 1H), 5.25 (s, 2H), 5.21 (s, 2H), 5.09 (s, 2H), 1.60 (s, 6H); ^13^C NMR (101 MHz, CDCl_3_) δ 159.69, 153.54, 149.04, 148.12, 137.29, 137.26, 136.92, 131.85, 128.61, 128.60, 128.53, 128.02, 127.95, 127.63, 127.59, 127.46, 123.21, 122.86, 115.71, 111.66, 108.92, 108.69, 103.88, 77.73, 72.13, 71.64, 70.09, 27.65.


*6,6-dimethyl-6H-benzo[c]chromene-3,8,9-triol (*
**183**
*)*


Following general procedure H, **182** (0.16 g, 0.30 mmol) was reacted to afford 0.05 g (66%) of **183** as a yellow oil. ^1^H NMR (400 MHz, CD_3_OD) δ 7.38 (d, *J* = 8.5 Hz, 1H), 7.05 (s, 1H), 6.67 (s, 1H), 6.42 (dd, *J* = 8.4, 2.5 Hz, 1H), 6.30 (d, *J* = 2.4 Hz, 1H), 1.51 (s, 6H); ^13^C NMR (101 MHz, CD_3_OD) δ 157.43, 153-07, 144.50, 144.13, 130.26, 122.54, 121.03, 114.91, 110.13, 106.53, 107.95, 103.96, 77.10, 26.62; HRMS (ESI) m/z: 257.0826 [M − H]^−^, calculated for C_15_H_13_O_4_ 257.0819


**2,3,7-Trihydroxy-9H-fluoren-9-one (192)**



**Methyl 4,4′,5-trimethoxy-[1,1′-biphenyl]-2-carboxylate (186)**


Following general procedure D, **9** (0.55 g, 2.00 mmol) and **86** (0.33 g, 2.20 mmol), were reacted and purified by flash chromatography (pentane/EtOAc) to afford 0.54 g (89%) of **186** as a colorless oil. ^1^H NMR (400 MHz, CDCl_3_) δ 7.41 (s, 1H), 7.22 (d, *J* = 8.7 Hz, 2H), 6.93 (d, *J* = 8.7 Hz, 2H), 6.78 (s, 1H), 3.95 (s, 3H), 3.92 (s, 3H), 3.85 (s, 3H), 3.64 (s, 3H); ^13^C NMR (101 MHz, CDCl_3_) δ 168.31, 158.72, 151.13, 147.48, 137.02, 133.99, 129.54, 121.86, 113.63, 113.32, 112.81, 56.11, 56.03, 55.25, 51.80.


*2,3,7-Trimethoxy-9H-fluoren-9-one (*
**191**
*)*


The compound was synthetized following the literature procedure [32]. To a solution of **186** (0.500 g, 1.73 mmol) in CHCl_3_ (20.0 mL) trifluoromethanesulfonic acid (1.30 g, 8.67 mmol) was added. The reaction was refluxed for 2 h and then quenched with a solution of saturated NaHCO_3_ (20.0 mL). The organic phase was separate, dried (Na_2_SO_4_), filtered, and concentrated under reduced pressure to give the crude compound. Crystallization from MeOH afforded 0.44 g (94%) of **191** as red crystals. Mp = 178–180 °C; ^1^H NMR (400 MHz, CDCl_3_) δ 7.18 (d, *J* = 8.1 Hz, 1H), 7.10 (s, 1H), 7.07 (d, *J* = 2.5 Hz, 1H), 6.87–6.81 (m, 2H), 3.96 (s, 3H), 3.88 (s, 3H), 3.80 (s, 3H); ^13^C NMR (101 MHz, CDCl_3_) δ 192.88, 160.23, 154.68, 148.75, 140.15, 136.55, 136.19, 126.74, 119.98, 118.83, 109.70, 107.30, 102.90, 56.28, 56.18, 55.65.


*2,3,7-Trihydroxy-9H-fluoren-9-one (*
**192**
*)*


Following general procedure B, **191** (0.12 g, 0.44 mmol) was reacted to afford 0.16 g (95%) of **192** as brown powder. Mp = 340–342 °C; ^1^H NMR (400 MHz, DMSO-d_6_) δ 10.28–9.52 (m, 2H), 9.35 (br s, 1H), 7.25 (d, *J* = 7.8 Hz, 1H), 6.89 (s, 1H), 6.86 (s, 1H), 6.79 (d, *J* = 2.2 Hz, 1H), 6.78–6.71 (m, 1H); ^13^C NMR (101 MHz, DMSO-d_6_) δ 192.57, 158.12, 152.71, 145.11, 139.20, 136.71, 134.94, 125.47, 121.08, 120.06, 111.86, 111.13, 108.22; HRMS (ESI) m/z: 227.0359 [M − H]^−^, calculated for C_13_H_7_O_4_ 227.0350.


**2,3,6-trihydroxy-9H-fluoren-9-one (194)**



*Methyl 3′,4′,5-trimethoxy-[1,1′-biphenyl]-2-carboxylate (*
**187**
*)*


Following general procedure D, methyl 2-bromo-4-methoxybenzoate (**184**) (0.60 g, 2.44 mmol) and **157** (0.49 g, 2.69 mmol) were reacted and purified by flash chromatography to afford 0.62 g (84%) of **187** as a colorless oil. ^1^H NMR (400 MHz, CDCl_3_) δ 7.84 (d, *J* = 8.6 Hz, 1H), 6.93–6.80 (m, 5H), 3.92 (s, 3H), 3.88 (s, 3H), 3.87 (s, 3H), 3.65 (s, 3H); ^13^C NMR (101 MHz, CDCl_3_) δ 168.52, 161.63, 148.36, 144.82, 134.24, 132.18, 122.85, 120.51, 116.28, 112.24, 111.74, 110.63, 55.89, 55.85, 55.47, 51.77.


*2,3,6-Trimethoxy-9H-fluoren-9-one (*
**193**
*)*


The compound was synthetized following the literature procedure. [32] To a solution of **187** (0.200 g, 0.69 mmol) in CHCl_3_ (20.0 mL), trifluoromethanesulfonic acid (0.52 g, 3.47 mmol) was added. The reaction was refluxed for 2 h and then quenched with a solution of saturated NaHCO_3_ (20.0 mL). The organic phase was separated, dried (Na_2_SO_4_), filtered, and concentrated under reduced pressure to give the crude compound. Crystallization from MeOH afforded 0.12 g (64%) of **193** as orange crystals. Mp = 173–175 °C; ^1^H NMR (400 MHz, CDCl_3_) δ 7.48 (dd, *J* = 8.2, 1.5 Hz, 1H), 7.14 (s, 1H), 6.93 (d, *J* = 1.9 Hz, 1H), 6.90–6.78 (m, 1H), 6.61 (dd, *J* = 8.2, 2.1 Hz, 1H), 3.99 (s, 3H), 3.91 (s, 3H), 3.87 (s, 3H); ^13^C NMR (101 MHz, CDCl_3_) δ 191.95, 165.14, 154.03, 149.83, 146.49, 138.03, 128.13, 127.58, 125.59, 111.22, 106.82, 106.68, 103.36, 56.33, 56.22, 55.70.


*2,3,6-trihydroxy-9H-fluoren-9-one (*
**194**
*)*


Following general procedure B, **193** (0.12 g, 0.44 mmol) was reacted to afford 0.75 g (74%) of **194** as a brown powder. Mp> 350 °C; ^1^H NMR (400 MHz, DMSO-d_6_) δ 10.36 (br s, 1H), 9.86 (br s, 1H), 9.55 (br s, 1H), 7.25 (d, *J* = 8.0 Hz, 1H), 6.97 (s, 1H), 6.87 (s, 1H), 6.84 (d, *J* = 2.1 Hz, 1H), 6.49 (dd, *J* = 8.0, 2.1 Hz, 1H); ^13^C NMR (101 MHz, DMSO-d_6_) δ 191.36, 164.12, 151.68, 147.28, 146.45, 136.65, 127.00, 126.07, 125.63, 113.56, 111.29, 108.99, 107.98; HRMS (ESI) m/z: 227.0355 [M − H]^−^, calculated for C_13_H_7_O_4_ 227.0350.


**2,3,6,7-tetrahydroxy-9H-fluoren-9-one (196)**



*Methyl 3′,4,4′,5-tetramethoxy-[1,1′-biphenyl]-2-carboxylate (*
**188**
*)*


Following general procedure D, **93** (0.60 g, 2.18 mmol) and **157** (0.50, 2.74 mmol), were reacted and purified by flash chromatography to afford 0.62 g (86%) of **188** as a yellow solid. Mp = 123–125 °C; ^1^H NMR (400 MHz, CDCl_3_) δ 7.39 (s, 1H), 6.90 (d, *J* = 8.2 Hz, 1H), 6.85 (dd, *J* = 8.2, 2.0 Hz, 1H), 6.81 (d, *J* = 1.5 Hz, 2H), 3.96 (s, 3H), 3.93 (s, 3H), 3.92 (s, 3H), 3.88 (s, 3H), 3.64 (s, 3H); ^13^C NMR (101 MHz, CDCl_3_) δ 168.62, 151.21, 148.47, 148.32, 147.75, 137.04, 134.49, 122.28, 120.77, 113.67, 112.84, 112.11, 110.77, 56.29, 56.22, 56.06, 56.02, 52.04.


*2,3,6,7-Tetramethoxy-9H-fluoren-9-one (*
**195**
*)*


The compound was synthetized following the literature procedure. [32] To a solution of **188** (0.3 g, 0.90 mmol) in CHCl_3_ (20.0 mL) trifluoromethanesulfonic acid (0.68 g, 4.51 mmol) was added. The reaction was refluxed for 2 h and then quenched with a solution of saturated NaHCO_3_ (20.0 mL). The organic phase was separate, dried (Na_2_SO_4_), filtered, and concentrated under reduced pressure to give the crude compound. Crystallization from MeOH afforded 0.15 g (54%) of **195** red crystals. Mp = 201–203 °C; ^1^H NMR (400 MHz, CDCl_3_) δ 7.12 (s, 2H), 6.87 (s, 2H), 4.00 (s, 6H), 3.90 (s, 6H); ^13^C NMR (101 MHz, CDCl_3_) δ 192.61, 154.03, 148.85, 138.81, 127.22, 107.34, 102.95, 56.30.


*2,3,6,7-tetrahydroxy-9H-fluoren-9-one (*
**195**
*)*


Following general procedure B, **194** (0.15 g, 0.56 mmol) was reacted to afford 0.082 g (70%) of **195** as dark-brown powder. Mp = 346–348 °C; ^1^H NMR (400 MHz, DMSO-d_6_) δ 9.72 (s, 2H), 9.26 (s, 2H), 6.79 (s, 2H), 6.79 (s, 2H); ^13^C NMR (101 MHz, DMSO-d_6_) δ 191.99, 151.57, 144.95, 137.79, 126.40, 111.55, 108.16; HRMS (ESI) m/z: 243.0311 [M − H]^−^, calculated for C_13_H_7_O_5_ 243.0299.


**7-Hydroxy-2,3-dimethoxy-9H-fluoren-9-one (197)**



*Methyl 4′-hydroxy-4,5-dimethoxy-[1,1′-biphenyl]-2-carboxylate (*
**189**
*)*


Following general procedure D, **9** (0.55 g, 2.00 mmol), 4-hydroxyphenylboronic acid (**185**) (0.30 g, 2.20 mmol), were reacted and purified by flash chromatography to afford 0.46 g (80%) of **189** as a white solid. Mp = 189–191 °C; ^1^H NMR (400 MHz, DMSO-d_6_) δ 9.42 (s, 1H), 7.22 (s, 1H), 7.05 (d, *J* = 8.5 Hz, 2H), 6.83 (s, 1H), 6.74 (d, *J* = 8.6 Hz, 2H), 3.81 (s, 3H), 3.78 (s, 3H), 3.54 (s, 3H); ^13^C NMR (101 MHz, DMSO-d_6_) δ 168.69, 156.98, 151.20, 147.45, 136.11, 131.79, 129.85, 122.25, 115.25, 113.94, 113.01, 56.13, 56.08, 52.04.

7-Hydroxy-2,3-dimethoxy-9H-fluoren-9-one (**197**)

To a solution of **189** (0.29 g, 1.00 mmol) in CHCl_3_ (10.0 mL) was added trifluoromethanesulfonic acid (0.75 g, 5.00 mmol). The reaction was refluxed for 12 h and then quenched with a solution of saturated NaHCO_3_ (20.0 mL). The organic phase was separates, dried (Na_2_SO_4_), filtered, and concentrated under reduced pressure to give the crude compound. Crystallization from MeOH afforded 0.09 g (35%) of **197** as a red solid. Mp = 257–259 °C; ^1^H NMR (400 MHz, DMSO-d_6_) δ 9.84 (s, 1H), 7.41 (d, *J* = 7.9 Hz, 1H), 7.26 (s, 1H), 7.05 (s, 1H), 6.83 (d, *J* = 2.3 Hz, 1H), 6.80 (dd, *J* = 7.9, 2.4 Hz, 1H), 3.88 (s, 3H), 3.76 (s, 3H); ^13^C NMR (101 MHz, DMSO-d_6_) δ 192.71, 158.51, 155.34, 148.71, 140.62, 136.37, 134.68, 125.83, 121.70, 120.19, 111.38, 107.74, 104.60, 56.55, 56.23; HRMS (ESI) m/z: 255.0663 [M − H]^−^, calculated for C_15_H_11_O_4_ 255.0651.


**7-Hydroxy-9H-fluoreno[2,3-d][1,3]dioxol-9-one (198)**



*Methyl 6-(4-hydroxyphenyl)benzo[d][1,3]dioxole-5-carboxylate (*
**190**
*)*


Following general procedure D, **142** (0.26 g, 1.00 mmol), 4-hydroxyphenylboronic acid (**185**) (0.15 g, 1.10 mmol), were reacted and purified by flash chromatography to afford 0.46 g (80%) of **189** as a white solid. Mp = 135–137 °C; ^1^H NMR (400 MHz, CDCl_3_) δ 7.30 (s, 1H), 7.08 (d, *J* = 8.5 Hz, 2H), 6.79–6.70 (m, 3H), 6.33–6.22 (m, 1H), 6.03 (s, 2H), 3.68 (s, 3H); ^13^C NMR (101 MHz, CDCl_3_) δ 168.90, 155.32, 150.13, 146.44, 139.06, 133.24, 129.50, 123.28, 115.13, 111.03, 109.81, 101.88, 52.20.


*7-Hydroxy-9H-fluoreno[2,3-d][1,3]dioxol-9-one (*
**198**
*)*


The compound was synthetized following the literature procedure. [32] To a solution of **190** (0.20 g, 0.73 mmol) in CHCl_3_ (10.0 mL) was added trifluoromethanesulfonic acid (0.52 g, 3.47 mmol). The reaction was refluxed for 12 h and then quenched with a solution of saturated NaHCO_3_ (20.0 mL). The organic phase was separated, dried (Na_2_SO_4_), filtered, and concentrated under reduced pressure to give the crude compound. Crystallization from MeOH afforded 0.04 g (23%) of **198** as a red solid. Mp = 298–300 °C; ^1^H NMR (400 MHz, DMSO-d_6_) δ 9.89 (s, 1H), 7.38 (d, *J* = 7.9 Hz, 1H), 7.24 (s, 1H), 7.01 (s, 1H), 6.87–6.73 (m, 2H), 6.09 (s, 2H); ^13^C NMR (101 MHz, DMSO-d_6_) δ 192.01, 158.68, 153.89, 147.50, 142.97, 136.22, 134.23, 127.66, 121.87, 120.35, 111.39, 104.96, 102.75, 102.19; HRMS (ESI) m/z: 239.0350 [M − H]^−^, calculated for C_14_H_7_O_4_ 239.0359.


**9H-Fluorene-2,3,7-triol (202)**



*2,3,7-tris(benzyloxy)-9H-fluoren-9-one (*
**199**
*)*


Following general procedure K, **1** (0.30 g, 1.31 mmol) was reacted to afford 0.63 g (96%) of **199** as a red solid. Mp = 141–143 °C; ^1^H NMR (400 MHz, CDCl_3_) δ 7.50–7.28 (m, 14H), 7.23 (s, 1H), 7.21–7.14 (m, 2H), 6.98–6.90 (m, 2H), 5.26 (s, 2H), 5.16 (s, 2H), 5.08 (s, 2H); ^13^C NMR (101 MHz, CDCl_3_) δ 192.69, 159.41, 154.69, 148.48, 140.32, 136.64, 136.62, 136.50, 136.39, 136.33, 128.65, 128.63, 128.52, 128.12, 127.96, 127.46, 127.30, 127.12, 120.20, 120.08, 110.92, 110.60, 105.78, 71.37, 71.15, 70.40.


*2,3,7-Tris(benzyloxy)-9H-fluoren-9-ol (*
**200**
*)*


To a solution of **199** (0.65 g, 1.30 mmol) in THF (20.0 mL) and MeOH (20.0 mL) at 0 °C, NaBH_4_ (0.10 g, 2.60 mmol) was added. The mixture was stirred at room temperature for 2 h. The reaction was then quenched with water (20.0 mL). The mixture was extracted with CH_2_Cl_2_ (3 × 20.0 mL), dried over Na_2_SO_4_, and the solvent removed under reduced pressure to afford 0.51 g (78%) of **200** as a white solid. Mp = 145–147 °C; ^1^H NMR (400 MHz, CDCl_3_) δ 7.59–7.26 (m, 16H), 7.21–7.13 (m, 2H), 7.07 (s, 1H), 6.93 (dd, *J* = 8.3, 2.4 Hz, 1H), 5.37–5.26 (m, 1H), 5.14 (d, *J* = 3.8 Hz, 2H), 5.11 (s, 2H), 5.03 (s, 2H), 2.26 (s, 1H); ^13^C NMR (101 MHz, CDCl_3_) δ 158.38, 149.95, 148.34, 147.85, 138.53, 137.18, 136.91, 133.50, 133.04, 128.59, 128.51, 128.51, 128.48, 127.99, 127.85, 127.82, 127.49, 127.39, 127.34, 119.74, 115.59, 112.07, 111.72, 106.32, 74.89, 71.54, 71.48, 70.25, 30.37.


*2,3,7-tris(benzyloxy)-9-methoxy-9H-fluorene (*
**201**
*)*


The compound was synthetized following the literature procedure [33]. To a solution of **200** (0.30 g, 0.60 mmol) in MeOH (30.0 mL) was added H_2_SO_4_ (0.5 mL). The mixture was refluxed for 1 h and then concentrated under reduced pressure to afford 0.28 g (91%) of **201** as a white solid. Mp = 131–133 °C; ^1^H NMR (400 MHz, CDCl_3_) δ 7.58–7.42 (m, 7H), 7.43–7.29 (m, 11H), 7.20 (d, *J* = 2.4 Hz, 1H), 7.18 (d, *J* = 0.7 Hz, 1H), 7.17 (s, 1H), 6.97 (dd, *J* = 8.2, 2.4 Hz, 1H), 5.44 (s, 1H), 5.23 (s, 2H), 5.20 (s, 2H), 5.11 (s, 2H), 2.96 (s, 3H); ^13^C NMR (101 MHz, CDCl_3_) δ 158.28, 150.31, 148.19, 144.55, 137.23, 137.19, 136.90, 135.02, 134.73, 134.18, 128.58, 128.51, 128.44, 127.98, 127.85, 127.80, 127.53, 127.47, 127.35, 119.72, 115.55, 112.78, 112.12, 106.24, 71.68, 71.59, 70.34, 51.73.


*9H-Fluorene-2,3,7-triol (*
**202**
*)*


Following general procedure H, **201** (0.270 g, 0.52 mmol) was reacted to afford 0.075 g (67%) of **202** as a brown solid. Mp = 282–284 °C; ^1^H NMR (400 MHz, DMSO-d_6_) δ 9.17 (s, 1H), 8.69 (s, 2H), 7.35 (d, *J* = 8.2 Hz, 1H), 7.01 (s, 1H), 6.85 (q, *J* = 1.0 Hz, 2H), 6.67 (dd, *J* = 8.2, 2.3 Hz, 1H), 3.57 (s, 2H); ^13^C NMR (101 MHz, DMSO-d_6_) δ 155.98, 145.10, 144.89, 144.22, 133.79, 133.51, 133.42, 119.47, 113.96, 112.66, 112.49, 106.44, 36.19; HRMS (ESI) m/z: 213.0557 [M − H]^−^, calculated for C_13_H_9_O_3_ 213.0561


**(3,4-Dihydroxyphenyl)(3-hydroxyphenyl)methanone (209)**



*(3,4-Dimethoxyphenyl)(3-methoxyphenyl)methanone (*
**206**
*)*


The compound was synthetized following the literature procedure [34]. A solution of 3-methoxybenzoic acid (**203**) (0.82 g, 5.42 mmol) in SOCl_2_ (10.0 mL) was refluxed for 2 h. The solvent was removed under reduced pressure, washed with dichloromethane, and concentrated again, twice. The substituted benzoyl chloride was isolated as a yellow oil in quantitative yield. This was transferred into a round-bottomed flask and diluted with dichloromethane (20.0 mL). Aluminum trichloride (0.53 g, 3.98 mmol) was then added and the mixture was cooled to 0 °C. 1,2-Dimethoxybenzene (**205**) (0.50 g, 3.62 mmol) was added dropwise, and the mixture was refluxed for 2 h. The temperature was cooled to room temperature and the organic layer was washed with ice water (30.0 mL), 2 M HCl (20.0 mL), and saturated aqueous sodium hydrogen carbonate (20.0 mL), dried over anhydrous sodium sulfate, and concentrated under reduced pressure to give a yellow oil. Purification by flash chromatography (90/10 pentane/EtOAc) afforded 0.62 g (63%) of **206** as a colorless oil. ^1^H NMR (400 MHz, CDCl_3_) δ 7.49 (d, *J* = 2.0 Hz, 1H), 7.41–7.34 (m, 2H), 7.30 (ddd, *J* = 5.2, 2.6, 1.3 Hz, 2H), 7.11 (ddd, *J* = 8.1, 2.6, 1.2 Hz, 1H), 6.89 (d, *J* = 8.4 Hz, 1H), 3.96 (s, 3H), 3.94 (s, 3H), 3.85 (s, 3H); ^13^C NMR (101 MHz, CDCl_3_) δ 195.30, 159.44, 153.01, 148.95, 139.57, 130.16, 129.08, 125.48, 122.34, 118.19, 114.21, 112.08, 109.70, 56.08, 56.04, 55.44.


*(3,4-Dihydroxyphenyl)(3-hydroxyphenyl)methanone (*
**209**
*)*


Following general procedure B, **206** (0.27 g, 1.00 mmol) was reacted to afford 0.21 g (91%) of **209** as yellow powder. Mp = 193–195 °C; ^1^H NMR (400 MHz, DMSO-d_6_) δ 9.84 (br s, 1H), 9.73 (br s, 1H), 9.39 (br s, 1H), 7.29 (t, *J* = 7.8 Hz, 1H), 7.21 (d, *J* = 2.1 Hz, 1H), 7.09 (dd, *J* = 8.2, 2.1 Hz, 1H), 7.06–6.92 (m, 3H), 6.83 (d, *J* = 8.2 Hz, 1H); ^13^C NMR (101 MHz, DMSO-d_6_) δ 194.71, 157.53, 150.97, 145.50, 140.04, 129.78, 128.74, 123.73, 120.33, 119.09, 117.32, 116.03, 115.49; HRMS (ESI) m/z: 243.0514 [M − H]^−^, calculated for C_13_H_11_O_4_ 229.0506.


**(3,4-Dihydroxyphenyl)(4-hydroxyphenyl)methanone (210)**



*(3,4-Dimethoxyphenyl)(4-methoxyphenyl)methanone (*
**207**
*)*


The compound was synthetized following the literature procedure [34]. A solution of 4-methoxybenzoic acid (**204**) (0.82 g, 5.42 mmol) in SOCl_2_ (10.0 mL) was refluxed for 2 h. The solvent was removed under reduced pressure, washed with dichloromethane, and concentrated again, twice. The substituted benzoyl chloride was isolated as a yellow oil in quantitative yield. This was transferred into a round-bottomed flask and diluted with dichloromethane (20.0 mL). Aluminum trichloride (0.53 g, 3.98 mmol) was then added and the mixture was cooled to 0 °C. **205** (0.50 g, 3.62 mmol) was added dropwise and the mixture was refluxed for 2 h. The temperature was cooled to room temperature and the organic layer was washed with ice water (30.0 mL), 2 M HCl (20.0 mL), and saturated aqueous sodium hydrogen carbonate (20.0 mL), dried over anhydrous sodium sulfate, and concentrated under reduced pressure to give a yellow oil. Crystallization from MeOH afforded 0.61 g (62%) of 207 as a white powder. Mp = 87–89 °C; ^1^H NMR (400 MHz, CDCl_3_) δ 7.84–7.74 (m, 2H), 7.43 (d, *J* = 2.0 Hz, 1H), 7.36 (dd, *J* = 8.3, 1.9 Hz, 1H), 7.01–6.92 (m, 2H), 6.90 (d, *J* = 8.3 Hz, 1H), 3.96 (s, 3H), 3.94 (s, 3H), 3.89 (s, 3H); ^13^C NMR (101 MHz, CDCl_3_) δ 194.46, 162.82, 152.55, 148.87, 132.21, 130.79, 130.70, 124.80, 113.43, 112.20, 109.69, 56.06, 56.02, 55.46.


*(3,4-Dihydroxyphenyl)(4-hydroxyphenyl)methanone (*
**210**
*)*


Following general procedure B, **207** (0.27 g, 1.00 mmol) was reacted to afford 0.20 g (88%) of **210** as yellow powder. Mp = 202–204 °C; ^1^H NMR (400 MHz, DMSO-d_6_) δ 7.63–7.49 (m, 2H), 7.16 (d, *J* = 2.1 Hz, 1H), 7.04 (dd, *J* = 8.1, 2.0 Hz, 1H), 6.89–6.82 (m, 2H), 6.81 (d, *J* = 8.2 Hz, 1H); ^13^C NMR (101 MHz, DMSO-d_6_) δ 193.54, 161.52, 150.24, 145.40, 132.38, 129.52, 129.39, 123.19, 117.26, 115.35; HRMS (ESI) m/z: 243.0512 [M − H]^−^, calculated for C_13_H_11_O_4_ 229.0506.


**Bis(3,4-dihydroxyphenyl)methanone (211)**



*Bis(3,4-dimethoxyphenyl)methanone (*
**208**
*)*


The compound was synthetized following the literature procedure. [34] A solution of **7** (1.00 g, 5.49 mmol) in SOCl_2_ (10.0 mL) was refluxed for 2h. It was concentrated in vacuo, washed with dichloromethane, and concentrated again, twice. The substituted benzoyl chloride was isolated as a yellow oil in quantitative yield. This was transferred into a round-bottomed flask and diluted with dichloromethane (40.0 mL). Aluminum trichloride (0.80 g, 6.03 mmol) was then added and the mixture was cooled to 0 °C. Compound **205** (0.76 g, 5.48 mmol) was added dropwise and the mixture was refluxed for 2 h. The temperature was cooled to room temperature and the organic layer was washed with ice water (30 mL), 2 M HCl (20 mL) and saturated aqueous sodium hydrogen carbonate (20.0 mL), dried over anhydrous sodium sulfate, and concentrated in vacuo to give a yellow oil. Purification by flash chromatography (90/10 pentane/EtOAc) afforded 1.05 g (63%) of **208** as a white solid. Mp = 149–151 °C; ^1^H NMR (400 MHz, cdcl_3_) δ 7.42 (d, *J* = 2.1 Hz, 2H), 7.37 (ddd, *J* = 8.3, 2.0, 0.6 Hz, 2H), 6.89 (d, *J* = 8.3 Hz, 2H), 3.95 (s, 6H), 3.93 (s, 6H).


*Bis(3,4-dihydroxyphenyl)methanone (*
**211**
*)*


Following general procedure B, **208** (0.20 g, 0.66 mmol) was reacted to afford 0.14 g (89%) of **211** as yellow powder. Mp = 244–246 °C; ^1^H NMR (400 MHz, DMSO-d_6_) δ 9.69 (s, 2H), 9.31 (s, 2H), 7.14 (d, *J* = 2.1 Hz, 2H), 7.04 (dd, *J* = 8.2, 2.1 Hz, 2H), 6.81 (d, *J* = 8.2 Hz, 2H); ^13^C NMR (101 MHz, DMSO-d_6_) δ 193.61, 150.08, 145.29, 129.70, 123.06, 117.29, 115.30; HRMS (ESI) m/z: 245.0462 [M − H]^−^, calculated for C_13_H_12_O_5_ 245.0455.

### 3.2. Kinase Assay

The assay was performed externally at BPS Bioscience (San Diego, CA, USA). Pyruvate kinase (PKL and PKR) reactions were conducted in triplicate at room temperature for 30 min in a 25 µL mixture containing 50 mM tris, pH 7.4, 10 mM MgCl_2_, 100 mM KCl, 0.05% Tween, 0.1 mM ADP, 0.125 mM PEP, pyruvate kinase (see Appendix A), and the test compounds (see Appendix A). The final DMSO concentration in the reaction was 1% *v/v*. After enzymatic reactions, 25 μL of Kinase-Glo Max reagent was added to each well and luminescence was measured using a BioTek SynergyTM 2 microplate reader. The Kinase-Glo Max luminescence assay kit measures PK activity by quantitating the amount of ATP produced following a PK reaction. Enzyme activity assays were performed in triplicate at each concentration. The luminescence data were analyzed using GraphPad Prism. In the absence of the compound, the intensity (Ce) in each data set was defined as 100% activity. In the absence of enzyme, the intensity (C0) in each data set was defined as 0% activity. The percent activity in the presence of each compound was calculated according to the following equation: % activity ¼ (CeC0)/(CeeC0), where C ¼ is the luminescence in the presence of the compound. The values of % activity versus a series of compound concentrations were plotted using non-linear regression analysis of the sigmoidal dose–response curve generated with the equation YBþ(T−B)/1 + 10^((LogEC50_X)_Hill Slope)^, where Y ¼ percent activity, B ¼ minimum percent activity, T ¼ maximum percent activity, X ¼ logarithm of compound, and Hill Slope ¼ slope factor or Hill coefficient. The IC_50_ value was determined by the concentration causing a half-maximal percent activity.

## 4. Conclusions

We explored a series of pyruvate kinase inhibitors based on the urolithin C structure. The synthesis and evaluation of PKL/PKR inhibitors were described for 53 first-in-class compounds. The lactone group seems to be the most promising site for further chemical modifications and for the development of new active derivatives.

Unfortunately, no crystal structure was obtained with the enzyme co-crystallized with these structures and more work is needed in the future. However, from these preliminary data, it seems plausible that urolithin C derivatives bind in the interface of the tetrameric structure of PKL. This binding could explain not only their non-competitive inhibition but also their shift to activators due to small chemical modifications. The results presented here are the first results that explore this class of molecule.

## Data Availability

Data is contained within the article or Appendix A.

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
