# Peer review of "Exploration of Novel Urolithin C Derivatives as Non-Competitive Inhibitors of Liver Pyruvate Kinase"

_pharmaceuticals, 2023, doi:10.3390/ph16050668_

Round 1

Reviewer 1 Report

This is a routine work, but it is well executed and presented and will be useful to the scientific community 

The work is quite standard and routine as I said in my short review so, it is not very original but has some value as it sheds some light on possible physico-chemical features of small molecules that could be potential inhibitors of  liver pyruvate kinase.   

The work has two parts, experimental (very routine) and computational (qsar), also very routine. They basically did a qsar (routine but at least not bindly clicks on buttons as many published qsar studies) and then synthesized over 50 analogues of Urolithin C which were later tested by an external company.   

Again, a very very standar and routine work, but with some value, I would say it may be published in pharmaceuticals, I would not recommend it for another journal with better ranking but pharmaceuticals accepts papers like this regularly, so I would say it is acceptable as it is.  

The work is well presented, well written and it is easy to follow.   

Hope this help with the final decision, 

Reviewer 2 Report

The article “Exploration of Novel Urolithin C Derivatives as non-competitive inhibitors of Liver Pyruvate Kinase” submitted by Grotli and co-workers, describes the synthesis of Urolithin analogs. These analogs provide detailed insight into the structural influence of the inhibition of PKL and PKR. This work is based on their previous finding, where Ellagic acid analog (Urolithin C, 1) was a non-competitive inhibitor against PKL. The article is well-written, and the numbering is correct. 

In this manuscript, 54 analogs were tested for their biological activities. 

It was found that compound 82 possesses the most promising biological profile.

As no therapy is currently available to treat NAFLD, this work should be of great interest to this journal's audience.

As the authors put great effort into studying the SAR, it would be helpful for the readers if they could write a summary paragraph before the “Material and methods” section.

It would be helpful if the author could write a summary paragraph at the end of the discussion.

I recommend accepting this manuscript with minor revisions.
